# AGENTRACE: BENCHMARKING EFFICIENCY IN LLM AGENT FRAMEWORKS

## ABSTRACT

Large Language Model (LLM) agents are rapidly gaining traction across domains such as intelligent assistants, programming aids, and autonomous decision systems. While existing benchmarks focus primarily on evaluating the effectiveness of LLM agents, such as task success rates and reasoning correctness, the efficiency of agent frameworks remains an underexplored but critical factor for real-world deployment. In this work, we introduce AgentRace, the first benchmark specifically designed to systematically evaluate the efficiency of LLM agent frameworks across representative workloads. AgentRace enables controlled, reproducible comparisons of runtime performance, scalability, communication overhead, and tool invocation latency across popular frameworks on diverse task scenarios and workflows. Our in-depth experiments reveal 14 insights and 15 underlying mechanisms for developing efficient LLM agents. We believe AgentRace will become a valuable resource for guiding the design and optimization of next-generation efficient LLM agent systems. The platform and results are available at the anonymous website `https://agent-race.github.io/`.

## 1 INTRODUCTION

Large Language Models (LLMs) (OpenAI, 2023; Touvron et al., 2023; Liu et al., 2024a; Naveed et al., 2023; Bai et al., 2023) have rapidly gained widespread popularity due to their exceptional capabilities in natural language understanding and generation, significantly impacting various applications including chatbots, content creation, and programming assistants. With these advancements, LLM agents (Wang et al., 2024; Guo et al., 2024; Zhao et al., 2024; Zhang et al., 2024; Ni & Buehler, 2024), which are autonomous entities powered by LLMs capable of executing complex tasks through intelligent interactions, have emerged as a promising area of research and practical implementation.

To accelerate the development of LLM agents, numerous benchmarks and datasets (Andriushchenko et al., 2024; Chang et al., 2024; Huang et al., 2023; Shen et al., 2024) have been proposed to assess LLM agents, primarily focusing on evaluating their effectiveness and reliability in task completion. These benchmarks typically measure task success rates, correctness of generated outputs, overall functional capabilities, and safety of agents.

However, for LLM agents to be widely deployed in real-world scenarios in the future, the efficiency of their frameworks is critically important. Efficient execution, scalability, and minimal communication overhead are essential for ensuring timely responses and practical usability, particularly in resource-constrained and latency-sensitive environments. Despite the proliferation of LLM agent frameworks, such as LangChain (LangChain, 2025), AutoGen (Wu et al., 2023), and AgentScope (Gao et al., 2024), a systematic benchmark evaluating these frameworks' performance efficiency remains absent.

To bridge this significant gap, we introduce **AgentRace**, the first efficiency-focused benchmark platform for LLM agent frameworks, including cost, computational, and communication efficiency. AgentRace enables controlled, reproducible comparisons across frameworks and workflows, aiming to answer the following key research questions:

1. *What are the primary efficiency bottlenecks in current LLM agent frameworks (e.g., model inference latency, tool calling overhead)?*

2. *What caused the inefficiency of existing LLM agent frameworks?*

3. *How to improve the efficiency of agent execution?*

AgentRace features a modular and extensible design. It supports **7** LLM agent frameworks, **12** types of tools, **3** commonly used workflows, **6** task scenarios, and **4** metrics. The benchmark can be executed with a single command line, facilitating rapid experimentation and reproducibility. We conduct a comprehensive assessment of the efficiency of popular LLM agent frameworks and reveal **14** insights and **15** underlying mechanisms for developing efficient LLM agents. The platform and results are made available through an anonymous website `https://agent-race.github.io/`.

In summary, our contributions include:

- We introduce AgentRace, the first benchmark platform that systematically evaluates the efficiency of LLM agent frameworks with modular design, filling a critical gap left by existing benchmarks that primarily focus on task success or reasoning correctness.
- We conduct a comprehensive and in-depth assessment of efficiency across frameworks, revealing previously undocumented sources of inefficiency.
- We provide actionable insights for both practitioners and researchers to optimize the deployment of efficient LLM-based agents.
- We release the entire benchmark suite and experimental results, providing a platform to identify the efficiency issues of LLM agents.

## 2 BACKGROUND AND RELATED WORK

### 2.1 LLM AGENTS

LLMs agents (Yao et al., 2023; Zhao et al., 2024) are systems that combine the generative capabilities of LLMs with additional components such as memory, planning, and tool usage to perform complex tasks autonomously. These agents can interpret user inputs, plan actions, interact with external tools, and adapt based on feedback, enabling more dynamic and context-aware behaviors. Many agents have been developed, where some are generic agents that are designed to execute general tasks and some are specialized agents for some concrete task. For example, ReAct (Yao et al., 2023) is a typical general agent workflow, where the agent thinks and take actions interatively. MetaGPT (Hong et al., 2023) is an agent designed for software development, where each agent plays a different role to simulate a software company. In this work, we aim to evaluate the efficiency of different LLM agent frameworks, thus focusing on using the widely used general agent workflows.

Several recent studies (Wang et al., 2025b; Chen et al., 2025; Wang et al., 2025c) have examined efficiency from a system-design perspective. For example, the OPPO AI Agent Team analyzes the effectiveness-efficiency trade-off of agent pipelines and introduces "cost-of-pass" as a metric to quantify computational cost (Wang et al., 2025b). OPTIMA (Chen et al., 2025) focuses on multi-agent settings and evaluates token efficiency and communication overhead enabled by optimized training strategies. These studies highlight the importance of efficiency in LLM agents, reinforcing the motivation of our work to benchmark efficiency across widely used agent frameworks.

### 2.2 LLM AGENT FRAMEWORKS

The development and deployment of LLM agents have been facilitated by various frameworks that provide tools and abstractions for building agentic systems. There have been many LLM agent frameworks. For example, LangChain (LangChain, 2025) offers a modular framework for developing applications with LLMs, supporting integrations with various data sources and tools. It provides a low-level agent orchestration framework, a purpose-built deployment platform, and debugging tools. Besides LangChain, there are also many other popular LLM agent frameworks. In our platform, we select some popular and easy-to-use frameworks for integration. For the detailed introduction of these frameworks, please refer to Section 3.1.

### 2.3 BENCHMARKS FOR LLM AGENTS

There have been many benchmarks for LLM agents (Andriushchenko et al., 2024; Chang et al., 2024; Huang et al., 2023; Shen et al., 2024; Liu et al., 2024b). However, most of these benchmarks

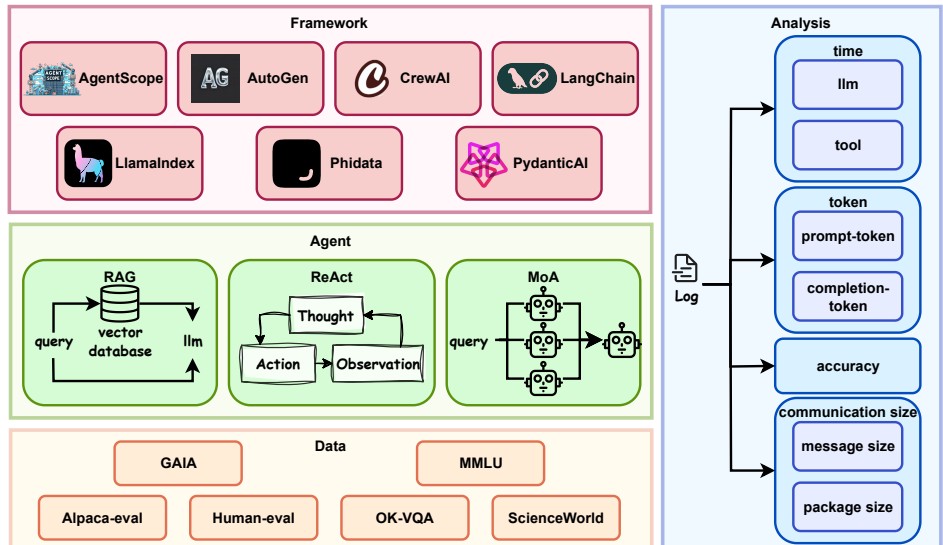

Figure 1: The architecture of AgentRace.

usually focus on ability or trustworthiness perspectives, and do not exploit the efficiency part. For example, AgentBench (Liu et al., 2024b) report *Step Success Rate* as the main metric showing the independent accuracy of each action step, due to the current struggles for LLMs to ensure overall task success rates. Beyond benchmarks focusing solely on success rates, AgentBoard (Chang et al., 2024) proposes a comprehensive evaluation framework for LLM agents. It introduces a fine-grained *Progress Rate* metric to track incremental advancements during task execution, along with an open-source toolkit for multi-faceted analysis. WORFBENCH (Huang et al., 2023) introduces a unified framework for evaluating workflow generation, including both linear and graph-structured workflows. Its evaluation metric, WORFEVAL, quantifies generation performance across these tasks. Although the benchmark measures end-to-end efficiency through *Task Execution Time*, it omits a detailed breakdown of computational costssuch as tool execution latency. This lack of granularity obscures potential bottlenecks in workflow optimization. MASArena (MAS, 2025) provides a convenient multi-dimensional framework for agent evaluation, but it lacks a unified implementation for diverse workflows and heterogeneous tool integrations. Moreover, its evaluation benchmarks are limited to domains such as mathematics, code, and textual reasoning.

## 3 DESIGN OF AGENTRACE

### 3.1 MODULES

To systematically evaluate the efficiency and scalability of LLM agent frameworks, we introduce a modular benchmark platform AgentRace. As shown in Figure 1, this platform comprises four interconnected modules, including **Data**, **Agent**, **Framework**, and **Analysis**, designed to capture diverse agent frameworks, execution workflows, task complexities, and performance analysis.

**Data Module: Diverse Task Coverage**  The Data module defines the core tasks used in our benchmark and plays a critical role in ensuring that LLM agent frameworks are evaluated across a wide range of real-world scenarios. Our design is guided by two key considerations: (1) task diversity in terms of reasoning complexity, tool usage, and interaction patterns; and (2) alignment with widely adopted benchmarks to enable meaningful and comparable evaluations. We select five representative datasets that reflect varying levels of difficulty, domain coverage, and agent requirements, including **GAIA** (Mialon et al., 2023), **HumanEval** (Chen et al., 2021), **MMLU** (Hendrycks et al., 2020), **AlpacaEval** (Dubois et al., 2024), **OK-VQA** (Marino et al., 2019), and **ScienceWorld** (Wang et al., 2022). The datasets cover tool-intensive, structured reasoning, retrieval-augmented workflows, multi-agent, multi-modal scenarios and multi-step planning. The details about the datasets are available at Appendix A.1. The above coverage enables a holistic evaluation of agent frameworks

under varied demands, including tool usage, memory handling, retrieval integration, and inter-agent communication.

**Agent Module: Workflow Diversity**   The Agent module captures the diversity of reasoning patterns exhibited by modern LLM-based agents. In designing this module, our goal is to represent a wide range of real-world task execution strategies while ensuring broad compatibility with existing agent frameworks. We instantiate agents using three widely adopted and conceptually distinct workflow paradigms, including **ReAct (Reasoning and Acting)** (Yao et al., 2023), **RAG (Retrieval-Augmented Generation)**, and **MoA (Mixture of Agents)** (Wang et al., 2025a). These workflows reflect sequential prompting, retrieval-grounded answering, and distributed multi-agent collaboration. By supporting all three within our benchmark, we enable a comprehensive evaluation of agent frameworks under varying reasoning styles and system architectures. The details about the workflows are available at Appendix A.2.

**Framework Module: Broad Ecosystem Coverage**   The Framework module integrates a wide spectrum of open-source LLM agent frameworks including **LangChain** (LangChain, 2025), **AutoGen** (Wu et al., 2023), **AgentScope** (Gao et al., 2024), **CrewAI** (Lee, 2025), **LlamaIndex** (LlamaIndex, 2025), **Phidata** (agno-agi, 2025), and **PydanticAI** (PydanticAI, 2025), each with distinct design philosophies, runtime environments, and abstraction layers. In selecting the frameworks, we focus on two primary considerations: (1) their popularity and influence in the developer and research communities, and (2) the feasibility of easy deployment and integration within our benchmarking platform. In particular, our implementations are designed to extend functionalities absent from certain frameworks, while leveraging native components whenever available so as not to replace or override existing optimizations.

**Analysis Module: Measuring Efficiency**   The Analysis module defines the core metrics used to evaluate the system-level efficiency of LLM agent frameworks. We focus on three dimensions: **computational efficiency**, **cost efficiency**, and **communication efficiency**. Specifically, we measure the following four key metrics: (1) **Execution Time**: The total wall-clock time from agent invocation to task completion. This includes the full execution pipeline, including LLM inference, tool calls, etc. (2) **Token Consumption**: The total number of input and output tokens processed by the LLM during the task. This reflects the computational cost of inference and directly impacts the monetary cost in API-based deployments. (3) **Communication Size**: The total volume of data exchanged between agents. This metric captures inefficiencies in prompt formatting, serialization, and inter-agent message passing, particularly relevant in multi-agent setting. (4) **Accuracy**: To ensure correctness is preserved during efficiency evaluation, we also include a task-specific accuracy metric. This ensures that frameworks are functionally correct.

## 3.2   PIPELINE

The design of the AgentRace benchmark pipeline is illustrated in Figure 2. The pipeline is fully modular and consists of three main stages: (1) configuration, (2) execution and monitoring, and (3) analysis and visualization. In the configuration stage, users specify experimental parameters (e.g., framework, workflow, dataset, and tools) in a YAML file. The executor parses this file and instantiates the corresponding agent with unified interfaces. During execution, the agent interacts with the chosen framework and tools under controlled settings, while a monitoring layer is dynamically attached to capture runtime behavior. Finally, the analysis stage aggregates the collected traces into structured logs and performance visualizations for reproducibility and cross-framework comparison.

**Tracer and Logger**   The monitoring layer is designed to provide fine-grained yet low-overhead instrumentation. We implement two complementary components: a *logger* for recording high-level events and a *tracer* for intercepting fine-grained tool calling operations. The logger tracks each LLM inference call, data retrieval request, and inter-agent communication, capturing metadata such as token count, latency, and payload size. To address the scalability challenge of monitoring diverse tool invocations, we introduce a generic tracer wrapper, `traced_tool`, that instruments tool execution transparently. Developers can annotate a tool with a single wrapper, after which its statistics are automatically recorded. This design allows AgentRace to maintain both extensibilitynew tools can

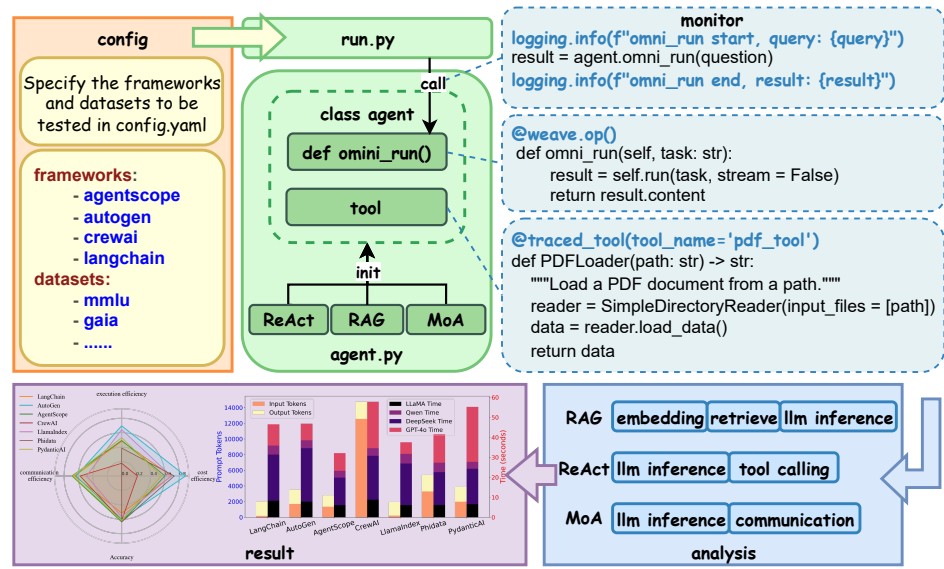

Figure 2: The pipeline of AgentRace.

Table 1: The supported functionalities of AgentRace. ✓denotes that the functionality is implemented in AgentRace. ◯ denotes that the functionality is supported in the original framework.

|  |  | LangChain | AutoGen | AgentScope | CrewAI | LlamaIndex | Phidata | PydanticAI |
|---|---|---|---|---|---|---|---|---|
| **Workflow** | ReAct | ◯ | ✓ | ◯ | ✓ | ◯ | ✓ | ✓ |
|  | RAG | ◯ | ✓ | ◯ | ✓ | ◯ | ◯ | ✓ |
|  | MoA | ✓ | ◯ | ✓ | ◯ | ◯ | ✓ | ✓ |
| **Tools** | Search | ◯ | ✓ | ◯ | ◯ | ◯ | ◯ | ✓ |
|  | PDF loader | ◯ | ✓ | ✓ | ✓ | ◯ | ✓ | ✓ |
|  | CSV reader | ◯ | ✓ | ✓ | ◯ | ◯ | ◯ | ✓ |
|  | XLSX reader | ◯ | ✓ | ✓ | ✓ | ◯ | ✓ | ✓ |
|  | Text file reader | ◯ | ✓ | ✓ | ◯ | ◯ | ◯ | ✓ |
|  | doc reader | ◯ | ✓ | ✓ | ✓ | ◯ | ✓ | ✓ |
|  | MP3 loader | ◯ | ✓ | ◯ | ✓ | ◯ | ✓ | ✓ |
|  | Figure loader | ✓ | ✓ | ◯ | ◯ | ◯ | ✓ | ✓ |
|  | Video loader | ✓ | ✓ | ✓ | ✓ | ✓ | ✓ | ✓ |
|  | Code executor | ◯ | ◯ | ◯ | ◯ | ◯ | ◯ | ✓ |
|  | data retrieval | ◯ | ✓ | ◯ | ✓ | ◯ | ◯ | ✓ |
|  | LeetCode solver | ✓ | ✓ | ✓ | ✓ | ✓ | ✓ | ✓ |

be integrated without modifying the core frameworkand reproducibility, as all traces are stored in a standardized log format compatible with downstream analysis modules.

## 3.3 FUNCTIONALITIES

The core functionalities supported by AgentRace are summarized in Table 1. Our benchmark currently supports three representative agent workflows executed across seven widely used LLM agent frameworks, utilizing a unified pool of eleven tools. While some of these capabilities are natively supported by the frameworks, approximately 50% of the functionalities are implemented by ourselves to ensure full compatibility and coverage. To maintain a fair comparison across frameworks, we adopt a standardized implementation for any functionality that is not natively provided. This ensures that differences in evaluation metrics stem from the underlying framework behavior, rather than implementation gaps. For more implementation details, please refer to Appendix A and B.9.

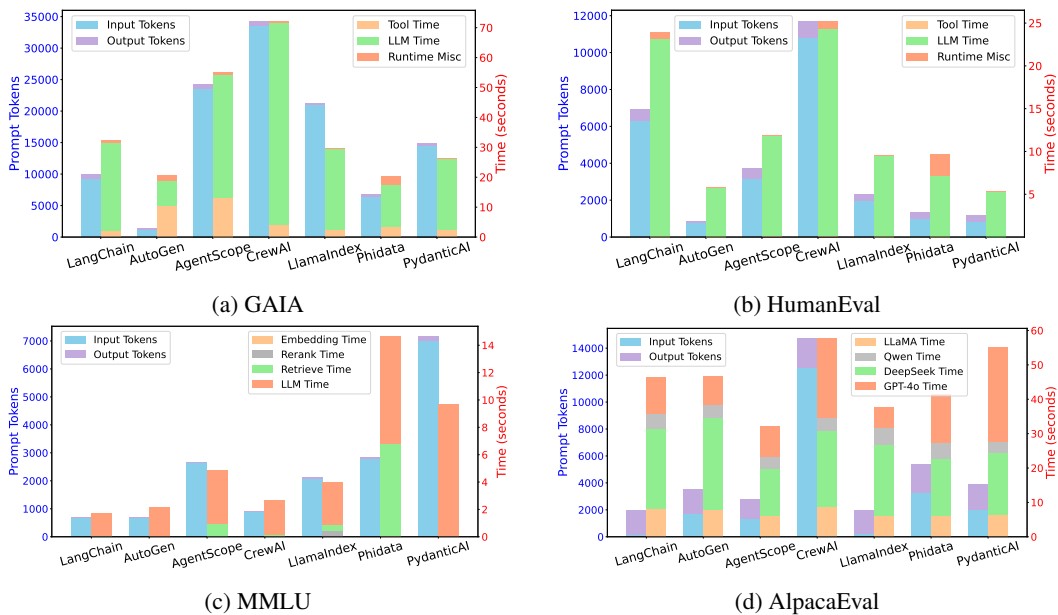

Figure 3: Token consumption and execution time per query of different frameworks.

# 4 EXPERIMENTS AND INSIGHTS

We conduct in-depth analysis for the efficiency of LLM agent frameworks. Due to the page limit, we present the representative results in the main paper. For additional experimental details and results, please refer to the Appendix.

## 4.1 EXPERIMENTAL SETUP

**Setting** We evaluate 7 LLM agent frameworks using our benchmarking platform, AgentRace, ensuring a standardized and reproducible execution environment. By default, with three repeated runs, experiments are conducted on a Linux server equipped with 12-core Intel(R) Xeon(R) Silver 4214R CPUs and a single NVIDIA RTX 3080 Ti GPU. While most of our metrics and findings are independent of hardware setup, we have also added experiments on additional servers to demonstrate the robustness of our results in Appendix B.8.

**Datasets** We use six representative datasets across different agent workflows: GAIA, HumanEval, OK-VQA and ScienceWorld are executed with the ReAct workflow, MMLU is evaluated using RAG, and AlpacaEval is tested under the MoA.

**Models** Unless otherwise specified, GPT-4o is used as the default LLM. We also conduct experiments using other models in Appendix B.7.1. For MoA, we instantiate the first-layer agents with a diverse set of open models: LLaMA-3.3-70B-Instruct-Turbo, Qwen2.5-7B-Instruct-Turbo, and DeepSeek-V3. We use TogetherAI (tog, 2024) for querying these models. GPT-4o is used as the aggregation agent to integrate their outputs. In the RAG setting, the MMLU test set is used to construct the retrieval database.

## 4.2 EXECUTION TIME AND TOKEN CONSUMPTION

*Insight 1: LLM inference usually dominates runtime across all agent frameworks, and inefficient prompt engineering, such as appending full histories and using verbose prompts, exacerbates both latency and cost.*

**Key Observations** Figure 3 presents the breakdown of agent execution time across four benchmark scenarios. The results on OK-VQA are available at Appendix B.2. Across all settings, LLM inference

consistently dominates runtime. Even in the GAIA scenario, which is explicitly designed to be tool-intensive and involves frequent calls to external APIs, LLM inference accounts for more than 85% of the total execution time in most frameworks. This highlights that LLM inference, due to its computational demands and frequent invocation, remains the primary bottleneck in agent execution, regardless of the complexity or type of task. Moreover, we observe that the cost of LLM inference is further exacerbated by large variations in token efficiency across frameworks. There is a strong positive correlation between LLM inference time and token consumption.

**Underlying Mechanism-1: Appending Full History to Prompts**    We observe that CrewAI and AgentScope elevate token usage arises from their design choice. In their implementation, the LLM stores all intermediate inputs and outputs in memory and appends this memory to each new prompt. As a result, the prompt length grows with every step of reasoning, causing a high token consumption.

**Underlying Mechanism-2: Using Verbose Prompts**    In the ReAct workflow, LlamaIndex consumes a significant amount of prompts, primarily due to the observation portion returned to the LLM after tool invocation. Additionally, for queries that fail to execute successfully, the number of reasoning and action iterations increases, leading to a corresponding growth in the observation-related prompts. For a more detailed analysis of the underlying causes, please refer to Appendix B.2.

**Potential Optimizations**    These findings underscore the importance of efficient prompt engineering and memory management in agent framework design. Strategies such as selective memory summarization, compact formatting, and prompt compression are crucial for reducing token usage.

## 4.3   TOOL CALLING

*Insight 2: Tool execution efficiency varies widely across frameworks, with search and figure-related tools introducing disproportionately high latency.*

**Key Observations**    We analyze the execution cost of various tool types across multiple LLM agent frameworks, as illustrated in Figure 4. The results reveal substantial variation in tool execution efficiency between frameworks, particularly for high-cost operations. Among all tool categories, search and figure-related tools usually incur the highest latency, often dominating total tool execution time within a workflow. For instance, the figure loader takes 2.7 seconds to execute in CrewAI, but exceeds 30 seconds in AgentScope, indicating considerable framework-dependent overhead. In contrast, lightweight tools such as

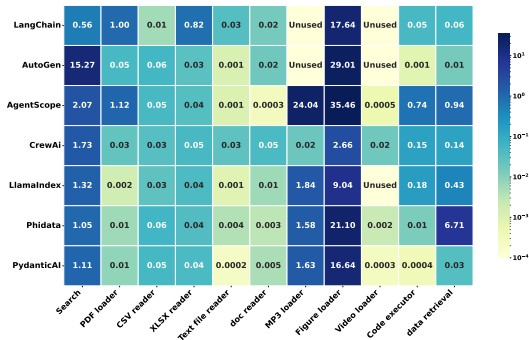

Figure 4: The execution time per call for each tool.

`Text_file_reader` and `doc_reader` typically complete in under a millisecond, demonstrating minimal variance.

Additionally, some frameworks (e.g., AgentScope) show disproportionately high total tool processing time, driven primarily by inefficient handling of image processing or multimedia tasks. This highlights the importance of optimizing high-latency tools, particularly in scenarios where tool invocation is frequent or tightly coupled with LLM inference.

**Underlying Mechanism-3: Orchestration Depth and I/O Overhead**    The pronounced disparity in execution times can be attributed to heterogeneous orchestration layers and I/O pathways across frameworks. Heavy operations, especially image-centric routines in figure-related tools, trigger large data transfers and repeated external API calls, amplifying serialization and network overhead. Frameworks with leaner orchestration logic (e.g., CrewAI) perform these steps with fewer intermediate abstractions, thereby reducing latency, whereas frameworks with deeper abstraction stacks (e.g., AgentScope) accumulate additional processing overhead. Consequently, tool latency scales not only

with the intrinsic cost of the operation but also with the efficiency of each frameworks data handling, scheduling, and resource management pipelines.

**Potential Optimizations**   While LLM inference remains the dominant bottleneck in most of our benchmarks, more complex, tool-heavy scenarios, such as document analysis or multimodal agent tasks, may shift the performance bottleneck toward tool execution. Frameworks aiming to support such use cases must pay greater attention to optimizing tool orchestration and external API integration.

## 4.4   RAG

*Insight 3: While agents usually involve external databases for information retrieval, the database performance is overlooked in several frameworks. Vector database is recommended.*

**Key Observations**   While RAG workflows are increasingly adopted to enhance factual grounding, our benchmarking reveals that database performance, particularly during embedding and retrieval, is a critical yet frequently neglected factor. Figure 3c illustrates the variation in retrieval latency across frameworks, exposing significant performance disparities.

**Underlying Mechanism-4: Embedding-Pipeline Design**   One notable example is AgentScope, which demonstrates high vectorization latency. This stems from its design: during the database setup phase, AgentScope invokes a large embedding model to compute dense vector representations. The latency of this embedding model, often implemented as a separate LLM call, substantially increases the overall vectorization time. Similarly, Phidata exhibits elevated vectorization latency due to its use of a two-step pipeline. First, its built-in `csv_tool` loads documents row-by-row; then, it applies a `SentenceTransformer` model to compute embeddings. Our benchmark confirms that Phidata's `csv_tool` itself is a relatively slow component, compounding the overall vectorization time. From our observation, vector databases such as Faiss (Douze et al., 2024) are good choices.

**Potential Optimizations**   These observations highlight the need for more attention to retrieval pipeline design, especially in frameworks that aim to support real-time or large-scale RAG deployments. Optimization opportunities include batching document embeddings, using faster embedding models, minimizing redundant file reads, and caching frequent queries.

## 4.5   COMMUNICATION SIZE

*Insight 4: Inefficient communication architecture and package design lead to high communication overhead in the multi-agent setting.*

**Key Observations**   In multi-agent frameworks, communication between agents is often overlooked as a source of inefficiency. However, our analysis reveals large discrepancies in communication size across frameworks, as shown in Table 2. These differences arise not only from framework-specific message formats but also from architectural design choices.

**Underlying Mechanism-5: Inefficient Communication Architecture**   Frameworks such as Cre-wAI, which adopt a centralized communication pattern, exhibit significantly higher communication costs. In these designs, a central agent coordinates multiple sub-agents by sequentially delegating subtasks and collecting responses. For example, in CrewAI's MoA implementation, the center agent queries three sub-agents in sequence and aggregates their outputs. Each LLM invocation by the center agent accumulates prior messages in memory, causing the prompt size and the communication payload to grow linearly with the number of sub-agents.

**Underlying Mechanism-6: Package Design**   In addition to the core message, Phidata returns a duplicated `content` field that mirrors the final message. This, combined with additional metadata fields, results in large communication sizes.

**Potential Optimizations**   These findings indicate that communication cost is not merely a function of task complexity but also of framework design. Future agent frameworks should consider decentralized communication protocols and agent sampling to reduce unnecessary transfer overhead.

Table 2: Communication size between agents (Unit: Byte). We report the content size (e.g., the transferred outputs from the last agent) and overhead size (e.g., header), separated by /.

|  |  | LangChain | AutoGen | AgentScope | CrewAI | LlamaIndex | Phidata | PydanticAI |
|---|---|---|---|---|---|---|---|---|
| **From Global Agent** | Agent1 | 165.07/0 | 209.08/44.01 | 284.078/0 | 514.962/0 | 1180.078/898 | 354.508/0 | 96.022/0 |
|  | Agent2 | 165.07/0 | 209.08/44.01 | 284.078/0 | 483.740/0 | 1171.078/889 | 341.160/0 | 95.425/0 |
|  | Agent3 | 165.07/0 | 209.08/44.01 | 284.078/0 | 619.516/0 | 1164.078/882 | 343.219/0 | 97.116/0 |
| **To Aggregation Agent** | Agent1 | 1983.02/3 | 2066.04/52.4 | 1659.318/0 | 2497.929/0 | 2022.417/33.689 | 6128.259/2639.113 | 2000.542/0 |
|  | Agent2 | 2011.83/3 | 2071.24/57.38 | 1511.311/0 | 1754.701/0 | 2054.878/39.118 | 6131.272/2629.426 | 1927.093/0 |
|  | Agent3 | 2072.98/3 | 2156.04/66.81 | 1889.247/0 | 2151.097/0 | 2116.377/48.641 | 5715.126/2465.817 | 1892.344/0 |

Table 3: Scalability Evaluation of AlpacaEval.

|  | Worker Agents | LangChain | AutoGen | AgentScope | CrewAI | LlamaIndex | Phidata | PydanticAI |
|---|---|---|---|---|---|---|---|---|
| Time (Unit: Second) | 3 | 36.50 | 36.85 | 32.12 | 64.00 | 27.32 | 50.22 | 46.45 |
|  | 6 | 37.96 | 47.34 | 67.61 | 120.54 | 36.87 | 60.42 | 42.24 |
|  | 9 | 47.11 | 50.84 | 93.36 | 212.76 | 43.85 | 63.84 | 110.78 |
|  | 12 | 59.73 | 55.60 | 122.99 | 218.34 | 53.77 | 78.80 | 111.40 |
|  | 15 | 66.08 | 46.43 | 153.78 | 245.26 | 67.23 | 83.42 | 62.13 |
| Total Token | 3 | 3516.85 | 3537.22 | 2800.75 | 14732.43 | 1933.51 | 5398.71 | 3894.06 |
|  | 6 | 7430.69 | 7211.57 | 5143.28 | 34558.34 | 3869.52 | 6940.13 | 7172.68 |
|  | 9 | 10401.23 | 10653.76 | 7547.34 | 55923.96 | 5557.50 | 7785.16 | 9256.82 |
|  | 12 | 13801.78 | 13692.51 | 10068.83 | 61244.79 | 7190.98 | 8819.67 | 9384.31 |
|  | 15 | 16894.12 | 16886.17 | 12480.56 | 80200.01 | 8873.19 | 9938.26 | 11170.89 |
| Communication Size (Unit: Byte) | 3 | 6563.04 | 6920.56 | 5912.11 | 8021.94 | 9708.91 | 19013.54 | 6108.54 |
|  | 6 | 14029.26 | 14383.36 | 10506.82 | 17863.90 | 19965.41 | 21684.95 | 12206.18 |
|  | 9 | 20468.68 | 22325.97 | 16275.87 | 24769.81 | 29936.97 | 21320.89 | 16278.34 |
|  | 12 | 27541.48 | 28782.73 | 22032.48 | 26822.83 | 39846.67 | 22383.08 | 16394.10 |
|  | 15 | 34178.20 | 35606.42 | 27526.39 | 30897.88 | 49926.39 | 23251.44 | 19198.06 |

## 4.6 SCALABILITY

*Insight 5: MoA scalability is governed by agent-invocation policy.*

**Key Observations**  We evaluate the scalability of the MoA workflow by increasing the number of worker agents from 3 to 6, 9, 12, and 15, while keeping the additional agents identical in configuration to the original ones. Table 3 reports the results on AlpacaEval. For frameworks such as AgentScope and LangChain, both execution time and token consumption grow almost linearly with the number of worker agents, reflecting sequential scheduling policies. In contrast, frameworks like PydanticAI exhibit a significantly slower growth rate, suggesting a fundamentally different invocation strategy.

**Underlying Mechanism-7: Parallel Execution**  In PydanticAI, the observed runtime is shorter than the aggregate of individual tool and LLM invocation times. This efficiency stems from its parallel execution architecture: agent calls and tool invocations are dispatched asynchronously, allowing multiple operations to overlap in time. As a result, the end-to-end latency is effectively bounded by the slowest operation rather than the sum of all operations.

**Potential Optimizations**  Our analysis indicates that task-level parallelism remains largely underexplored in current frameworks. Incorporating asynchronous scheduling and concurrent invocation can substantially improve scalability in multi-agent workflows, especially under real-world conditions where latency and throughput are critical.

## 5 CONCLUSION

We introduce AgentRace, a comprehensive benchmark platform for evaluating the efficiency of LLM agent frameworks. AgentRace covers a diverse set of datasets, agent workflows, and frameworks, enabling a fair and reproducible comparison across real-world scenarios. Through extensive and in-depth experiments, we reveal key insights and underlying mechanisms. These findings highlight critical optimization opportunities in the design and deployment of LLM-based agents. We hope AgentRace provides a guideline for future work in developing efficient, scalable, and robust agent systems, and we plan to continuously extend the benchmark as the LLM agent ecosystem evolves.

**Reproducibility Statement** We have provided our code on an anonymous website `https://agent-race.github.io/`. We have also provided the experimental details in Appendix A and reproducibility verification in Appendix B.8.

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

## A EXPERIMENTAL DETAILS

### A.1 DETAILS ABOUT THE DATASETS

We select six representative datasets that reflect varying levels of difficulty, domain coverage, and agent requirements: (1) **GAIA** (Mialon et al., 2023): A comprehensive benchmark for general-purpose AI assistants. GAIA includes real-world, multi-hop queries that require reasoning over documents, tool invocation, and web interaction. It is the most tool-intensive dataset in our suite, designed to assess the full-stack capabilities of LLM agents. Notably, GPT-4 with plugins achieves only 15% accuracy, while humans reach 92%, indicating significant headroom for improvement. (2) **HumanEval** (Chen et al., 2021): A code generation benchmark from OpenAI consisting of Python programming problems. Tasks require precise algorithmic reasoning and strict correctness, with deterministic evaluation via unit tests. This dataset helps us evaluate agents capacity for structured reasoning and program synthesis. (3) **MMLU** (Hendrycks et al., 2020): MMLU spans 57 academic subjects and provides multiple-choice questions across STEM, humanities, and social sciences. We use it to test retrieval-augmented workflows, as it simulates closed-book knowledge challenges and supports grounding in external sources. (4) **AlpacaEval** (Dubois et al., 2024): An instruction-following benchmark that evaluates natural language understanding and response quality. It consists of 805 prompts and uses GPT-4 as a reference evaluator. This dataset is well-suited for multi-agent settings where coordination, aggregation, and language alignment are essential. (5) **OK-VQA** (Marino et al., 2019): A visual question answering benchmark that requires commonsense knowledge beyond images. It contains 14,000 questions over 14,000 images and emphasizes reasoning with external world knowledge. The dataset is for evaluating the efficiency of LLM agent frameworks when handling multimodal tasks. (6) **ScienceWorld** (Wang et al., 2022): A text-based, interactive science learning environment designed to evaluate procedural reasoning. Agents are required to operate within a simulated world, carrying out multi-step experiments grounded in elementary-school science curricula. The dataset thus probes real-world execution efficiency under long-horizon reasoning and multi-step planning.

### A.2 DETAILS ABOUT THE WORKFLOWS

AgentRace includes the following workflow paradigms: (1) **ReAct (Reasoning and Acting)** (Yao et al., 2023): This paradigm interleaves natural language reasoning with tool-based actions. By prompting the LLM to first generate intermediate thoughts and then take corresponding actions, ReAct enables agents to dynamically plan and interact with their environment. (2) **RAG (Retrieval-Augmented Generation)** (Lewis et al., 2020): RAG introduces an explicit retrieval step before generation, allowing agents to ground their outputs in relevant external knowledge. In our benchmark, RAG highlights the performance of agent frameworks in integrating retrieval modules, managing memory contexts, and efficiently handling long documents. (3) **MoA (Mixture of Agents)** (Wang et al., 2025a): MoA represents a multi-agent architecture where multiple agents collaborate to solve a task. Each agent is often instantiated with a different LLM. An aggregation agent then composes their outputs to form the final answer. This setting captures the growing trend of using multiple LLMs in coordination, and allows us to benchmark frameworks on communication, modularity, and scalability.

### A.3 DETAILS ABOUT THE FRAMEWORKS

We integrate the following frameworks: (1) **LangChain** (LangChain, 2025) is a widely adopted framework that offers modular components for building LLM-based applications. It emphasizes tool chaining, prompt templating, memory integration, and external API orchestration. (2) **AutoGen** (Wu et al., 2023), developed by Microsoft, facilitates the creation of advanced LLM agents through multi-agent conversations and automated task planning. (3) **AgentScope** (Gao et al., 2024) supports rapid development of multi-agent systems through a low-code interface. It emphasizes collaboration among agent roles, enabling scalable deployment of agent collectives with minimal boilerplate. (4) **CrewAI** (Lee, 2025) is a lightweight yet expressive Python framework designed for fast iteration. It provides both high-level abstractions and low-level control. (5) **LlamaIndex** (LlamaIndex, 2025) focuses on context-augmented LLM applications by connecting structured and unstructured data sources to LLMs. (6) **Phidata** (agno-agi, 2025) is a framework for building multi-modal AI agents and workflows with memory, knowledge, tools, and reasoning, enabling collaborative problem-solving

through teams of agents. (7) **PydanticAI** (PydanticAI, 2025) is an agent framework that is designed for easy development of production-grade applications.

### A.4 VERSIONS OF EVALUATED FRAMEWORKS

All LLM agent frameworks employed in this study are contemporaneous, with the specific version numbers reported in Table 4.

Table 4: Versions of the LLM Agent frameworks employed in this paper.

| Framework | Version | Framework | version |
|---|---|---|---|
| LangChain | 0.3.22 | LlamaIndex | 0.12.30 |
| AutoGen | 0.8.2 | Phidata | 2.7.10 |
| AgentScope | 0.1.3 | PydanticAI | 0.1.0 |
| CrewAI | 0.114.0 | | |

### A.5 HYPERPARAMETERS

In all experiments, the temperature was set to 0, the top k to 1 (if available), and all other parameters were set to their default values unless otherwise specified.

Except for the cases explicitly noted in Appendix C, all workflows employ the default prompts provided by their respective frameworks, and the datasets are used without any modification to the original queries. For the ScienceWorld dataset specifically, we additionally cap the maximum number of ReAct iterations per query at 50 to prevent excessively long interaction loops.

## B ADDITIONAL RESULTS

### B.1 ACCURACY

Table 5: Accuracy of each framework on each dataset.

| Dataset | LangChain | AutoGen | AgentScope | CrewAI | LlamaIndex | Phidata | PydanticAI |
|---|---|---|---|---|---|---|---|
| **GAIA** | 0.152±0.012 | 0.107±0.003 | 0.212±0.012 | 0.222±0.009 | 0.198±0.015 | 0.191±0.026 | 0.157±0.012 |
| **HumanEval** | 0.573 | 0.884 | 0.884 | 0.872 | 0.872 | 0.902 | 0.921 |
| **MMLU** | 0.820 | 0.817 | 0.827 | 0.813 | 0.745 | 0.792 | 0.788 |
| **OK-VQA** | - | 0.366 | 0.568 | 0.428 | 0.381 | 0.337 | 0.310 |
| ScienceWorld | 0.245±0.036 | - | 0.270±0.045 | 0.113±0.008 | 0.321±0.027 | 0.186±0.033 | 0.155±0.020 |

Table 5 presents the accuracy of each framework, while for the ScienceWorld dataset it reports the score rate. In general, the accuracy differences among frameworks are relatively small when using the same underlying LLM. However, there are still some notable exceptions.

> *Insight 6: The complete absence of output constraints in LLMs may lead to tool invocation failures, whereas excessively strict output validation can incur substantial token overhead and decrease the response success rate.*

**Key Observations**   In our evaluation, we find that when the model skips tool invocation and instead provides a direct answer (this happens especially with some of the simpler queries in the HumanEval dataset), the framework retries the prompt, often multiple times. Each retry includes previous failed attempts in the context, leading to a rapid increase in prompt length and token consumption as well as a lower likelihood of producing a clean, valid output on later attempts.

**Underlying Mechanism-8: Structured Output Misalignment**   Some frameworks, such as LlamaIndex, require tool inputs to conform to a strict dictionary format. However, GPT-4o does not consistently produce structured outputs that align with these expectations, leading to frequent tool invocation failures. This issue can be partially mitigated if the framework explicitly enforces the

Table 6: Accuracy comparison under different memory window sizes of CrewAI on the GAIA dataset.

| Memory Window Size | 1 | 25 | 35 | Max |
|---|---|---|---|---|
| Accuracy | 0.236 | 0.248 | 0.242 | 0.218 |
| Average Token Comsumption per Query | 79767.8 | 85032.61 | 87013.7 | 95426.57 |

format requirement during the registration phase or input schema definition. In contrast, other frameworks such as LangChain adopt stricter enforcement mechanisms. ReAct-style agents in these systems perform rigid output validation and initiate automatic retries when the model's response deviates from the expected invocation structure. While such mechanisms increase robustness against malformed outputs, they may backfire in certain scenarios.

*Insight 7: Larger memory windows do not necessarily improve accuracy and can substantially degrade efficiency.*

**Key Observations**    To investigate whether the built-in historical memory of the CrewAI framework affects accuracy, we compared four settings on the GAIA dataset: (i) using a memory window size of 1, (ii) using a memory window size of 25, (iii) using a memory window size of 35, and (iv) using the maximum memory window size. Here, the memory window size indicates the interval of queries after which the Agent is re-initialized (e.g., every 1, 25, or 35 queries), while the maximum setting corresponds to initialization only at the very beginning of the task. Results are shown in Table 6.

**Underlying Mechanism-9: Accuracy-Efficiency Tradeoff**    We observe that as the memory window size increases, token consumption rises steadily, while accuracy first improves and then declines. This indicates that incorporating an appropriate amount of memory can enhance task performance; however, excessive memory not only leads to escalating token costs and reduced efficiency, but also fails to yield higher accuracy. In practice, the memory window size should therefore be tuned to achieve a reasonable balance between accuracy and efficiency.

An additional point to clarify is that the GAIA dataset exhibits relatively low accuracy. This is primarily because GAIA tasks often require complex task planning and the use of multiple tools, posing significant challenges for all evaluated frameworks. It is important to note that the primary focus of this study is not on accuracy, but rather on comparing the performance overhead (e.g., time, token usage) across different frameworks. Therefore, we ensure that the accuracy across frameworks remains broadly comparable, without conducting detailed task-level progress analysis as seen in some related work. By carefully controlling experimental parameters, the fairness of our comparisons remains valid, even in the presence of lower absolute accuracy.

## B.2    DETAILED EVALUATION RESULTS

Figure 5 presents the token and time consumption of OK-VQA.

Table 7, 8, 9, 10, 11, and 12 presents the detailed results obtained in this experiment. Unless stated otherwise, the times reported in the table are in seconds per query. The missing data corresponds to instances where the LLM failed to invoke the required tool correctly during the experiment (e.g., by not returning outputs in the expected format or by not selecting the appropriate tool for invocation). In the ScienceWorld dataset, environment interactions require strict synchronization, whereas the AutoGen framework operates asynchronously, making evaluation infeasible. The following are some noteworthy observations.

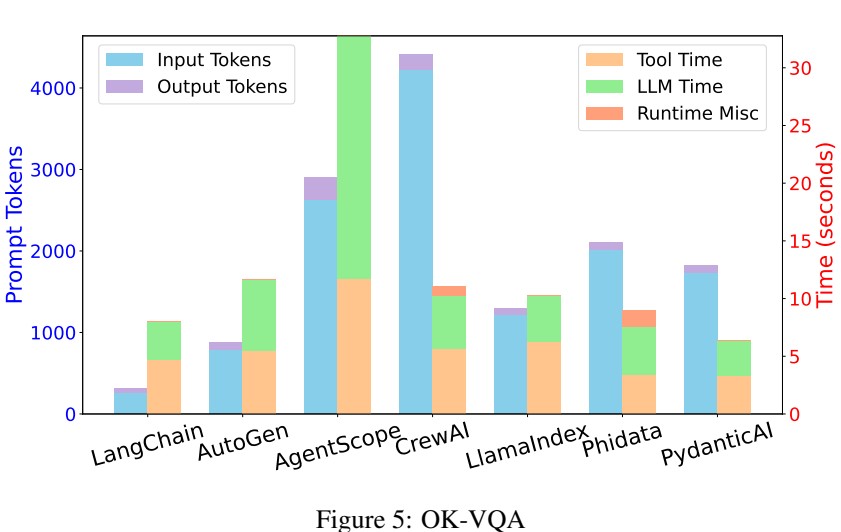

Figure 5: OK-VQA

Table 7: GAIA Detailed Results

|  |  | LangChain | AutoGen | AgentScope | CrewAI | LlamaIndex | Phidata | PydanticAI |
|---|---|---|---|---|---|---|---|---|
| **Token** | Prompt | 9358.35 | 1159.48 | 23520.479 | 33621.857 | 20935.364 | 6386.667 | 14459.17 |
|  | Output | 637.92 | 180.66 | 785.891 | 664.511 | 304.976 | 323.558 | 320.588 |
|  | Total | 9996.27 | 1340.15 | 24306.37 | 34286.369 | 21240.339 | 6710.224 | 14779.758 |
| **Time** | llm | 29.491 | 8.464 | 41.17 | 67.68 | 27.244 | 14.375 | 23.779 |
|  | Search | 1.58856 | 9.4219 | 7.291 | 4.031 | 1.4399 | 1.83012 | 1.2275 |
|  | PDF loader | 0.02423455 | 0.0009297 | 0.217 | 0.00965 | 0.0001352 | 0.001147 | 0.001395 |
|  | CSV reader | 0.00003333 | 0.000336 | 0.000297 | 0.000196 | 0.00016616 | 0.0007207 | 0.0003148 |
|  | XLSX reader | 0.06422606 | 0.002387 | 0.00405 | 0.00422 | 0.004254 | 0.003858 | 0.003795 |
|  | Text file reader | 0.0004194 | 0.00002909 | 0.0000193 | 0.00123 | 0.000034839 | 0.0002107 | 8.6865E-06 |
|  | Doc reader | 0.00009758 | 0.0002212 | 0.00000883 | 0.000278 | 0.0001135 | 0.000073355 | 0.000056241 |
|  | MP3 loader | - | - | 0.729 | 0.000346 | 0.03341 | 0.03821 | 0.02965 |
|  | Figure loader | 0.5345976 | 1.05489 | 4.083 | 0.03164 | 0.8767 | 1.4065 | 1.2104 |
|  | Video loader | - | - | 0.0000271 | 0.000999 | - | 1.38445E-05 | 3.1952E-06 |
|  | Code executor | 0.0152988 | 0.00005333 | 0.752 | 0.09565 | 0.05782 | 0.003035 | 0.0001414 |
|  | Total tool time | 2.22746732 | 10.4807 | 13.076 | 4.18 | 2.4126 | 3.2839 | 2.4732 |
|  | Total time | 32.492 | 20.76 | 55.092 | 72.195 | 29.795 | 20.396 | 26.238 |

Table 8: HumanEval Detailed Results

| Framework | Token | | | Time | | |
|---|---|---|---|---|---|---|
|  | Prompt | Output | Total | LLM | Code executor | Total |
| LangChain | 6326.36 | 617.13 | 6943.49 | 23.221 | 0.0034 | 23.968 |
| AutoGen | 767.45 | 106.34 | 873.79 | 5.822 | 0.0002 | 5.846 |
| AgentScope | 3180.689 | 561.518 | 3742.207 | 11.738 | 0.131 | 11.906 |
| CrewAI | 10817.65 | 892.798 | 11710.45 | 24.22 | 0.0258 | 25.24 |
| LlamaIndex | 1985.6 | 342.793 | 2328.152 | 9.52 | 0.003069 | 9.611 |
| Phidata | 967.329 | 354.427 | 1321.756 | 7.181 | - | 9.692 |
| PydanticAI | 812.951 | 352.543 | 1165.494 | 5.258 | 0.000007158 | 5.276 |

Table 9: MMLU Detailed Results

| Framework | Token | | | Time | | | |
|---|---|---|---|---|---|---|---|
|  | Prompt | Output | Total | LLM | Embedding | Retrieve | Total |
| LangChain | 701.514 | 4.035 | 705.55 | 1.677 | 11.833 | 0.055 | 1.79 |
| AutoGen | 679.788 | 3.956 | 683.744 | 2.171 | 6.526 | 0.015 | 2.182 |
| AgentScope | 2664.315 | 2.878 | 2667.193 | 3.893 | 92.472 | 0.935 | 4.931 |
| CrewAI | 884.536 | 13.189 | 897.724 | 2.51 | 7.718 | 0.14 | 5 |
| LlamaIndex | 2079.702 | 50.339 | 2130.042 | 3.125 | 4.931 | 0.4303 | 3.575 |
| Phidata | 2797.441 | 37.347 | 2834.788 | 7.849 | 341.611 | 6.708 | 17.014 |
| PydanticAI | 6996.242 | 170.135 | 7166.378 | 9.685 | 5.977 | 0.03454 | 9.824 |

Table 10: AlpacaEval Detailed Results

| | | | LangChain | AutoGen | AgentScope | CrewAI | LlamaIndex | Phidata | PydanticAI |
|---|---|---|---|---|---|---|---|---|---|
| **Token** | llama | prompt | 70.49 | 70.49 | 85.451 | 298.25 | 70.49 | 118.846 | 61.347 |
| | | output | 428.55 | 431.96 | 382.45 | 518.95 | 430.216 | 438.078 | 429.543 |
| | | total | 499.04 | 502.45 | 467.901 | 817.201 | 500.707 | 556.924 | 490.889 |
| | qwen | prompt | 64.84 | 64.85 | 61.815 | 258.083 | 64.81 | 93.899 | 41.217 |
| | | output | 446.05 | 447.45 | 311.109 | 398.618 | 441.738 | 463.795 | 433.739 |
| | | total | 510.91 | 512.31 | 372.924 | 656.702 | 506.548 | 557.694 | 474.957 |
| | deepseek | prompt | 38.5 | 38.5 | 52.478 | 313.01 | 38.485 | 83.391 | 31.802 |
| | | output | 501.11 | 503.37 | 416.639 | 571.79 | 495.306 | 440.691 | 434.81 |
| | | total | 539.61 | 541.87 | 469.117 | 884.808 | 533.791 | 524.082 | 485.612 |
| | gpt | prompt | 1522.48 | 1529.96 | 1138.243 | 11694.576 | 42.083 | 3003.319 | 1845.724 |
| | | output | 444.81 | 450.63 | 352.564 | 679.15 | 350.386 | 756.689 | 596.876 |
| | | total | 1967.29 | 1980.59 | 1490.807 | 12373.72 | 392.47 | 3760.009 | 2442.6 |
| **Time** | llama | | 8.275 | 7.812 | 6.063 | 8.835 | 6.069 | 6.152 | 6.503 |
| | qwen | | 4.48 | 3.977 | 3.415 | 3.837 | 4.787 | 4.707 | 3.441 |
| | deepseek | | 23.084 | 26.745 | 13.726 | 21.946 | 20.829 | 16.456 | 17.79 |
| | aggregator | | 10.699 | 8.274 | 8.89 | 23.114 | 5.849 | 14.208 | 27.486 |
| | total | | 36.502 | 36.854 | 32.119 | 64 | 27.318 | 50.217 | 46.45 |
| **Communication** | prompt to agent1 | | 165.07/0 | 209.08/44.01 | 284.078/118 | 514.962/0 | 1180.078/898 | 354.508/0 | 96.022/0 |
| | prompt to agent2 | | 165.07/0 | 209.08/44.01 | 284.078/118 | 483.740/0 | 1171.078/889 | 341.160/0 | 95.425/0 |
| | prompt to agent3 | | 165.07/0 | 209.08/44.01 | 284.078/118 | 619.516/0 | 1164.078/882 | 343.219/0 | 97.116/0 |
| | agent1 to aggregator | | 1983.02/3 | 2066.04/52.24 | 1659.318/124 | 2497.929/0 | 2022.417/33.689 | 6128.259/2639.113 | 2000.542/0 |
| | agent2 to aggregator | | 2011.83/3 | 2071.24/57.38 | 1511.311/122 | 1754.701/0 | 2054.878/39.118 | 6131.272/2629.426 | 1927.093/0 |
| | agent3 to aggregator | | 2072.98/3 | 2156.04/66.81 | 1889.247/126 | 2151.097/0 | 2116.377/48.641 | 5715.126/2465.817 | 1892.344/0 |

Table 11: OK-VQA Detailed Results

| Framework | token | | | time | | |
|---|---|---|---|---|---|---|
| | **prompt** | **output** | **total** | **llm** | **Figure loader** | **total** |
| LangChain | $261.033 \pm 0.462$ | $52.567 \pm 0.473$ | $313.633 \pm 0.058$ | $2.948 \pm 0.354$ | $4.716 \pm 0.115$ | $7.664 \pm 0.426$ |
| AutoGen | $791.133 \pm 0.635$ | $89.467 \pm 1.882$ | $880.600 \pm 1.609$ | $6.197 \pm 0.060$ | $5.171 \pm 0.233$ | $11.368 \pm 0.240$ |
| AgentScope | $2621.367 \pm 30.029$ | $283.433 \pm 3.355$ | $2902.567 \pm 30.346$ | $15.537 \pm 5.753$ | $9.043 \pm 3.031$ | $24.580 \pm 8.779$ |
| CrewAI | $4510.933 \pm 254.635$ | $269.600 \pm 120.951$ | $4780.600 \pm 318.263$ | $4.657 \pm 0.121$ | $5.578 \pm 1.236$ | $10.990 \pm 1.536$ |
| LlamaIndex | $1219.300 \pm 1.682$ | $83.833 \pm 0.208$ | $1303.167 \pm 1.909$ | $5.548 \pm 1.486$ | $5.476 \pm 0.627$ | $11.024 \pm 0.998$ |
| Phidata | $2019.167 \pm 2.401$ | $88.500 \pm 0.600$ | $2107.500 \pm 2.193$ | $4.132 \pm 0.054$ | $3.930 \pm 0.403$ | $9.039 \pm 0.027$ |
| PydanticAI | $1728.367 \pm 1.674$ | $92.100 \pm 0.608$ | $1820.433 \pm 2.223$ | $3.034 \pm 0.012$ | $3.352 \pm 0.057$ | $6.390 \pm 0.025$ |

*Insight 8: Token consumption may vary across frameworks even when executing the same workflow, owing to differences in implementation strategies.*

**Key Observations** In the results of ReAct workflow, it can be observed that even when using the same ReAct workflow, AgentScope exhibits a significant discrepancy in token usage between the GAIA and HumanEval datasets, with exceptionally high token consumption on GAIA. This is primarily because AgentScope includes the entire memory of the agent in the prompt during every LLM invocation. As the number of reasoning steps increases, the prompt length grows rapidly. While this issue is less apparent in the relatively simple HumanEval dataset, it becomes prominent in the more complex GAIA tasks.

The high token usage observed in CrewAI's ReAct workflow can be attributed to the same reason. In fact, this issue is even more pronounced in CrewAI than in AgentScope, with significantly elevated token consumption observed across both the GAIA and HumanEval datasets.

**Underlying Mechanism-10: Overly Detailed Observations** In contrast, the majority of token consumption in LlamaIndex and Pydantic arises from the observation segments returned to the LLM after tool invocations. In the GAIA dataset, where tasks are complex and involve frequent tool usage, this results in substantial prompt token overhead.

There are also some issues observed in the MoA workflow. For example, PydanticAI does not require the invocation of all sub-agents during MoA execution, thereby reducing token consumption and runtime overhead. For further details, please refer to the Insight 8 in Appendix B.4.1.

Another example is that in the CrewAI framework, MoA is centrally managed by a global agent, which also plays the role of aggregation agent. The global agent receives the task and sequentially assigns it to sub-agents (e.g., agent1, agent2, agent3). Each sub-agent completes its part and returns the result to the global agent, which then decides the next step. After all agents have responded, the

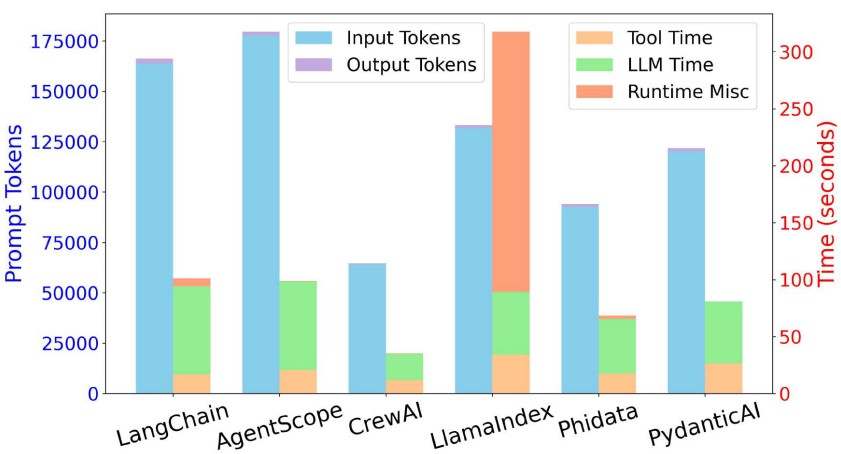

Figure 6: ScienceWorld

global agent summarizes the results and outputs the final answer. In this setup, the global agent calls the LLM multiple timesonce after each sub-agents response. Because LLMs retain the full context of previous inputs and outputs in a single session, each new call includes all prior interactions. This leads to token accumulation, especially by the third or fourth step, where the prompt becomes much longer. As a result, total token usage becomes higher than in frameworks with different coordination or memory strategies. This phenomenon will become more apparent in Scalability part as the number of sub agents increases. For further details, please refer to the Insight 4 in Section 4.5.

## B.3 RESULTS ON SCIENCEWORLD

The results on the ScienceWorld dataset are presented in Figure 6, with detailed results and score-range statistics shown in Table 12, 13, 14, 15, 16, 17 and 18. Under the multi-turn reasoning setting, we identify several issues that remain largely hidden when the number of turns is small.

*Insight 9: The scalability of misc overhead in multi-turn reasoning is architecture-dependent, emerging when agent frameworks link context growth to repeated aggregation and parsing operations.*

**Key Observations** As illustrated in the figure, a substantial portion of LlamaIndex's total execution time is attributed to runtime misc. This overhead originates directly from the particular internal implementation of LlamaIndexs ReAct workflow, which integrates the LLM output with tool results at the end of every reasoning turn. While this implementation detail imposes negligible cost when the number of turns is limited, the burden becomes increasingly pronounced as the contextual length expands across turns.

**Underlying Mechanism-11: Context GrowthInduced Increases in Aggregation and Parsing Time** In the LlamaIndex framework, we select a subset of queries and visualize how their tool-observation aggregation time and output-parsing time evolve as the number of ReAct iterations increases, as illustrated in Figure 7. Evidently, the context expansion induced by additional ReAct rounds introduces substantial miscellaneous latency into the overall system execution.

In this dataset, CrewAI exhibits relatively low token consumption because it frequently terminates early after incorrectly assuming that the task has been completed. Aside from the anomalies discussed above, the relative efficiency of the remaining frameworks is broadly consistent with the results observed on the other datasets.

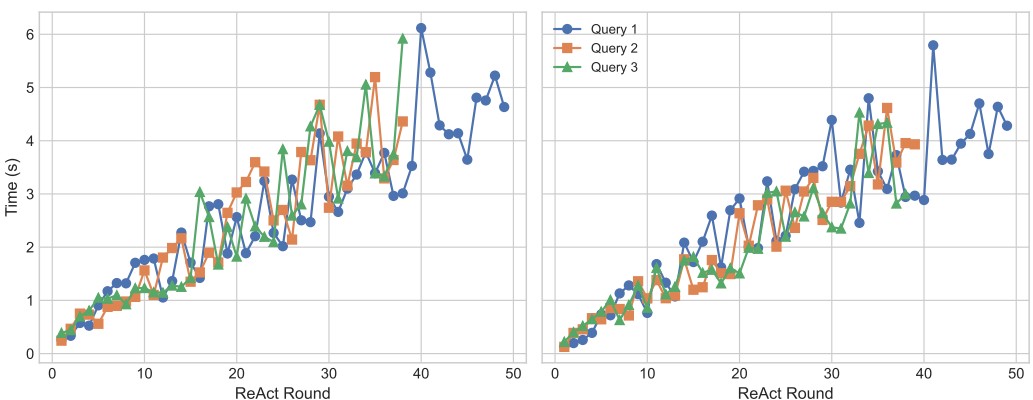

Figure 7: Visualization of the tool-observation aggregation time per ReAct round in the LlamaIndex framework (left), and the output-parsing time per ReAct round in the LlamaIndex framework (right).

Table 12: ScienceWorld Detailed Results

| | | LangChain | AgentScope | CrewAI | LlamaIndex | Phidata | PydanticAI |
|---|---|---|---|---|---|---|---|
| **Token** | Prompt | 163844.6 ± 10233.6 | 177881.9 ± 16973.9 | 64194.4 ± 28803.6 | 131962.6 ± 3138.8 | 92710.4 ± 5904.6 | 120264.6 ± 8372.3 |
| | Output | 2409.3 ± 426.7 | 1698.6 ± 126.7 | 473.6 ± 252.9 | 1373.1 ± 39.1 | 1362.9 ± 127.5 | 1582.5 ± 28.0 |
| | Total | 166253.9 ± 10640.5 | 179580.5 ± 17099.1 | 64668.0 ± 29056.1 | 133335.8 ± 3177.9 | 94073.3 ± 6028.8 | 121847.1 ± 8394.7 |
| **Time** | llm | 77.385 ± 14.858 | 77.637 ± 8.703 | 23.304 ± 8.138 | 55.284 ± 6.154 | 47.635 ± 7.586 | 54.456 ± 2.584 |
| | observe | 0.002 ± 0.000 | 0.004 ± 0.001 | 0.007 ± 0.001 | 0.002 ± 0.001 | 0.000 ± 0.001 | 0.003 ± 0.001 |
| | execute action | 2.014 ± 0.099 | 2.144 ± 0.096 | 0.241 ± 0.015 | 2.830 ± 0.099 | 1.946 ± 0.235 | 2.488 ± 0.128 |
| | embedding | 14.777 ± 1.737 | 18.548 ± 1.142 | 11.396 ± 0.837 | 31.138 ± 2.500 | 15.819 ± 2.751 | 23.878 ± 4.387 |
| | total | 101.095 ± 16.223 | 98.920 ± 8.026 | 35.29 ± 8.654 | 317.573 ± 7.120 | 68.424 ± 8.301 | 80.220 ± 1.345 |

Table 13: ScienceWorld Score-Stratified Results Using LangChain

| Score Range | | [0,20) | [20,40) | [40,60) | [60,80) | [80,100) | 100 |
|---|---|---|---|---|---|---|---|
| Sample Size | | 15.000 ± 1.732 | 9.333 ± 1.528 | 3.000 ± 0.000 | 1.000 ± 1.000 | 0.000 ± 0.000 | 1.667 ± 1.155 |
| score | | 7.469 ± 1.789 | 26.436 ± 2.331 | 47.778 ± 1.925 | 67.000 ± 2.828 | – | 100.000 ± 0.000 |
| **time** | llm | 82.788 ± 12.766 | 85.643 ± 22.848 | 43.126 ± 21.067 | 71.022 ± 6.053 | – | 30.905 ± 5.461 |
| | observe | 0.002 ± 0.000 | 0.002 ± 0.000 | 0.004 ± 0.002 | 0.002 ± 0.002 | – | 0.003 ± 0.003 |
| | execute action | 2.344 ± 0.117 | 2.066 ± 0.099 | 0.875 ± 0.786 | 1.587 ± 0.723 | – | 0.895 ± 0.261 |
| | embedding | 13.526 ± 3.245 | 20.263 ± 1.323 | 5.539 ± 2.520 | 15.169 ± 9.559 | – | 8.497 ± 1.003 |
| | total | 106.515 ± 14.780 | 115.746 ± 23.735 | 52.230 ± 26.594 | 92.907 ± 18.716 | – | 41.715 ± 7.163 |
| **token** | prompt | 178752.5 ± 11023.4 | 184478.1 ± 33892.0 | 80192.3 ± 49486.8 | 142843.8 ± 44820.3 | – | 56232.1 ± 12681.6 |
| | completion | 2540.3 ± 309.5 | 2853.9 ± 930.5 | 1064.6 ± 601.1 | 2428.5 ± 393.9 | – | 694.6 ± 282.4 |
| | total | 181292.8 ± 11111.9 | 187332.0 ± 34786.7 | 81256.9 ± 50086.5 | 145272.3 ± 45214.2 | – | 56926.7 ± 12941.4 |

Table 14: ScienceWorld Score-Stratified Results Using AgentScope

| Score Range | | [0,20) | [20,40) | [40,60) | [60,80) | [80,100) | 100 |
|---|---|---|---|---|---|---|---|
| Sample Size | | 17.000 ± 1.000 | 6.000 ± 1.000 | 4.000 ± 0.000 | 0.333 ± 0.577 | 0.000 ± 0.000 | 2.667 ± 2.082 |
| score | | 8.340 ± 1.276 | 30.505 ± 0.692 | 49.500 ± 0.000 | 64.000 | – | 100.000 ± 0.000 |
| **time** | llm | 91.354 ± 13.255 | 62.076 ± 18.889 | 61.714 ± 21.463 | 31.744 | – | 48.020 ± 15.137 |
| | observe | 0.005 ± 0.002 | 0.003 ± 0.001 | 0.003 ± 0.000 | 0.004 | – | 0.004 ± 0.002 |
| | execute action | 2.573 ± 0.318 | 1.946 ± 0.855 | 1.092 ± 0.429 | 0.625 | – | 1.194 ± 0.503 |
| | embedding | 19.964 ± 4.394 | 22.393 ± 10.963 | 10.095 ± 11.085 | 6.228 | – | 8.744 ± 6.065 |
| | total | 114.615 ± 15.498 | 86.850 ± 28.690 | 73.367 ± 33.019 | 38.778 | – | 58.227 ± 21.504 |
| **token** | prompt | 216993.3 ± 41394.1 | 129536.5 ± 45095.0 | 145026.8 ± 44153.2 | 60403.0 | – | 81335.4 ± 26057.4 |
| | completion | 1987.9 ± 292.2 | 1320.9 ± 334.2 | 1488.9 ± 417.8 | 704.0 | – | 1035.2 ± 263.1 |
| | total | 218981.2 ± 41686.3 | 130857.4 ± 45427.3 | 146515.7 ± 44570.9 | 61107.0 | – | 82370.6 ± 26292.4 |

Table 15: ScienceWorld Score-Stratified Results Using CrewAI

| Score Range | | [0,20) | [20,40) | [40,60) | [60,80) | [80,100) | 100 |
|---|---|---|---|---|---|---|---|
| Sample Size | | $26.000 \pm 1.000$ | $2.667 \pm 0.577$ | $1.000 \pm 0.000$ | $0.333 \pm 0.577$ | $0.000 \pm 0.000$ | $0.000 \pm 0.000$ |
| score | | $7.231 \pm 0.513$ | $26.444 \pm 1.347$ | $58.000 \pm 0.000$ | 63.000 | – | – |
| time | llm | $23.725 \pm 9.332$ | $18.106 \pm 8.503$ | $18.365 \pm 6.418$ | 20.977 | – | – |
| | observe | $0.006 \pm 0.001$ | $0.007 \pm 0.007$ | $0.006 \pm 0.011$ | 0.019 | – | – |
| | execute action | $0.242 \pm 0.019$ | $0.214 \pm 0.036$ | $0.239 \pm 0.062$ | 0.354 | – | – |
| | embedding | $10.989 \pm 1.214$ | $13.379 \pm 2.639$ | $8.738 \pm 5.829$ | 31.174 | – | – |
| | total | $35.328 \pm 10.468$ | $31.778 \pm 10.090$ | $27.667 \pm 12.503$ | 53.000 | – | – |
| token | prompt | $66868.0 \pm 32135.4$ | $42852.7 \pm 19720.6$ | $36523.3 \pm 24687.5$ | 39185.0 | – | – |
| | completion | $496.1 \pm 281.5$ | $305.5 \pm 196.8$ | $199.3 \pm 128.7$ | 247.0 | – | – |
| | total | $67364.1 \pm 32416.5$ | $43158.2 \pm 19915.4$ | $36722.7 \pm 24816.2$ | 39432.0 | – | – |

Table 16: ScienceWorld Score-Stratified Results Using LlamaIndex

| Score Range | | [0,20) | [20,40) | [40,60) | [60,80) | [80,100) | 100 |
|---|---|---|---|---|---|---|---|
| Sample Size | | $17.667 \pm 0.577$ | $4.000 \pm 1.732$ | $2.000 \pm 1.000$ | $0.667 \pm 0.577$ | $0.333 \pm 0.577$ | $5.333 \pm 0.577$ |
| score | | $7.285 \pm 1.397$ | $32.233 \pm 2.892$ | $48.111 \pm 2.715$ | $79.000 \pm 0.000$ | - | $100.000 \pm 0.000$ |
| time | llm | $62.620 \pm 8.169$ | $59.292 \pm 3.084$ | $30.854 \pm 16.198$ | $29.842 \pm 3.449$ | 117.295 | $34.229 \pm 6.060$ |
| | observe | $0.001 \pm 0.001$ | $0.000 \pm 0.000$ | $0.000 \pm 0.000$ | $0.003 \pm 0.004$ | 0.000 | $0.006 \pm 0.007$ |
| | execute action | $3.257 \pm 0.231$ | $3.118 \pm 0.274$ | $1.421 \pm 0.766$ | $1.342 \pm 0.067$ | 6.892 | $1.591 \pm 0.202$ |
| | embedding | $34.829 \pm 2.940$ | $35.791 \pm 6.322$ | $19.312 \pm 13.152$ | $14.265 \pm 1.028$ | 154.870 | $12.615 \pm 2.355$ |
| | total | $367.898 \pm 33.450$ | $344.795 \pm 93.646$ | $151.927 \pm 101.836$ | $104.051 \pm 5.878$ | 974.875 | $151.561 \pm 53.023$ |
| token | prompt | $149947.1 \pm 10507.0$ | $145832.7 \pm 30441.3$ | $68755.1 \pm 34604.7$ | $62796.5 \pm 6004.0$ | 302576.0 | $75850.4 \pm 19426.6$ |
| | completion | $1523.6 \pm 92.3$ | $1526.0 \pm 348.1$ | $795.5 \pm 345.5$ | $755.5 \pm 50.2$ | 2441.0 | $904.3 \pm 143.7$ |
| | total | $151470.7 \pm 10597.5$ | $147358.7 \pm 30786.6$ | $69550.6 \pm 34950.0$ | $63552.0 \pm 6054.2$ | 305017.0 | $76754.7 \pm 19570.3$ |

Table 17: ScienceWorld Score-Stratified Results Using Phidata

| Score Range | | [0,20) | [20,40) | [40,60) | [60,80) | [80,100) | 100 |
|---|---|---|---|---|---|---|---|
| Sample Size | | $20.000 \pm 1.000$ | $5.000 \pm 1.000$ | $2.667 \pm 1.528$ | $0.667 \pm 0.577$ | $0.000 \pm 0.000$ | $1.667 \pm 1.528$ |
| score | | $4.227 \pm 0.690$ | $26.428 \pm 0.700$ | $52.278 \pm 4.956$ | $61.500 \pm 2.121$ | – | $100.000 \pm 0.000$ |
| time | llm | $51.317 \pm 11.779$ | $38.832 \pm 12.498$ | $39.743 \pm 13.801$ | $32.669 \pm 31.491$ | – | $33.321 \pm 16.319$ |
| | observe | $0.000 \pm 0.000$ | $0.002 \pm 0.003$ | $0.001 \pm 0.001$ | $0.000 \pm 0.000$ | – | $0.000 \pm 0.000$ |
| | execute action | $1.993 \pm 0.364$ | $2.132 \pm 0.482$ | $1.275 \pm 0.714$ | $1.480 \pm 1.392$ | – | $1.462 \pm 0.665$ |
| | embedding | $14.523 \pm 1.784$ | $21.583 \pm 6.349$ | $15.580 \pm 18.653$ | $10.354 \pm 9.985$ | – | $10.503 \pm 6.413$ |
| | total | $70.976 \pm 12.416$ | $65.406 \pm 19.693$ | $59.283 \pm 32.139$ | $46.867 \pm 44.381$ | – | $47.614 \pm 23.774$ |
| token | prompt | $100955.9 \pm 10072.5$ | $70879.6 \pm 26332.4$ | $82879.0 \pm 41041.2$ | $57747.5 \pm 60268.8$ | – | $50285.8 \pm 31204.4$ |
| | completion | $1464.9 \pm 264.1$ | $1056.3 \pm 288.9$ | $1125.8 \pm 636.3$ | $1003.5 \pm 1034.5$ | – | $1029.3 \pm 584.5$ |
| | total | $102420.8 \pm 10336.3$ | $71935.9 \pm 26618.2$ | $84004.8 \pm 41664.1$ | $58751.0 \pm 61303.3$ | – | $51315.2 \pm 31788.9$ |

Table 18: ScienceWorld Score-Stratified Results Using PydanticAI

| Score Range | | [0,20) | [20,40) | [40,60) | [60,80) | [80,100) | 100 |
|---|---|---|---|---|---|---|---|
| Sample Size | | $22.333 \pm 1.528$ | $3.667 \pm 1.155$ | $2.667 \pm 0.577$ | $0.000 \pm 0.000$ | $0.000 \pm 0.000$ | $1.333 \pm 0.577$ |
| score | | $4.148 \pm 0.610$ | $29.044 \pm 2.746$ | $49.778 \pm 3.006$ | – | – | $100.000 \pm 0.000$ |
| time | llm | $55.837 \pm 0.833$ | $68.125 \pm 13.935$ | $36.102 \pm 24.343$ | – | – | $24.629 \pm 1.191$ |
| | observe | $0.003 \pm 0.001$ | $0.003 \pm 0.001$ | $0.005 \pm 0.003$ | – | – | $0.003 \pm 0.001$ |
| | execute action | $2.603 \pm 0.201$ | $3.541 \pm 1.234$ | $0.740 \pm 0.430$ | – | – | $1.279 \pm 0.570$ |
| | embedding | $24.097 \pm 4.821$ | $42.766 \pm 8.607$ | $5.276 \pm 1.901$ | – | – | $9.065 \pm 8.437$ |
| | total | $82.234 \pm 4.142$ | $111.584 \pm 13.598$ | $42.222 \pm 25.986$ | – | – | $34.015 \pm 8.169$ |
| token | prompt | $120573.9 \pm 15709.7$ | $169962.6 \pm 37787.0$ | $79238.9 \pm 71862.6$ | – | – | $42432.5 \pm 3102.6$ |
| | completion | $1591.5 \pm 178.2$ | $1966.1 \pm 273.8$ | $1212.6 \pm 1170.5$ | – | – | $756.5 \pm 239.3$ |
| | total | $122165.5 \pm 15848.1$ | $171928.7 \pm 38022.4$ | $80451.4 \pm 73030.9$ | – | – | $43189.0 \pm 2968.9$ |

## B.4 SCALABILITY

### B.4.1 THE NUMBER OF WORKER AGENTS

To evaluate the scalability of the MoA workflow, we increase the number of worker agents from 3 to 6, 9, 12, and 15, while keeping the newly added agents identical in configuration to the original ones. Metrics from agents using the same LLM are aggregated for reporting. To clearly illustrate how efficiency evolves with increasing numbers of worker agents, we list separate tables (Table 19, 20, 21, 22, 23, 24, 25) for each framework.

Table 19: Scalability Evaluation of AlpacaEval Using AgentScope

| Number of Worker Agent | | | 3 | 6 | 9 | 12 | 15 |
|---|---|---|---|---|---|---|---|
| **Token** | llama | prompt | 85.451 | 137.84 | 206.76 | 275.68 | 344.6 |
| | | output | 382.45 | 796.68 | 1204.91 | 1641.18 | 2021.9 |
| | | total | 467.901 | 934.52 | 1411.67 | 1916.86 | 2366.5 |
| | qwen | prompt | 61.815 | 89.92 | 134.88 | 179.84 | 224.8 |
| | | output | 311.109 | 555.47 | 848.94 | 1139.47 | 1497.77 |
| | | total | 372.924 | 645.39 | 983.82 | 1319.31 | 1722.57 |
| | deepseek | prompt | 52.478 | 71.74 | 107.61 | 143.48 | 179.35 |
| | | output | 416.639 | 841.37 | 1253.25 | 1704.54 | 2100.42 |
| | | total | 469.117 | 913.11 | 1360.86 | 1848.02 | 2279.77 |
| | gpt | prompt | 1138.243 | 2237.83 | 3351.55 | 4542.02 | 5677.57 |
| | | output | 352.564 | 412.43 | 439.44 | 442.62 | 434.15 |
| | | total | 1490.807 | 2650.26 | 3790.99 | 4984.64 | 6111.72 |
| **Time** | llama | | 6.063 | 12.76 | 19.307 | 25.547 | 35.311 |
| | qwen | | 3.415 | 6.523 | 10.819 | 13.866 | 18.237 |
| | deepseek | | 13.726 | 32.81 | 48.833 | 67.114 | 84.318 |
| | gpt | | 8.89 | 15.468 | 14.33 | 14.373 | 15.813 |
| | total | | 32.119 | 67.607 | 93.357 | 122.987 | 153.784 |
| **Communication** | prompt to agent1 | | 284.078/118 | 389.8/236 | 584.7/354 | 779.6/472 | 974.5/590 |
| | prompt to agent2 | | 284.078/118 | 389.8/236 | 584.7/354 | 779.6/472 | 974.5/590 |
| | prompt to agent3 | | 284.078/118 | 389.8/236 | 584.7/354 | 779.6/472 | 974.5/590 |
| | agent1 to aggregator | | 1659.318/124 | 3256.270/250 | 4960.120/375 | 6718.820/500 | 8266.330/625 |
| | agent2 to aggregator | | 1511.311/122 | 2375.700/246 | 4051.120/369 | 5477.260/492 | 7080.860/615 |
| | agent3 to aggregator | | 1889.247/126 | 3705.450/254 | 5510.530/381 | 7497.600/508 | 9255.700/635 |

Table 20: Scalability Evaluation of AlpacaEval Using AutoGen

| Number of Worker Agent | | | 3 | 6 | 9 | 12 | 15 |
|---|---|---|---|---|---|---|---|
| **Token** | llama | prompt | 70.49 | 104.14 | 158.76 | 211.68 | 264.6 |
| | | output | 431.96 | 1004.94 | 1526.56 | 2028.21 | 2529.62 |
| | | total | 502.45 | 1109.08 | 1685.32 | 2239.89 | 2794.22 |
| | qwen | prompt | 64.85 | 93.18 | 140.88 | 187.84 | 234.8 |
| | | output | 447.45 | 993.87 | 1532.12 | 1940.46 | 2419.98 |
| | | total | 512.31 | 1087.05 | 1673 | 2128.3 | 2654.78 |
| | deepseek | prompt | 38.5 | 40.68 | 62.61 | 83.48 | 104.35 |
| | | output | 503.37 | 1109.77 | 1686.42 | 2249.68 | 2802.48 |
| | | total | 541.87 | 1150.45 | 1749.03 | 2333.16 | 2906.83 |
| | gpt | prompt | 1529.96 | 3194.7 | 4830 | 6290.22 | 7807.46 |
| | | output | 450.63 | 670.29 | 716.41 | 700.94 | 722.88 |
| | | total | 1980.59 | 3864.99 | 5546.41 | 6991.16 | 8530.34 |
| **Time** | llama | | 7.812 | 14.667 | 25.424 | 34.833 | 37.816 |
| | qwen | | 3.977 | 12.653 | 21.064 | 28.736 | 35.71 |
| | deepseek | | 26.745 | 46.011 | 71.345 | 71.98 | 104.207 |
| | gpt | | 8.274 | 19.816 | 22.41 | 30.398 | 17.817 |
| | total | | 36.854 | 47.339 | 50.843 | 55.6 | 46.428 |
| **Communication** | prompt to agent1 | | 209.08/44.01 | 236.48/86.04 | 359.7/129.06 | 479.6/172.08 | 599.5/215.1 |
| | prompt to agent2 | | 209.08/44.01 | 236.48/86.04 | 359.7/129.06 | 479.6/172.08 | 599.5/215.1 |
| | prompt to agent3 | | 209.08/44.01 | 236.48/86.04 | 359.7/129.06 | 479.6/172.08 | 599.5/215.1 |
| | agent1 to aggregator | | 2066.04/52.24 | 4618.13/103.41 | 7069.64/156.52 | 9297.61/208.1 | 11541.91/258.55 |
| | agent2 to aggregator | | 2071.24/57.38 | 4450.9/112.37 | 6777.28/172.84 | 8661.68/217.75 | 10768.69/271.29 |
| | agent3 to aggregator | | 2156.04/66.81 | 4604.89/128.62 | 7399.95/204.09 | 9384.64/258.11 | 11497.32/317.86 |

Table 21: Scalability Evaluation of AlpacaEval Using LangChain

| Number of Worker Agent | | | 3 | 6 | 9 | 12 | 15 |
|---|---|---|---|---|---|---|---|
| **Token** | llama | prompt | 70.49 | 105.84 | 158.76 | 211.68 | 264.60 |
| | | output | 428.55 | 1054.54 | 1518.52 | 2037.28 | 2537.08 |
| | | total | 499.04 | 1160.38 | 1677.28 | 2248.96 | 2801.68 |
| | qwen | prompt | 64.84 | 93.92 | 140.88 | 187.84 | 234.80 |
| | | output | 446.05 | 1007.95 | 1446.68 | 2017.43 | 2436.53 |
| | | total | 510.91 | 1101.87 | 1587.56 | 2205.27 | 2671.33 |
| | deepseek | prompt | 38.50 | 41.74 | 62.61 | 83.48 | 104.35 |
| | | output | 501.11 | 1132.22 | 1677.97 | 2224.98 | 2792.75 |
| | | total | 539.61 | 1173.96 | 1740.58 | 2308.46 | 2897.10 |
| | gpt | prompt | 1522.48 | 3300.82 | 4734.44 | 6353.31 | 7823.07 |
| | | output | 444.81 | 693.66 | 661.37 | 685.78 | 700.94 |
| | | total | 1967.29 | 3994.48 | 5395.81 | 7039.09 | 8524.01 |
| **Time** | llama | | 8.275 | 12.061 | 19.123 | 23.213 | 34.437 |
| | qwen | | 4.480 | 10.838 | 16.584 | 24.812 | 29.335 |
| | deepseek | | 23.084 | 40.801 | 66.156 | 73.476 | 115.888 |
| | gpt | | 10.699 | 13.741 | 17.592 | 33.688 | 32.068 |
| | total | | 36.502 | 37.958 | 47.112 | 59.725 | 66.075 |
| **Communication** | prompt to agent1 | | 165.07/0 | 153.76/0 | 230.64/0 | 307.52/0 | 384.4/0 |
| | prompt to agent2 | | 165.07/0 | 153.76/0 | 230.64/0 | 307.52/0 | 384.4/0 |
| | prompt to agent3 | | 165.07/0 | 153.76/0 | 230.64/0 | 307.52/0 | 384.4/0 |
| | agent1 to aggregator | | 1983.02/3 | 4703.67/6 | 6787.84/9 | 9117.26/13 | 11314.20/17 |
| | agent2 to aggregator | | 2011.83/3 | 4334.61/6 | 6286.30/9 | 8621.19/13 | 10546.46/17 |
| | agent3 to aggregator | | 2072.98/3 | 4529.70/6 | 6702.62/9 | 8880.47/13 | 11164.34/17 |

Table 22: Scalability Evaluation of AlpacaEval Using PydanticAI

| Number of Worker Agent | | | 3 | 6 | 9 | 12 | 15 |
|---|---|---|---|---|---|---|---|
| **Token** | llama | prompt | 61.347 | 95.5 | 126.29 | 139.77 | 161.71 |
| | | output | 429.543 | 938.08 | 1273.35 | 1327.71 | 1559.8 |
| | | total | 490.889 | 1033.58 | 1399.64 | 1467.48 | 1721.51 |
| | qwen | prompt | 41.217 | 58.39 | 76.32 | 80.85 | 94.52 |
| | | output | 433.739 | 939.44 | 1213.31 | 1257.55 | 1608.87 |
| | | total | 474.957 | 997.83 | 1289.63 | 1338.4 | 1703.39 |
| | deepseek | prompt | 31.802 | 41.44 | 50.5 | 50.88 | 58 |
| | | output | 434.81 | 931.31 | 1210.15 | 1150.95 | 1311.62 |
| | | total | 485.612 | 972.75 | 1260.65 | 1201.83 | 1369.62 |
| | gpt | prompt | 1845.724 | 3531.53 | 4673.28 | 4739.51 | 5684.52 |
| | | output | 596.876 | 636.99 | 633.62 | 637.09 | 691.85 |
| | | total | 2442.6 | 4168.52 | 5306.9 | 5376.6 | 6376.37 |
| **Time** | llama | | 6.503 | 15.15 | 16.68 | 19.71 | 21.15 |
| | qwen | | 3.441 | 8.38 | 11.2 | 11.59 | 13.41 |
| | deepseek | | 17.79 | 33.34 | 42.14 | 40.71 | 47.19 |
| | gpt | | 27.486 | 22.05 | 90.94 | 91.35 | 41.02 |
| | total | | 46.45 | 42.24 | 110.78 | 111.4 | 62.13 |
| **Communication** | prompt to agent1 | | 96.022/0 | 88.12/0 | 113.86/0 | 124.09/0 | 134.34/0 |
| | prompt to agent2 | | 95.425/0 | 93.84/0 | 118.13/0 | 119.19/0 | 131.11/0 |
| | prompt to agent3 | | 97.116/0 | 94.73/0 | 108.99/0 | 103.12/0 | 113.92/0 |
| | agent1 to aggregator | | 2000.542/0 | 4154.19/0 | 5693.77/0 | 6003.71/0 | 6851.79/0 |
| | agent2 to aggregator | | 1927.093/0 | 4002.04/0 | 5302.46/0 | 5314.6/0 | 6682.15/0 |
| | agent3 to aggregator | | 1892.344/0 | 3773.26/0 | 4941.13/0 | 4729.39/0 | 5284.75/0 |

Table 23: Scalability Evaluation of AlpacaEval Using CrewAI

| Number of Worker Agent | | | 3 | 6 | 9 | 12 | 15 |
|---|---|---|---|---|---|---|---|
| **Token** | llama | prompt | 298.25 | 536.95 | 706.39 | 760.54 | 795.95 |
| | | output | 518.95 | 1186.13 | 1495.89 | 1597.78 | 1741.84 |
| | | total | 817.201 | 1723.09 | 2202.26 | 2358.31 | 2537.63 |
| | qwen | prompt | 258.083 | 432.87 | 565.05 | 571.23 | 589.12 |
| | | output | 398.618 | 862.44 | 1123.05 | 1088.7 | 1119.49 |
| | | total | 656.702 | 1309.12 | 1688.11 | 1650.93 | 1708.61 |
| | deepseek | prompt | 313.01 | 432.87 | 526 | 544.75 | 668.04 |
| | | output | 571.79 | 1007.86 | 1147.04 | 1181.72 | 1436.84 |
| | | total | 884.808 | 1440.73 | 1673.04 | 1726.48 | 2104.88 |
| | gpt | prompt | 11694.576 | 28948.53 | 49040.19 | 54145.65 | 72234.23 |
| | | output | 679.15 | 1136.86 | 1320.35 | 1363.42 | 1614.66 |
| | | total | 12373.72 | 30085.4 | 50360.55 | 55509.07 | 73848.9 |
| **Time** | llama | | 8.835 | 20.9 | 32.04 | 44.25 | 27.61 |
| | qwen | | 3.837 | 7.7 | 16.64 | 13.84 | 14.61 |
| | deepseek | | 21.946 | 32.49 | 48.37 | 50.72 | 45.43 |
| | gpt | | 23.114 | 53.26 | 101.92 | 102.374 | 159.36 |
| | total | | 64 | 120.54 | 212.76 | 218.34 | 245.26 |
| **Communication** | prompt to agent1 | | 514.962/0 | 925.12/0 | 1425.23/0 | 1724.32/0 | 1963.23/0 |
| | prompt to agent2 | | 483.740/0 | 912.35/0 | 1252.74/0 | 1328/0 | 1456.32/0 |
| | prompt to agent3 | | 619.516/0 | 900.54.5/0 | 1386.75/0 | 1327.32/0 | 1587.73/0 |
| | agent1 to aggregator | | 2497.929/0 | 5921.52/0 | 7929.36/0 | 8623.56/0 | 9765.36/0 |
| | agent2 to aggregator | | 1754.701/0 | 4421.22/0 | 6342.21/0 | 7021.42/0 | 8126.57/0 |
| | agent3 to aggregator | | 2151.097/0 | 4783.14/0 | 6433.52/0 | 6798.21/0 | 7998.67/0 |

Table 24: Scalability Evaluation of AlpacaEval Using LlamaIndex

| Number of Worker Agent | | | 3 | 6 | 9 | 12 | 15 |
|---|---|---|---|---|---|---|---|
| **Token** | llama | prompt | 70.49 | 105.84 | 158.76 | 211.68 | 264.6 |
| | | output | 430.216 | 1007.91 | 1502.61 | 2012.48 | 2501.66 |
| | | total | 500.707 | 1113.75 | 1661.37 | 2224.16 | 2766.26 |
| | qwen | prompt | 64.81 | 93.92 | 140.88 | 187.84 | 234.8 |
| | | output | 441.738 | 972.25 | 1431.39 | 1914.73 | 2420.34 |
| | | total | 506.548 | 1066.17 | 1572.27 | 2102.57 | 2655.14 |
| | deepseek | prompt | 38.485 | 41.74 | 62.61 | 83.48 | 104.35 |
| | | output | 495.306 | 1107.88 | 1695.19 | 2216.87 | 2794.66 |
| | | total | 533.791 | 1149.62 | 1757.8 | 2300.35 | 2899.01 |
| | gpt | prompt | 42.083 | 24.68 | 24.68 | 24.68 | 24.68 |
| | | output | 350.386 | 515.31 | 541.38 | 539.22 | 528.1 |
| | | total | 392.47 | 539.99 | 566.06 | 563.9 | 552.78 |
| **Time** | llama | | 6.069 | 12.44 | 18.98 | 25.65 | 35.58 |
| | qwen | | 4.787 | 10.69 | 14.77 | 22.23 | 27.49 |
| | deepseek | | 20.829 | 41.18 | 61.97 | 81.83 | 93.12 |
| | gpt | | 5.849 | 9.39 | 9.66 | 10.4 | 16.06 |
| | total | | 27.318 | 36.87 | 43.85 | 53.77 | 67.23 |
| **Communication** | prompt to agent1 | | 1180.078/898 | 2181.8/1796.0 | 3272.7/2694.0 | 4363.6/3592.0 | 5454.5/4490.0 |
| | prompt to agent2 | | 1171.078/889 | 2163.8/1778.0 | 3245.7/2667.0 | 4327.6/3556.0 | 5409.5/4445.0 |
| | prompt to agent3 | | 1164.078/882 | 2149.8/1764.0 | 3224.7/2646.0 | 4299.6/3528.0 | 5374.5/4410.0 |
| | agent1 to aggregator | | 2022.417/33.689 | 4585.09/67.1 | 6813.56/99.75 | 9126.32/133.64 | 11342.05/169.22 |
| | agent2 to aggregator | | 2054.878/39.118 | 4372.32/72.31 | 6456.6/106.13 | 8647.86/143.64 | 10907.85/181.15 |
| | agent3 to aggregator | | 2116.377/48.641 | 4512.6/90.07 | 6923.71/137.8 | 9081.69/180.66 | 11437.99/227.25 |

Table 25: Scalability Evaluation of AlpacaEval Using Phidata

| Number of Worker Agent | | | 3 | 6 | 9 | 12 | 15 |
|---|---|---|---|---|---|---|---|
| **Token** | llama | prompt | 118.846 | 114.5 | 110.58 | 116.61 | 118.06 |
| | | output | 438.078 | 555.91 | 551.62 | 576.04 | 603.61 |
| | | total | 556.924 | 670.41 | 662.2 | 692.65 | 721.67 |
| | qwen | prompt | 93.899 | 87.57 | 83.21 | 90.21 | 91.97 |
| | | output | 463.795 | 634.08 | 621.29 | 663.48 | 707.2 |
| | | total | 557.694 | 721.65 | 704.5 | 753.69 | 799.17 |
| | deepseek | prompt | 83.391 | 76.54 | 72.94 | 77.18 | 78.63 |
| | | output | 440.691 | 505.74 | 525.82 | 527.8 | 545.07 |
| | | total | 524.082 | 582.28 | 598.76 | 604.98 | 623.7 |
| | gpt | prompt | 3003.319 | 4180.34 | 5040.94 | 5973.25 | 6991.86 |
| | | output | 756.689 | 785.45 | 778.76 | 795.1 | 801.86 |
| | | total | 3760.009 | 4965.79 | 5819.7 | 6768.35 | 7793.72 |
| **Time** | llama | | 6.152 | 6.55 | 6.55 | 9.12 | 10.33 |
| | qwen | | 4.707 | 6.75 | 5.27 | 6.09 | 6.56 |
| | deepseek | | 16.456 | 15.43 | 16.6 | 19.32 | 22.07 |
| | gpt | | 14.208 | 23.13 | 25.68 | 31.67 | 31.7 |
| | total | | 50.217 | 60.42 | 63.84 | 78.8 | 83.42 |
| **Communication** | prompt to agent1 | | 354.508/0 | 325.7/0 | 310.63/0 | 329.16/0 | 334.87/0 |
| | prompt to agent2 | | 341.160/0 | 309.01/0 | 293.84/0 | 319.17/0 | 326.58/0 |
| | prompt to agent3 | | 343.219/0 | 304.15/0 | 288.79/0 | 307.71/0 | 314.94/0 |
| | agent1 to aggregator | | 6128.259/2639.113 | 7105.44/3163.16 | 6961.87/3105.45 | 7291.8/3252.42 | 7582.27/3388.25 |
| | agent2 to aggregator | | 6131.272/2629.426 | 7475.54/3354.1 | 7269.53/3267.58 | 7792.87/3505.45 | 8121.06/3656.93 |
| | agent3 to aggregator | | 5715.126/2465.817 | 6165.11/2699.97 | 6196.23/2734.51 | 6342.37/2791.33 | 6571.72/2891.08 |

### B.4.2 Number of Tools

*Insight 10: Increasing the number of tools has only a minimal impact on execution time across frameworks, but it leads to a noticeable variation in LLM token usage and can cause execution failures when the input exceeds the LLMs maximum context length.*

**Key Observations** We conduct scalability experiments on the GAIA dataset, examining the effect of varying the number of tools across different frameworks. In addition to each frameworks original tool set, we introduce extra LeetCode-solving tools, which are irrelevant for solving the GAIA dataset. The results in Table 26 and 27 show that while increasing the number of tools has only a minimal impact on execution time, it leads to a noticeable increase in LLM token usage. In addition, it can be observed that as the number of tools increases, some test samples encountered execution failures because the input exceed the LLMs maximum context length (see Table 28). Notably, in the LlamaIndex framework, the addition of the extra LeetCode-solving tools results in a significant decrease in both token consumption and execution time.

**Underlying Mechanism-12: Reduced Tool-Call Tendency** Increasing the size of the tool inventory paradoxically reduces the agents propensity to invoke tools. On the same test set, adding 10 or 20 LeetCode-solving tools raises the number of queries that make no tool calls from 17 (no extras) to 27 and 25, respectively. Consistent with this shift, the total tool-call counts drop from 630 (0 extra tools) to 454 and 467 (10 and 20 extra tools). These patterns indicate a shallower ReAct trajectory, which in turn reduces LLM token consumption and overall execution time.

**Potential Optimizations** Building on these findings, agent frameworks should emphasize relevance-aware tool-set curation and dynamic exposure to tools to contain prompt growth and reduce the risk of context-length failures. Regulating ReAct depth and enforcing explicit token budgets can curb unnecessary tool exploration, while compact, standardized tool specifications help decouple token usage from catalog size.

Table 26: Effect of LeetCode-solving tools on execution time (seconds)

| | LangChain | AutoGen | AgentScope | CrewAI | LlamaIndex | Phidata | PydanticAI |
|---|---|---|---|---|---|---|---|
| no LeetCode-solving tools | 12.86 | 8.41 | 19.57 | 11.87 | 24.26 | 10.23 | 10.31 |
| 10 LeetCode-solving tools | 11.79 | 8.58 | 22.31 | 10.35 | 19.47 | 10.99 | 8.33 |
| 20 LeetCode-solving tools | 10.78 | 8.36 | 21.95 | 11.14 | 20.89 | 10.98 | 9.58 |

Table 27: Effect of LeetCode-solving tools on Token

| | no LeetCode-solving tools | | | 10 LeetCode-solving tools | | | 20 LeetCode-solving tools | | |
|---|---|---|---|---|---|---|---|---|---|
| | **Prompt** | **Output** | **Total** | **Prompt** | **Output** | **Total** | **Prompt** | **Output** | **Total** |
| LangChain | 7199.33 | 553.2 | 7753 | 11489.89 | 586.61 | 12076.50 | 12779.90 | 502.75 | 13282.65 |
| AutoGen | 1195.98 | 185.19 | 1381.18 | 2200.19 | 191.82 | 2392.01 | 3011.2 | 182.87 | 3194.07 |
| AgentScope | 17161.55 | 828.68 | 17990.23 | 31878.31 | 780.23 | 32658.54 | 32464.93 | 804.56 | 33269.48 |
| CrewAI | 16475.12 | 582.82 | 17057.95 | 11670.07 | 552.16 | 12222.23 | 17398.34 | 557.75 | 17956.09 |
| LlamaIndex | 101042.29 | 729.57 | 101771.86 | 35111.65 | 348.83 | 35460.48 | 32899.47 | 253.21 | 33152.68 |
| Phidata | 3293.59 | 270.75 | 3564.33 | 4957.96 | 295.79 | 5253.75 | 6104.55 | 267.34 | 6371.88 |
| PydanticAI | 13273.91 | 373.74 | 13647.66 | 12356.90 | 321.95 | 12678.85 | 16682.93 | 324.13 | 17025.06 |

Table 28: Number of Failed Runs

| | **LangChain** | **AutoGen** | **AgentScope** | **CrewAI** | **LlamaIndex** | **Phidata** | **PydanticAI** |
|---|---|---|---|---|---|---|---|
| no irrelevant tools | 0 | 0 | 1 | 1 | 1 | 0 | 1 |
| 10 irrelevant tools | 0 | 0 | 2 | 1 | 1 | 1 | 1 |
| 20 irrelevant tools | 0 | 0 | 4 | 3 | 1 | 0 | 1 |

## B.5 CORRELATION ANALYSIS

### B.5.1 TOKEN CONSUMPTION AND LATENCY

Intuitively, it is reasonable to expect a positive relationship between an agents token consumption and its execution time, since a larger number of tokens typically corresponds to longer LLM processing durations, which constitute a major component of overall runtime. To quantitatively verify this intuition, we conduct a correlation analysis between these two metrics.

We treat the measurement result of each query as an individual data point and compute the Pearson correlation coefficients between token counts and execution time for each framework, with the results summarized in Table 29. As shown, datasets that involve tools with inherently long processing times (e.g., MMLU with RAG retrieval and OK-VQA with image-processing tools) tend to yield smaller correlation coefficients. In contrast, datasets where LLM calls dominate the execution process (e.g., AlpacaEval and HumanEval) exhibit noticeably stronger correlations. In addition, we observe that the correlation between tokens and time in the AutoGen framework is notably weak, which may stem from its underlying asynchronous execution mechanism.

Table 29: Token−Time Pearson Correlation

| | LangChain | AutoGen | AgentScope | CrewAI | LlamaIndex | Phidata | PydanticAI |
|---|---|---|---|---|---|---|---|
| **GAIA** | 0.7554 | 0.11 | 0.7355 | 0.8745 | 0.6273 | 0.7821 | 0.6276 |
| **OK-VQA** | - | -0.0463 | 0.4527 | 0.295 | 0.0676 | 0.0284 | 0.0779 |
| **ScienceWorld** | 0.9558 | - | 0.9426 | 0.9581 | 0.9902 | 0.9265 | 0.8784 |

### B.5.2 TOKEN CONSUMPTION AND ACCURACY

*Insight 11: Token usage and accuracy are not strongly correlated, and spending more tokens or LLM calls does not reliably lead to better correctness.*

We also compute the Pearson correlation coefficients between token counts and accuracy. For GAIA and VQA, each successful query is assigned a value of 1 and each failed query a value of 0, whereas ScienceWorld uses the querys numerical score. The results are presented in Table 30. We observe that nearly all Pearson coefficients are negative, which is likely attributable to the fact that queries requiring a larger number of tokens tend to be inherently more challenging and therefore more prone to failure. We observe that the correlation between token consumption and accuracy is generally weak across all framework-dataset pairs (all $|r| < 0.35$). Moreover, most coefficients are negative, indicating that within a given framework successful queries tend to use slightly fewer tokens than failed ones. This suggests that simply "trying harder" with more steps and longer prompts does not

systematically improve correctness; on the contrary, harder instances often trigger longer trajectories that still end in failure. Overall, these results indicate that our efficiency measurements are not merely capturing frameworks that "fail quickly"—if anything, failures are frequently associated with *higher* token usage.

Table 30: Token−Accuracy Pearson Correlation

|  | Lanchain | AutoGen | AgentScope | CrewAI | LlamaIndex | Phidata | PydanticAI |
|---|---|---|---|---|---|---|---|
| **GAIA** | -0.1200 | -0.0123 | -0.0884 | -0.2026 | -0.0371 | -0.0551 | 0.2408 |
| **OK-VQA** | - | -0.1215 | -0.2054 | -0.1796 | -0.0440 | -0.1249 | -0.1447 |
| **ScienceWorld** | -0.2730 | - | -0.3113 | -0.0237 | -0.0919 | -0.1485 | -0.1485 |

We also compute the Pearson coefficient between the number of LLM calls and accuracy within a single framework. The results are available at Table 31, which are consistent with Table 30.

Table 31: Rounds−Accuracy Pearson Correlation

|  | LangChain | AutoGen | AgentScope | CrewAI | LlamaIndex | Phidata | PydanticAI |
|---|---|---|---|---|---|---|---|
| **GAIA** | -0.1008 | -0.1211 | -0.1508 | -0.2456 | 0.0053 | -0.0789 | 0.1868 |
| **OK-VQA** | - | - | -0.2093 | -0.2237 | -0.0406 | -0.0317 | -0.0515 |
| **ScienceWorld** | -0.2640 | - | -0.3115 | 0.0034 | -0.3141 | -0.0564 | -0.1566 |

### B.5.3 REASONING ROUNDS AND EFFICIENCY

The Pearson correlation coefficients for token count, time, and reasoning rounds across each framework are shown in Tables 32 and 33. Note that, aside from the previously mentioned cases where measurements are not conducted (the ScienceWorld dataset under AutoGen and the OK-VQA dataset under LangChain), the Pearson correlation cannot be calculated for AutoGen on OK-VQA because nearly all queries have exactly two reasoning rounds.

We observe that reasoning rounds and time both show a positive correlation with token count. In particular, the correlation between reasoning rounds and token count is very strong on the OK-VQA and ScienceWorld datasets, while it is weaker on GAIA. A likely explanation is that GAIA often requires reading file contents, which can introduce a large number of prompt tokens in a single reasoning step, thus weakening the overall correlation.

Table 32: Rounds−Time Pearson Correlation

|  | LangChain | AutoGen | AgentScope | CrewAI | LlamaIndex | Phidata | PydanticAI |
|---|---|---|---|---|---|---|---|
| **GAIA** | 0.7575 | 0.1287 | 0.2905 | 0.8947 | 0.5018 | 0.9017 | 0.5907 |
| **OK-VQA** | - | - | 0.4763 | 0.3035 | 0.1344 | 0.0635 | 0.0259 |
| **ScienceWorld** | 0.9448 | - | 0.9478 | 0.9575 | 0.9854 | 0.9354 | 0.8879 |

Table 33: Rounds−Token Pearson Correlation

|  | LangChain | AutoGen | AgentScope | CrewAI | LlamaIndex | Phidata | PydanticAI |
|---|---|---|---|---|---|---|---|
| **GAIA** | 0.9506 | 0.2231 | 0.6501 | 0.8346 | 0.7762 | 0.8468 | 0.6787 |
| **OK-VQA** | - | - | 0.9737 | 0.9565 | 0.9875 | 0.8611 | 0.8612 |
| **ScienceWorld** | 0.9861 | - | 0.9974 | 0.9675 | 0.9958 | 0.9858 | 0.9876 |

## B.6 EXTENDED ANALYSIS ON INSIGHT 1

Our experiments in Section 5.2 reveal a strong correlation between prompt token counts and execution time across frameworks (see Figure 3). This is primarily due to two factors: 1) the frequency of LLM calls and tool invocations per query; 2) memory accumulation across queries.

Figure 8 shows that CrewAI and AgentScope have significantly higher average LLM call frequencies per query (5.33 and 4.78) compared to LlamaIndex, PydanticAI, and Phidata (2.76, 2.79, and 3.38). This difference explains their greater token consumption and longer runtimes, which stem from more frequent LLM calls and the resulting memory accumulation.

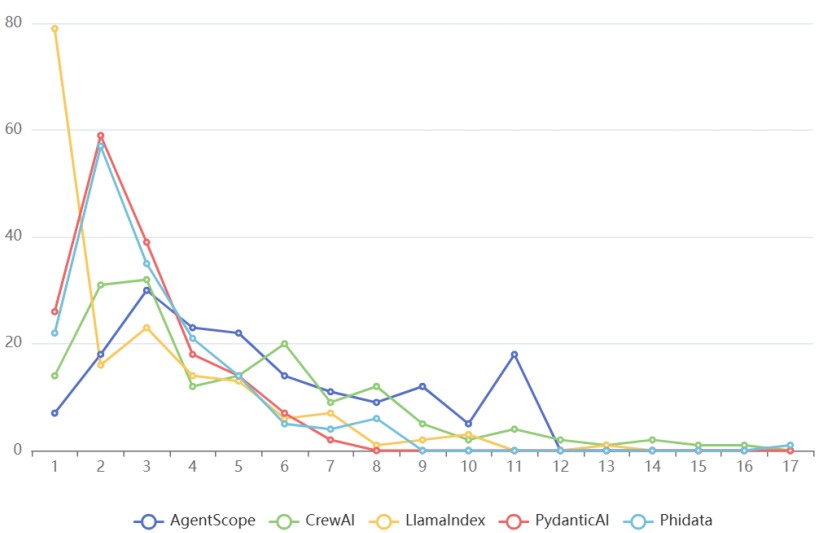

Figure 8: LLM Call Frequency per Query across Different Frameworks

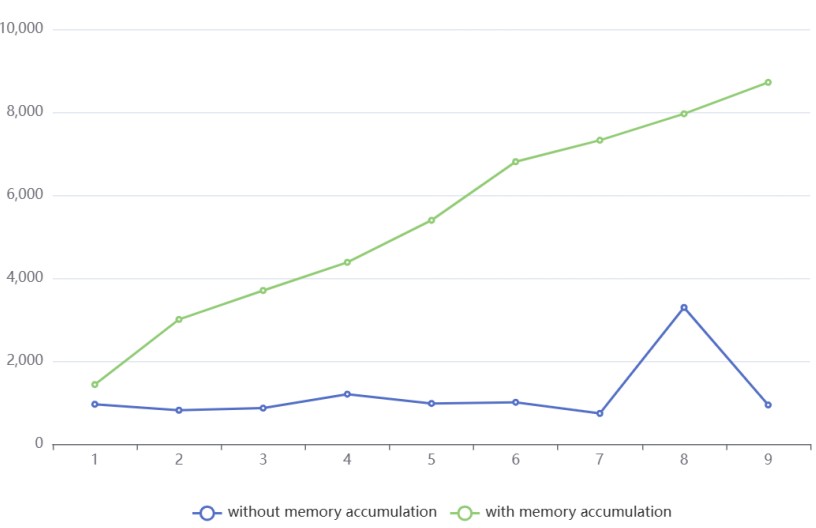

Figure 9: Memory Accumulation Impact

During the experiments, we observe the following patterns, indicating that some frameworks invoke tools more frequently than others:

1) AgentScope and CrewAI frequently invoke the Web tool to obtain precise results, leading to substantially higher token usage due to lengthy text outputs. In our tests, they called the Web tool 494 and 608 times respectively, far exceeding the maximum of 102 observed in other frameworks.

2) AgentScope often writes and executes code to solve problems, which requires returning large code blocks that further increase token usage. It used the code execution tool 122 times, while other frameworks did so no more than 21 times.

Moreover, AgentScope stands out for retaining conversational memory across queries by continuously appending prior interactions to the prompt. Unlike earlier tests that re-instantiated the Agent to avoid memory buildup, running 9 GAIA queries without resets confirmed significant memory accumulation (see Figure 9).

Meanwhile, in our MoA workflow experiments, we observed that some frameworks invoke worker agents in parallel, whereas others do so serially. Specifically, we observe that CrewAIs built-in MoA workflow integrates the previous worker agents output with the initial prompt, performs a secondary summarization, and then passes the result to the next worker agent. To further explore this behavior, we varied the order of worker agents in CrewAI and present the results in Table 34. Here, GLM, Qwen, DS, and GPT denote GLM-Z1-Rumination-32B-0414, Qwen2.5-7B-Instruct, DeepSeek-V3, and GPT-4o, respectively.

Table 34: The Impact of Agent Execution Order on Tokens

| Order | GLM→Qwen →DS | | | DS→Qwen→GLM | | | Qwen→DS →GLM | | |
|---|---|---|---|---|---|---|---|---|---|
| | Prompt | Output | Total | Prompt | Output | Total | Prompt | Output | Total |
| GLM | 1296.82 | 734.62 | 2031.44 | 2953.52 | 1909.5 | 4863.02 | 584.74 | 324.9 | 909.64 |
| Qwen | 241.86 | 383.12 | 624.98 | 279.96 | 557.84 | 837.8 | 255.92 | 525.3 | 781.22 |
| DS | 447.0 | 968.5 | 1415.5 | 279.36 | 568.14 | 847.5 | 246.38 | 556.64 | 803.02 |
| GPT | 36750.26 | 1119.44 | 37869.7 | 36732.26 | 1129.24 | 37861.5 | 17375.24 | 455.8 | 17831.04 |

## B.7 MODEL DIVERSITY

### B.7.1 CLAUDE-BASED RESULTS

Table 35: Claude-Based HumanEval Results

| Framework | Token | | | Time | | | Accuracy |
|---|---|---|---|---|---|---|---|
| | Prompt | Output | Total | LLM | Code executor | Total | |
| LangChain | 5568.08 | 675.88 | 6243.96 | 41.644 | 0.0140 | 41.932 | 0.585 |
| AutoGen | 920.84 | 292.88 | 1213.71 | 12.847 | 0.00047 | 13.182 | 0.823 |

Given that the majority of our experiments are implemented with GPT-4o, and considering the widespread adoption of open-source models, we additionally evaluate the Claude-3-Opus model on the HumanEval dataset within the LangChain and AutoGen frameworks. The results are presented in Table 35.

Notably, AutoGen exhibited slightly lower accuracy compared to GPT-based agents. Upon inspection, we found that Claude did not fabricate test data when invoking the Python execution tool, which rendered the self-checking mechanism ineffective. In LangChain, Claude occasionally emitted tool outputs directly, bypassing the expected format and causing execution failures.

These behaviors suggest that when using Claude-3-Opus as the underlying model for ReAct-style agents, further prompt adaptation may be necessary to ensure compatibility with existing framework toolchains.

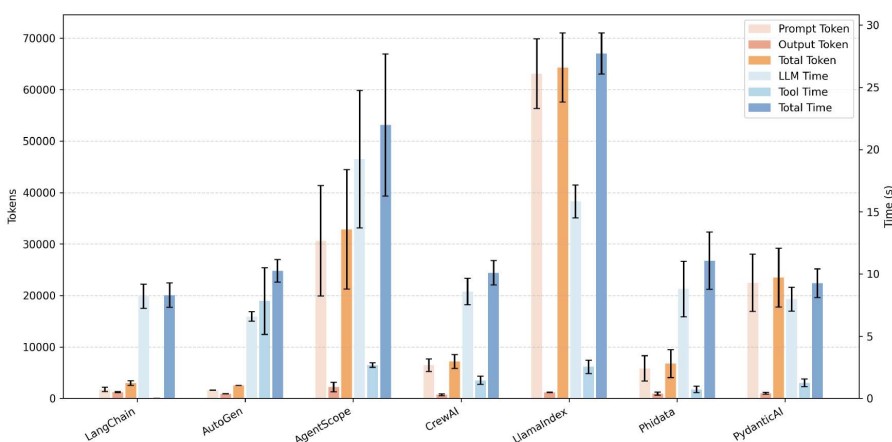

Figure 10: Consistency of Token Consumption and Latency Across Repeated GAIA Experiments Using the Qwen3-Next-80B-A3B-Instruct Model

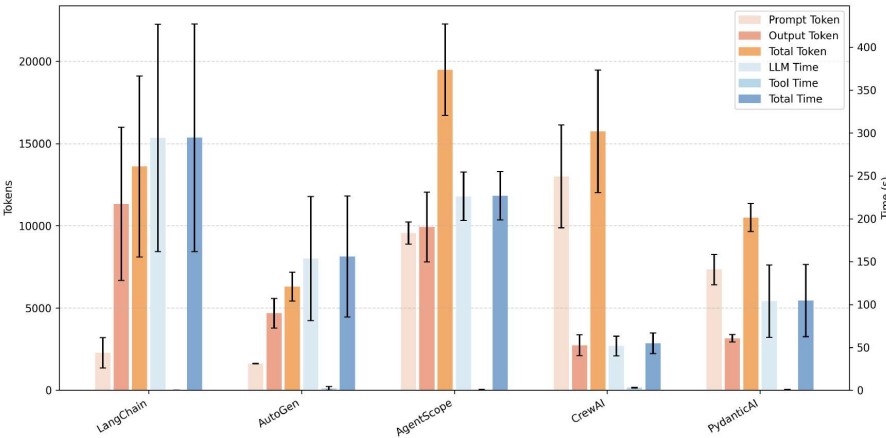

Figure 11: Consistency of Token Consumption and Latency Across Repeated GAIA Experiments Using the GLM-Z1-32B Model

### B.7.2 SLM-BASED RESULTS

To evaluate the capability of different frameworks to address complex problems when equipped with small language models, we conduct experiments on the GAIA dataset using 2 open-source SLMs: Qwen3-Next-80B-A3B-Instruct and GLM-Z1-32B. The experimental results are presented in Table 36 and 37, while the visualization of the stability of repeated runs is illustrated in Figure 10 and 11. The behaviour when running with these models is similar to our previous results. For example, LLM inference still dominates the end-to-end runtime (Insight 1), and tools that search for figure-related content introduce disproportionately high latency (Insight 2).

> *Insight 12: Different LLM-agent frameworks exhibit varying levels of adaptability to SLMs. Some frameworks remain robust when deployed with SLMs, whereas others fail to perform the task effectively.*

**Key Observations** It is worth noting that when the model is configured as GLM-Z1-32B, Phidata and LlamaIndex exhibit substantial freezing and task-execution failures, preventing the evaluation of their efficiency performance. To further investigate whether this issue becomes more pronounced with smaller-parameter models, we evaluate the executability of each framework using Mistral-7B-Instruct. The results indicate that LlamaIndex, AutoGen, PydanticAI, and Phidata all fail to complete the tasks successfully.

**Underlying Mechanism-13: Failure to Produce Valid Tool-Invocation Outputs or Blank Responses** In our experiments, we observe that the LlamaIndex framework fails to complete tasks because the small language model frequently returns empty outputs, causing the ReAct process to stall and ultimately terminate abnormally. In addition, some frameworks are unable to produce correctly formatted tool-invocation outputs, which prevents them from leveraging tools to retrieve necessary information. AutoGen is one such example. Unlike LlamaIndex, however, AutoGen does not terminate abruptly; instead, the small model subsequently generates hallucinated content, effectively pretending to have produced valid tool-call results.

Table 36: GAIA Detailed Results based on Qwen3-Next-80B-A3B-Instruct

|  |  | LangChain | AutoGen | AgentScope | CrewAI | LlamaIndex | Phidata | PydanticAI |
|---|---|---|---|---|---|---|---|---|
| **Token** | Prompt | 1798.33 | 1625.48 | 24076.86 | 8211.85 | 72089.98 | 3879.22 | 14663.88 |
|  | Output | 1344.44 | 874.24 | 1867.73 | 789.21 | 1189.57 | 876.74 | 814.09 |
|  | Total | 3142.77 | 2499.72 | 25944.59 | 9001.05 | 73279.55 | 4755.96 | 15477.97 |
| **Time** | llm | 9.09 | 6.95 | 16.70 | 8.57 | 17.69 | 8.39 | 6.91 |
|  | Search | - | 2.8603 | 1.1637 | 1.6999 | 1.2161 | 0.2309 | 0.4400 |
|  | PDF loader | - | 0.006604 | 0.30925 | 0.00221 | 0.00482 | 0.00422 | 0 |
|  | CSV reader | - | - | - | - | - | - | - |
|  | XLSX reader | - | - | - | 0.00139 | 0.01975 | 0.00603 | 0.00362 |
|  | Text file reader | - | 0.000184 | 0.000048 | 0.00054 | 0.00207 | 0.00017 | 0.000005 |
|  | Doc reader | - | - | - | - | 0.00015 | 0.00003 | - |
|  | MP3 loader | - | 0.51547 | 0.05150 | 0.13936 | 0.24851 | 0.12460 | 0.38613 |
|  | Figure loader | - | 0.71069 | 0.60076 | - | 0.29506 | - | 0.52619 |
|  | Video loader | - | - | 0.20742 | - | 0.10099 | - | - |
|  | Code executor | 0.00104 | 0.00002 | 0.24325 | 0.00278 | 1.41922 | 0.00119 | 0.00005 |
|  | Total tool time | 0.0010 | 4.09 | 2.58 | 1.85 | 3.31 | 0.37 | 1.36 |
|  | Total time | 9.27 | 11.40 | 19.34 | 10.44 | 29.72 | 10.32 | 8.29 |

Table 37: Accurate GAIA Detailed Results based on GLM-Z1-32B

|  |  | LangChain | AutoGen | AgentScope | CrewAI | PydanticAI |
|---|---|---|---|---|---|---|
| **Token** | Prompt | 3148.43 | 1625.94 | 9555.97 | 13626.33 | 8647.96 |
|  | Output | 13077.57 | 3713.18 | 10086.24 | 2527.60 | 3012.35 |
|  | Total | 16226.0 | 5339.12 | 19642.21 | 16153.93 | 11660.31 |
| **Time** | llm | 315.69 | 83.56 | 225.24 | 47.49 | 67.17 |
|  | Search | - | 4.02 | 0.37 | 2.50 | 0.31 |
|  | PDF loader | - | - | - | - | 0.0000051 |
|  | CSV reader | - | - | - | - | - |
|  | XLSX reader | - | - | - | 0.017 | 0.0069 |
|  | Text file reader | - | - | - | - | - |
|  | Doc reader | - | - | - | - | - |
|  | MP3 loader | - | - | - | 0.63 | 0.64 |
|  | Figure loader | - | 0.96 | 0.24 | - | - |
|  | Video loader | - | - | - | - | - |
|  | Code executor | 0.0021 | 0.000018 | 0.51 | 0.011 | - |
|  | Total tool time | 0.0021 | 4.98 | 1.14 | 3.15 | 0.96 |
|  | Total time | 315.81 | 88.73 | 226.44 | 50.93 | 68.14 |

## B.8 REPRODUCIBILITY VERIFICATION

Table 38: HumanEval Run 2

| Framework | Token | | | Time | | |
|---|---|---|---|---|---|---|
| | Prompt | Output | Total | LLM | Code executor | Total |
| LangChain | 6769.16 | 695.15 | 7464.31 | 27.063 | 0.01267 | 27.82 |
| AutoGen | 790.29 | 108.26 | 898.55 | 5.685 | 0.000353 | 5.711 |
| AgentScope | 2429.72 | 530.323 | 2960.043 | 13.42 | 0.121 | 13.57 |
| CrewAI | 10026.98 | 914.96 | 10941.95 | 29.75 | 0.0432 | 30.47 |
| LlamaIndex | 2052 | 347.9 | 2399.9 | 19.81 | 0.00381 | 19.84 |
| Phidata | 1083.32 | 376.46 | 1459.79 | 11 | 8.99E-05 | 16.3 |
| PydanticAI | 903.6 | 353.48 | 1257.08 | 9.13 | 2.32E-05 | 9.15 |

Table 39: HumanEval Run 3

| Framework | Token | | | Time | | |
|---|---|---|---|---|---|---|
| | Prompt | Output | Total | LLM | Code executor | Total |
| LangChain | 7953.34 | 832.63 | 8785.97 | 38.562 | 0.015723 | 39.471 |
| AutoGen | 769.72 | 105.78 | 875.5 | 8.027 | 0.000279 | 8.199 |
| AgentScope | 2804.341 | 568.36 | 3372.701 | 15.686 | 0.139 | 15.858 |
| CrewAI | 10822.16 | 867.08 | 11689.24 | 34.19 | 0.0342 | 34.98 |
| LlamaIndex | 2017.37 | 362.85 | 2380.23 | 20.61 | 0.00293 | 20.64 |
| Phidata | 1258.7 | 393.46 | 1652.16 | 9.36 | 0.000227 | 12.4 |
| PydanticAI | 874.49 | 340.66 | 1215.15 | 7.73 | 2.44E-05 | 7.74 |

Table 40: GAIA Run 1

| Frameworks | Token | | | Time | | | | |
|---|---|---|---|---|---|---|---|---|
| | Prompt | Output | Total | LLM | Search | PDF loader | CSV reader | XLSX reader |
| LangChain | 6493.9 | 562.42 | 7052.33 | 8.26 | 0.724 | 0.000713 | 2.73E-05 | - |
| AutoGen | 1078.7 | 183 | 1261.7 | 9.65 | 17.29 | 0.00347 | 0.00035 | 8.91E-05 |
| AgentScope | 19192.78 | 747.25 | 19940.02 | 12.03 | 1.32 | 1.48 | 0.000358 | 0.00147 |
| CrewAI | 31286.37 | 612.44 | 31898.81 | 34.55 | 4.66 | 0.0205 | 0.000138 | 0.00272 |
| LlamaIndex | 12370.81 | 688.83 | 13059.64 | 38.4 | 1.019 | 0.000618 | 4.63E-06 | 0.00196 |
| Phidata | 2387.39 | 260.78 | 2648.17 | 13.16 | 4.296 | 0.00257 | 8.37E-06 | 8.18E-05 |
| PydanticAI | 15680.58 | 410.12 | 16090.7 | 10.81 | 0.744 | 0.461 | 0.000302 | 0.000111 |

| time | | | | | | | |
|---|---|---|---|---|---|---|---|
| Text file reader | doc reader | MP3 loader | Figure loader | Video loader | Code executor | total tool time | total time |
| 0.0197 | - | - | - | - | 0.0176 | 0.762 | 10.15 |
| 4.63E-05 | 5.82E-05 | - | - | - | 1.15E-05 | 17.294 | 27.04 |
| 6.32E-06 | 2.52E-06 | 0.125 | 0.443 | 2.99E-06 | 0.996 | 4.359 | 16.575 |
| 0.000832 | 0.00015 | 0.000375 | 0.105 | - | 0.194 | 4.795 | 39.86 |
| 0.00113 | 3.94E-06 | 3.91E-06 | 0.839 | - | 2.248 | 47 |
| 4.24E-05 | 0.000141 | 0.098 | 0.075 | - | 0.000427 | 4.473 | 13.16 |
| 0.117 | 6.33E-05 | 0.0951 | 0.141 | - | 6.39E-05 | 1.558 | 11.68 |

Table 41: GAIA Run 2

| Frameworks | Token | | | Time | | | | |
| --- | --- | --- | --- | --- | --- | --- | --- | --- |
| | Prompt | Output | Total | LLM | Search | PDF loader | CSV reader | XLSX reader |
| LangChain | 6659.4 | 598.16 | 7257.56 | 17.61 | 0.78 | 0.000908 | 3.82E-05 | - |
| AutoGen | 1063.48 | 195.52 | 1259 | 4.206 | 11.477 | 0.000736 | 0.000223 | 0.000161 |
| AgentScope | 20787.67 | 785.02 | 21572.68 | 12.997 | 1.438 | 2.876 | 0.000248 | 0.000841 |
| CrewAI | 33422.3 | 564.65 | 33986.94 | 35.75 | 4.77 | 0.0072 | 0.000146 | 0.0023 |
| LlamaIndex | 15079.24 | 731.95 | 15811.19 | 35.69 | 1.196 | 0.000308 | 2.19E-06 | 0.0021 |
| Phidata | 2481.73 | 279.04 | 2760.76 | 5.25 | 4.055 | 0.00074 | 1.37E-05 | 0.000173 |
| PydanticAI | 11306.87 | 259.62 | 11566.48 | 5.361 | 1.12 | 0.535 | 0.000261 | 7.93E-05 |

| Text file reader | doc reader | MP3 loader | Figure loader | Video loader | Code executor | Total tool time | Total time |
| --- | --- | --- | --- | --- | --- | --- | --- |
| | | | | Time | | | |
| 0.0103 | - | - | - | - | 0.000699 | 0.797 | 18.89 |
| 3.39E-05 | 9.33E-05 | - | - | - | 1.58E-05 | 11.478 | 16.211 |
| 2.60E-06 | 2.00E-06 | 0.241 | 0.406 | 1.45E-06 | 0.285 | 5.248 | 18.55 |
| 0.000477 | 0.000141 | 0.000283 | 0.0314 | - | 0.00647 | 4.82 | 41.14 |
| 0.00042 | 9.75E-05 | 6.96E-06 | 0.399 | - | 1.196 | 2.794 | 46.28 |
| 0.000166 | 7.73E-05 | 0.144 | 0.108 | - | 0.000132 | 4.308 | 10.69 |
| 0.125 | 9.10E-05 | 0.186 | 0.126 | - | 1.75E-05 | 2.091 | 6.59 |

Table 42: GAIA Run 3

| Frameworks | Token | | | Time | | | | |
| --- | --- | --- | --- | --- | --- | --- | --- | --- |
| | Prompt | Output | Total | LLM | Search | PDF loader | CSV reader | XLSX reader |
| LangChain | 7262.24 | 651.28 | 7913.52 | 16.86 | 1.16 | 0.246 | 2.55E-05 | - |
| AutoGen | 1067.48 | 186.24 | 1253.72 | 10.59 | 17.33 | 0.000685 | 0.000285 | 0.000195 |
| AgentScope | 20689.4 | 761.78 | 21451.18 | 21.58 | 2.446 | 2.035 | 0.000199 | 0.0019 |
| CrewAI | 33866.8 | 621.44 | 34488.23 | 34.15 | 3.446 | 0.00617 | 0.000171 | 0.00251 |
| LlamaIndex | 19764.47 | 964 | 20728.47 | 61.89 | 2.395 | 0.00203 | 0.000678 | 0.00631 |
| Phidata | 2187.99 | 233.53 | 2421.52 | 13.81 | 3.92 | 0.000728 | 6.04E-06 | 0.000103 |
| PydanticAI | 13059.31 | 296.36 | 13355.67 | 15.76 | 0.783 | 0.637 | 3.79E-06 | 7.88E-05 |

| Text file reader | doc reader | MP3 loader | Figure loader | Figure loader | Code executor | Total tool time | Total time |
| --- | --- | --- | --- | --- | --- | --- | --- |
| | | | | time | | | |
| 0.00904 | - | - | - | - | 0.00125 | 1.417 | 18.78 |
| 1.70E-05 | 2.31E-04 | - | - | - | 2.00E-05 | 17.33 | 28.71 |
| 3.24E-06 | 4.85E-06 | 0.164 | 0.683 | 4.46E-06 | 1.88 | 7.215 | 29.03 |
| 0.00047 | 0.000141 | 0.000283 | - | - | 0.014 | 3.47 | 38.44 |
| 0.000464 | 0.000239 | 0.0405 | 0.69 | - | 0.307 | 3.443 | 74.998 |
| 0.000117 | 7.83E-05 | 0.0989 | 0.0788 | - | 0.000497 | 4.1 | 19.52 |
| 0.0382 | 5.67E-05 | 0.0824 | 0.151 | - | 5.66E-02 | 1.75 | 16.685 |

Table 43: GAIA Detailed Results on Correct Cases

| | | LangChain | AutoGen | AgentScope | CrewAI | LlamaIndex | Phidata | PydanticAI |
| --- | --- | --- | --- | --- | --- | --- | --- | --- |
| **Token** | Prompt | 6769.44 | 1147.29 | 16985.667 | 10267.65 | 7532.923 | 5351.03 | 13742.826 |
| | Output | 469.84 | 298.59 | 549.848 | 386 | 492.923 | 379.84 | 497.609 |
| | Total | 7239.28 | 1445.88 | 17535.515 | 10653.65 | 8025.846 | 5730.87 | 14240.435 |
| **Time** | llm | 15.239 | 9.482 | 27.945 | 9.7169 | 23.226 | 10.358 | 23.139 |
| | Search | 1.163 | 5.1078 | 1.321 | 1.607 | 0.6686 | 0.7241 | 0.9099 |
| | PDF loader | - | 0.00491 | 0.702 | 0.00768 | - | 0.006071 | - |
| | CSV reader | - | - | - | - | - | - | - |
| | XLSX reader | 0.270488 | 0.010234 | 0.00922 | 0.002742 | 0.006056 | 0.00682 | 0.0023606 |
| | Text file reader | - | 0.00001176 | - | 0.00104 | - | 0.000133 | 9.57586E-06 |
| | Doc reader | - | - | - | - | - | 0.314994 | - |
| | MP3 loader | - | - | 1.538 | - | - | - | - |
| | Figure loader | - | 1.44159412 | - | - | - | - | - |
| | Video loader | - | - | - | 0.000382 | - | - | - |
| | Code executor | 0.012068 | - | 0.0865 | 0.288544 | 0.09316 | 0.045673 | 3.34E-04 |
| | Total tool time | 1.44556 | 8.5645 | 3.657 | 1.9077 | 0.7679 | 1.0978 | 0.9126 |
| | Total time | 17.143 | 18.076 | 33.203 | 12.0688 | 24.0842 | 12.9508 | 24.104 |

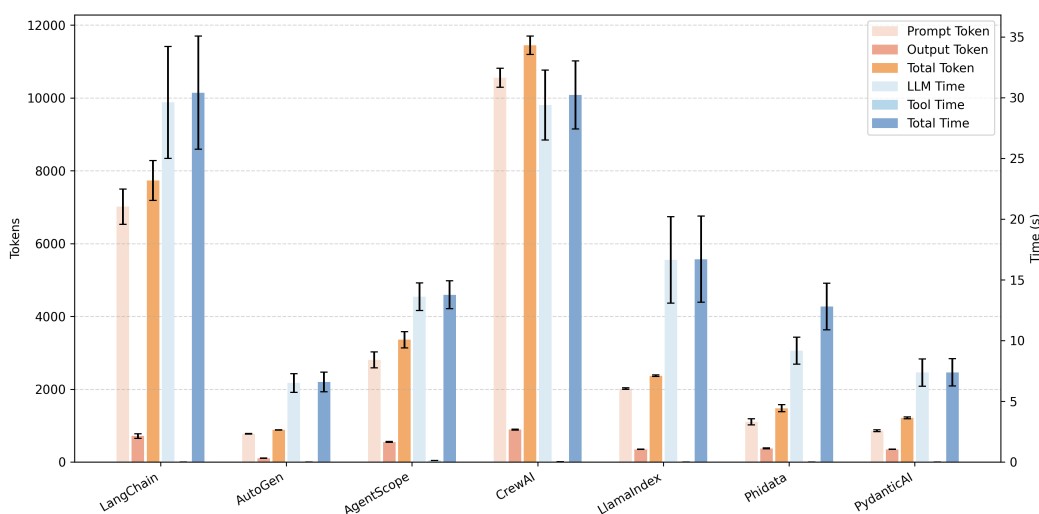

Figure 12: Consistency of Token Consumption and Latency in Repeated Experiments (HumanEval)

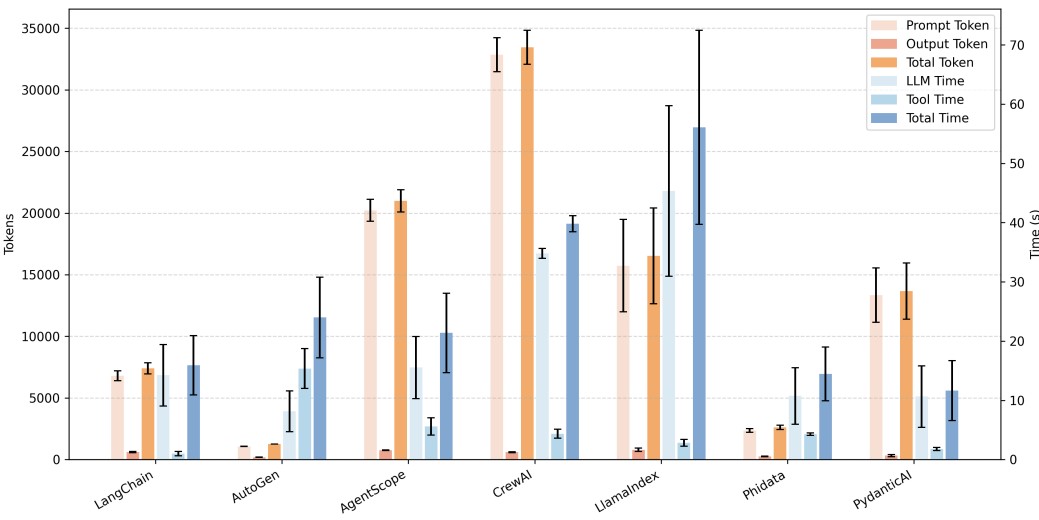

Figure 13: Consistency of Token Consumption and Latency in Repeated Experiments (GAIA)

*Insight 13: Experimental reproducibility is underpinned by the stability of token usage, while variability arises from stochastic tool behaviors and fluctuating LLM invocation dynamics.*

**Key Observations**    To verify the reliability and reproducibility of our results, we conduct repeated experiments on the HumanEval and GAIA datasets. The outcomes are reported in Table 8, 38, 39 for HumanEval and in Table 40, Table 41, Table 42 for GAIA. As illustrated by the error bars in Figure 12 and 13, the token consumption in our experiment is relatively stable. In general, the execution time is usually positively related to the token consumption.

**Underlying Mechanism-14: Stochastic Tool Behaviors**    Figure 13 indicates that the LlamaIndex framework yields a relatively high standard deviation on the GAIA dataset. This can be attributed to the stochastic nature of tool invocations and the consequent variations in the number of LLM invocation rounds.

**Underlying Mechanism-15: Fluctuating LLM invocation dynamics** The inherent randomness of certain LlamaIndex built-in toolssuch as the use of whisper in audio-visual modelsfurther amplifies this effect, resulting in a larger standard deviation in the GAIA test results.

Nevertheless, the overall trend remains reproducible.

In addition, to examine the impact of hardware differences, we rerun the GAIA benchmark on a machine equipped with a 40-series GPU with 48GB of memory. We then compare the results with the average values obtained from the RTX 3080 Ti setup, by computing the ratios of key metrics.

Table 44: Comparison of Token and Time Ratios Across Hardware Configurations

| Framework | Token ratio | | | Time ratio | | |
|---|---|---|---|---|---|---|
| | Prompt | Output | Total | LLM | Code executor | Total |
| LangChain | 1.080 | 1.060 | 1.083 | 0.639 | 0.713 | 0.638 |
| AutoGen | 0.998 | 1.062 | 1.008 | 0.501 | 1.124 | 0.923 |
| AgentScope | 0.996 | 1.108 | 1.000 | 0.975 | 0.543 | 0.858 |
| CrewAI | 1.034 | 0.984 | 1.033 | 1.044 | 1.217 | 1.076 |
| LlamaIndex | 1.354 | 1.401 | 1.356 | 1.014 | 1.093 | 1.032 |
| Phidata | 0.990 | 0.963 | 0.987 | 1.286 | 0.961 | 1.319 |
| PydanticAI | 0.912 | 0.915 | 0.912 | 0.629 | 1.164 | 0.622 |

As shown in Table 44, token usage remain largely consistent across most frameworks. Intuitively, the token consumption is independent of hardware setup. In terms of execution time, we observe significant speedup only for LangChain and Pydantic, indicating that these two frameworks benefit more from enhanced GPU capabilities, while others exhibit relatively stable performance regardless of GPU configuration.

## B.9 EFFECT OF DIFFERENT IMPLEMENTATIONS

*Insight 14: Different frameworks adopt distinct tool implementations and prompt designs, which can substantially impact efficiency.*

In our evaluation, whenever a framework does not support a required functionality, we implement the corresponding tool ourselves by adopting a popular tool. To isolate the impact of concrete tool implementations, we compare tools implemented by different frameworks against our own implementations on the same input dataset with 200 queries. As shown in Table 45, tool implementations vary substantially across frameworks: even for identical inputs, the same tool (e.g., XLSX reader, figure loader, code executor) can differ by more than an order of magnitude in runtime, depending on the frameworks internal design. This indicates that tool choice is not merely an engineering detail, but a key performance factor that can significantly affect the efficiency and responsiveness of multi-agent systems. Moreover, our implementations are typically the most lightweight, demonstrating that they introduce minimal overhead beyond the underlying operations.

Table 45: Tool Comparison

| Framework | PDF loader | CSV reader | XLSX reader | Text file reader | Doc reader | MP3 loader | Code executor |
|---|---|---|---|---|---|---|---|
| langchain | 0.04005 | 0.0178 | 0.90453 | 0.78435 | 0.00586 | 4.657798 | 0.002355 |
| autogen | x | x | x | x | x | x | 0.0000752 |
| agentscope | x | x | x | x | x | 5.9112 | 0.05335 |
| crewai | x | x | x | x | x | x | 0.0000752 |
| llamaindex | 0.011795 | 0.0562 | 0.34445 | 0.0010935 | 0.0026685 | 1.747 | 0.0001185 |
| phidata | x | 0.00086 | x | 0.0019865 | x | x | 0.000865 |
| pydantic | x | x | x | x | x | x | 0.0000752 |
| ours | 0.006545 | 0.00813 | 0.25137 | 0.0002695 | 0.01206 | 1.63566 | 0.0000752 |

We also analyze efficiency discrepancies between our independently implemented GAIA prompt and the framework-native prompts on successful cases, as summarized in Table 46. Since the Agentscope prompt is embedded and cannot be modified, the comparison is restricted to LangChain and LlamaIndex. The results show that our prompt can reduce both the total number of tokens and

the end-to-end latency by roughly 25%. This further highlights that prompt design is an important factor for improving the efficiency of multi-agent systems.

Table 46: GAIA Prompt Comparison

| Model | Prompt token | Output token | Total token | Total Time |
|---|---|---|---|---|
| llamaindex_our_prompt | 10835.26 | 412.84 | 11248.11 | 28.1029 |
| llamaindex | 10888.17433 | 582.111 | 11470.282 | 36.1454 |
| langchain_our_prompt | 3389.34 | 253.97 | 3543.31 | 8.333 |
| langchain | 3982.695 | 360.955 | 4343.645 | 12.54203333 |

## C   PROMPTS

### C.1   REACT

For frameworks that do not have a specific implementation of ReAct, we use the following prompt to build the ReAct workflow:

```
1  You are a ReAct-based assistant.
2  You analyze the question, decide whether to call a tool or directly
   answer, and then respond accordingly.
3  Use the following format:Question: the input question or request
4  Thought: you should always think about what to do\nAction: the action to
   take (if any)
5  Action Input: the input to the action (e.g., search query)
6  Observation: the result of the action
7  ... (this process can repeat multiple times)
8  Thought: I now know the final answer
9  Final Answer: the final answer to the original input question or request
10 Begin!
11 Question: {input}
```

#### C.1.1   LANGCHAIN

Within the ReAct workflow implemented via LangChain's AgentExecutor, we set the max_iterations parameter to 15 for experiments on the GAIA dataset and to 10 for those on the HumanEval dataset.

### C.2   RAG

For the following frameworks, we applied specific prompts to improve their token efficiency or to better align with the RAG workflow.

#### C.2.1   AUTOGEN

```
1  You are a helpful assistant. You can answer questions and provide
   information based on the context provided.
```

#### C.2.2   CREWAI

```
1  You are a specialized agent for RAG tasks.You just need to give the
   answer of the question. Don't need any othter word.Such as the answer is
   a number 5 ,you need output '5'.Or the answer is A,you need to output 'A'.
```

### C.2.3 PHIDATA

```
You are a RAG-based assistant. You analyze the question, and call the
search_knowledge_base tool to retrieve relevant documents from the
knowledge base, and then respond accordingly.
```

### C.2.4 PYDANTICAI

```
You're a RAG agent. please search information from the given task to
build a knowledge base and then retrieve relevant information from the
knowledge base.
```

## C.3 MOA

Unless otherwise specified, the following prompt is used for the aggregator agent.

### C.3.1 LANGCHAIN

```
You have been provided with a set of responses from various open-source
models to the latest user query. Your task is to synthesize these
responses into a single, high-quality response. It is crucial to
critically evaluate the information provided in these responses,
recognizing that some of it may be biased or incorrect. Your response
should not simply replicate the given answers but should offer a refined,
 accurate, and comprehensive reply to the instruction. Ensure your
response is well-structured, coherent, and adheres to the highest
standards of accuracy and reliability.
```

### C.3.2 AGENTSCOPE

```
You are an assistant called Dave,you should synthesize the answers from
Alice, Bob and Charles to arrive at the final response.
```

For the worker agent, we used the following prompt.

```
You are an assistant called Alice/Bob/Charles.
```

### C.3.3 CREWAI

```
You are an agent manager, and You need to assign the questions you
receive to each of your all agents, and summarize their answers to get a
more complete answer
You must give question to all the all agents, and you must summarize
their answers to get a more complete answer.\nYou need to be the best
```

For the worker agent, we used the following prompt.

```
You are one of the agents, you have to make your answers as perfect as
possible, there will be a management agent to choose the most perfect
answer among the three agents as output, you have to do your best to be
selected
```

### C.3.4 PHIDATA

```
Transfer task to all chat agents (There are 3 agents in your team)", "
Aggreagate responses from all chat agents
```

### C.3.5 PYDANTICAI

```
1  Your task is to aggregate all agents results to solve complex tasks.\nYou
    analyze the input, input the task to all tools that can run a single
   agent, and synthesize the results from all agents into a final response.
```

### C.4 GAIA

In this experiment, we used all levels of questions from the test subset of the GAIA dataset. Below are examples of prompts used in our system, depending on whether a file is attached:

```
1  question: A paper about AI regulation originally submitted to arXiv.org
   in June 2022 features a figure with three axes, each labeled with a pair
   of opposing terms. Which of these terms is used to describe a type of
   society in a Physics and Society article submitted to arXiv.org on August
    11, 2016?
```

```
1  question: The attached spreadsheet contains the inventory of a movie and
   video game rental store located in Seattle, Washington. What is the title
    of the oldest Blu-Ray listed in this spreadsheet? Return it exactly as
   it appears., file_name: 32102e3e-d12a-4209-9163-7b3a104efe5d.xlsx,
   file_path: path/to/32102e3e-d12a-4209-9163-7b3a104efe5d.xlsx
```

### C.5 HUMANEVAL

To avoid generating explanatory text or pseudo-code that hinders automated accuracy evaluation, we slightly modify the original HumanEval queries by adding minimal prompts.Below is an example of the prompt used for HumanEval problems:

```
1  from typing import List
2
3  def has_close_elements(numbers: List[float], threshold: float) -> bool:
4      """ Check if in given list of numbers, are any two numbers closer to
   each other than
5      given threshold.
6      >>> has_close_elements([1.0, 2.0, 3.0], 0.5)
7      False
8      >>> has_close_elements([1.0, 2.8, 3.0, 4.0, 5.0, 2.0], 0.3)
9      True
10     """
11
12 # Complete the function. Only return code. No explanation, no comments,
   no markdown.
```

### C.6 MMLU

For the MMLU dataset, we constructed the vector database used in the RAG workflow based on the development subset and evaluated the performance of each framework using the test subset. Given the large number of tasks in this dataset, we used only one-quarter of them in our experiments. Considering that tasks from the same domain tend to be spatially adjacent in the dataset, we selected one out of every four tasks in index order. This sampling strategy ensures broader domain coverage and maintains fairness in the evaluation.

Below is an example of the question in MMLU:

```
1  Question:Find the degree for the given field extension Q(sqrt(2), sqrt(3),
    sqrt(18)) over Q.
2  A.0
3  B.4
4  C.2
```

```
5   D.6
6   Answer with A, B, C, or D only
```

### C.7 ALPACAEVAL

In this experiment, we used the full set of tasks for the basic MoA experiments, and the first 100 tasks for extended experiments involving more agents. Below is an example of one such task.

## D BUGS AND FEATURES

This section summarizes the bugs or features of LLM agent frameworks that we discovered during our evaluation.

### D.1 LANGCHAIN

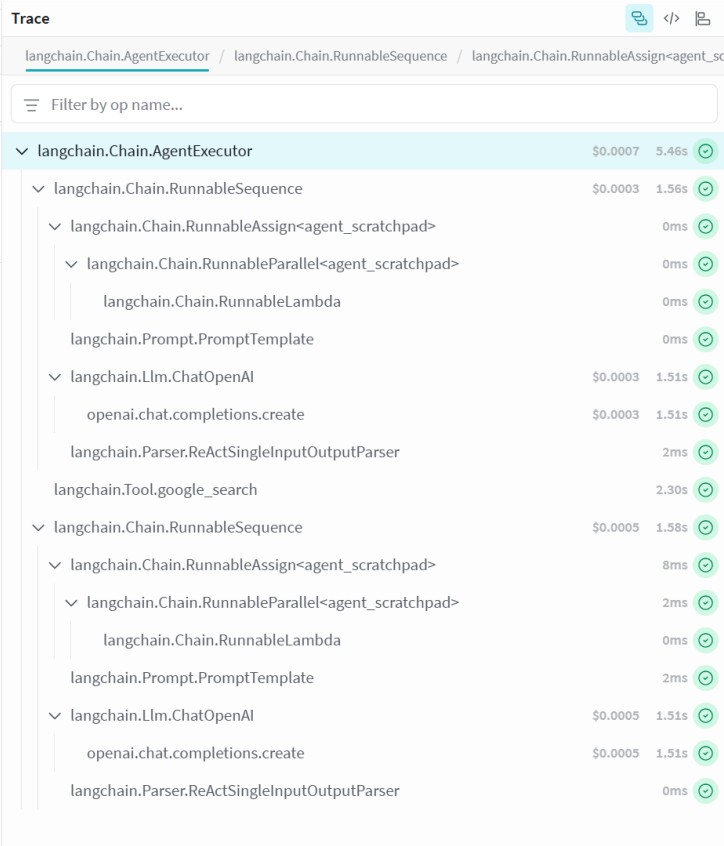

Figure 14: LangChain's high level of abstraction and encapsulation.

As shown in Figure 14, LangChain's high level of abstraction and encapsulation posed challenges in measuring specific metrics during our experiments.

2025-05-12 23:48:31.145 - root - INFO - tool name: pdf tool, tool time: 3.8433
2025-05-12 23:48:31.172 - root - INFO - omni run end, result: [Document(metadata={'producer': 'Project MUSE', 'creator': 'PyPDF', 'creationdate': '', 'author': 'David Greetham', 'language': 'English', 'title': 'Uncoupled: OR, How I Lost My Author(s)', 'xmp-dccreator': 'David Greetham', 'xmp-dc:language': 'English', 'xmp-dc:title': 'Uncoupled: OR, How I Lost My Author(s)', 'source': 'https://muse.jhu.edu/pub/3/article/240795/pdf', 'total_pages': 13},
page_content='Uncoupled : OR, How I Lost My Author(s) \nDavid Greetham\nTextual Cultures: Texts, Contexts, Interpretation, Volume 3, Number\n1, Spring 2008, pp. 44-55 (Article)\nPublished by Society for Textual Scholarship\nDOI:
\nFor additional information about this article\nhttps://doi.org/10.2979/tex.2008.3.1.44\nhttps://muse.jhu.edu/article/240795\n[103.224.173.84]   Project MUSE (2025-05-12 16:48 GMT)\n\x0cUncoupled\nOR, How I Lost My
Author(s)\nDavid Greetham\nAbstract\nThe essay traces the personal history of trying to "fi nd" an author, in the sense of a number \nof attempts to identify a "father figure" with whom an editori al and critical career might be
\nlinked. Describing such devices as the mapping out of a comprehensive series of abstract mod-\nels for charting the relationships between a translated and a translating text (in the case of \n\John Trevisa) and the construction of an
authorial i diolect where no autograph survives (in \nthe case of Thomas Hoccleve), the personal narrative now reg ards these attempts to reach \n\back into the authorial psyche as hubristic, even improper. In the place of such a single
autho-\nrial identity, the essay concludes by showing how authorial ity rather than authorship took \n\over in the development of a scholarly career, resulting in, for example, the founding of the in-\nterdisciplinary Society for Textual
Scholarship.\n For Speed Hill1\n I T R I E D T O F I N D, A N D B E L O Y A L T O, A N A U T H O R: I R E A L L Y D I D. But somehow \nthe "psychic connections" that my fellow-panelists identified, never quite \ntook with me. As a very
neophyte editor, I was co-o pted (suborned?) into \nthe Clarendon Press edition of John Trevisa' s Middle English translation of \nBartholomaeus Anglicus' s De Proprietatibus Rerum ( On the Properties of \nThings), or DPR, a
"monumental" tome, published in 1975. 2 But Trevisa, \nwhile a diligent scholar and even possibly a renegade Wycliffi te (or even \n 1. Since Speed Hill (1935–2007) organised these two panels on the psychic relations \nbetween editor
and author (and also contributed a paper), but was sadly unable to \nattend the sessions, I dedicate my own paper to him.\n 2. Se y m o u r 1975–1989. My textual contribution, Liber Quintus Decimus. De Prouin-\nciis, occurs in vol. 2, pp.
726–824, and I also worked on the notes for vol. 3.\n[103.224.173.84]   Project MUSE (2025-05-12 16:48 GMT)\n\x0cDavid Greetham: Uncoupled | 45\npossibly, according to David C. Fowler,3 the author of Piers Plowman), re-\nmained
somehow determinedly faceless, basically a faithful translator (and a \nsuccessful one) of "other men' s flowers" .\nBut I did try, and, to be honest, quite enjoyed the ultimately f ailed at-\ntempt, in part because it gave me the
opportunity to delve into some of the \nmore arcane areas of textual research (based on this biographic al fact of \nTrevisa' s having devoted himself to translation), and thereby to produce \nwhat I now regard as a series of too-neat,
too pat, and perhaps also too pre-\ntentious "models" for translation, which Fredson Bowers was generous \nenough to publish in Studies in Bibliography (see G r e e t h a m 1984). I don' t \nthink there' s any point now (and besides
which, I would probably find it too \nembarrassing a display of my youthful idealism) to try to actua lly explicate \nwhat these fanciful models mean, ranging as they do from what I called the \n "Perfect Linear" (in which all witnesses in
the translating language are uni-\nform in using the same, or very similar, term[s] to represent what appears to \nbe an "accurate" translation) to the more complex "Parallel Variance Class" \n\n(in which we seem to get two or more
plausible but inconsistent translations \nof the original text). I now believe that the very titles of thes e different \ntranslation models sound pretentious, in their attempts to map out every \npossible relationship between translating and
translated text: I think they \nshow my youthful earnestness, my attempts to situate Trevisa and his text as \nan ideal form to which I could declare allegiance, and thus embrace as a fa-\nther figure.4 The models were quite wonderful in
their enthusiasm, and were \nall dedicated to the project of gaining a more direct access to my author, ob-\nb 3. While I had obviously "repressed" this recognition in the oral delivery of this paper, \nthe recent death of David Fowler has
brought home to me more forcefully his role as \nthe missing "father figure" looming in this and other papers in this collection. It was \nFowler who read my first meagre publication in a medievalist newsletter mysteri-\nously called
Ralph, and on the strength of that and subsequent correspondence, in-\nvited me out to the University of Washington and other west coast institutions to \nlecture on Trevisa and textual studies. He became one of the first advisers to the
So-\nciety for Textual Scholarship, a contributor to TE X T and to our conferences, and a \ncontinued kindly presence in my early textual life. While this paper is formally ded-\nicated to Speed Hill, it also commemorates a generation of
such scholars (like Fred-\nson Bowers and Paul Oskar Kristeller) who showed me the textual light before I \nnever acknowledged that was the path I was to take, and continued to be wise, cour-\nnteous, and benevolent mentors till the end
of their lives.\n 4. The models were intended to chart every possible relatio nship between the trans-\nlated and translating text, from unambiguous to complex, including various types \nthat would cover not only variance in the witnesses
of the tra nslating text and \nunanimity in the source, but also variance in both source and translation.\n\x0c46 | Textual Cultures 3.1 (2008)\nscured not by a "veil of print" but by the veil of scribal confusion and mis-\ntransmission.
Our/my author lay beyond this confusio n, immanent as the \nlodestar for my arcane models. I think I have to recognise that the project \nwas ultimately not as effective at "unveiling" the author as I had hoped and \nwas thus also not
enough to endow Trevisa with those paternal qualities I \nwas presumably looking for.\nBut, what was worse, from the mid-seventies or so, and especia lly after \nthe first two volumes came out in 1975, I began to have second thoughts
\nabout the then-dominant editorial principles which we had unconsciously \nassumed were appropriate to the editing of Trevisa, and began to write a se-\nvries of essays not setting up an idealised author and an idealis ed text, but
\ndistancing myself both from these essentially Formalist/New Critical princi-\nnples and at the same time questioning the unexamined hegemony of any ed-\nvitorial/authorial/philosophical protocols that could be emp loyed on any \ntext.
This distancing (and embarrassment) has continued down to the pres-\nnent, for in an article recently published in the Italian journal Ecdot-\nica, I return again to the scene of the crime and use the personal and cultural \ndistancing
as a means of showing just what was then, and what might be \nnow: that the earlier search for an immanent author (mis)represented by the \nextant, but inevitably corrupt, witnesses, has given way to a recognition \nthat the variance
shown in these witnesses is valuable evidence for the so-\ncialization of the text ( G r e e t h a m 2006). In this case, my Ecdotica piece, \nwhile formally a response to Paul Eggert' s generous and comprehensive re-\nview-essay (2005) of

Figure 15: LangChain occasionally terminated processes prematurely.

Additionally, LangChain occasionally terminated processes prematurely after reading files from the GAIA dataset, returning the file content directly rather than proceeding with the expected operations (see Figure 15).

## D.2 AUTOGEN

Due to the default system prompt being relatively long and containing irrelevant instructions, the RAG workflow may consume unnecessary tokens or produce unexpected errors (e.g., attempting to invoke non-existent tools). Therefore, it is necessary for users to customize the system prompt.

## D.3 AGENTSCOPE

AgentScopes image and audio processing tools internally rely on OpenAI models, causing their execution time to partially overlap with that of the LLM itself. This overlap can lead to inflated or inaccurate measurements of LLM processing time. Researchers and practitioners should be mindful of this issue when conducting time-based evaluations involving AgentScope.

```python
def openai_image_to_text(
    image_urls: Union[str, list[str]],
    api_key: str,
    prompt: str = "Describe the image",
    model: Literal["gpt-4o", "gpt-4-turbo"] = "gpt-4o",
) -> ServiceResponse:
    """
    Generate descriptive text for given image(s) using a specified model, and
    return the generated text.

    Args:
        image_urls (`Union[str, list[str]]`):
            The URL or list of URLs pointing to the images that need to be
            described.
        api_key (`str`):
            The API key for the OpenAI API.
        prompt (`str`, defaults to `"Describe the image"`):
            The prompt that instructs the model on how to describe
            the image(s).
        model (`Literal["gpt-4o", "gpt-4-turbo"]`, defaults to `"gpt-4o"`):
            The model to use for generating the text descriptions.

    Returns:
        `ServiceResponse`:
```

```
                     A dictionary with two variables: `status` and `content`.
                     If `status` is `ServiceExecStatus.SUCCESS`,
                     the `content` contains the generated text description(s).

    Example:

            .. code-block:: python

                image_url = "https://example.com/image.jpg"
                api_key = "YOUR_API_KEY"
                print(openai_image_to_text(image_url, api_key))

            > {
            >     'status': 'SUCCESS',
            >     'content': "A detailed description of the image..."
            > }
    """
    openai_chat_wrapper = OpenAIChatWrapper(
        config_name="image_to_text_service_call",
        model_name=model,
        api_key=api_key,
    )
    messages = Msg(
        name="service_call",
        role="user",
        content=prompt,
        url=image_urls,
    )
    openai_messages = openai_chat_wrapper.format(messages)
    try:
        response = openai_chat_wrapper(openai_messages)
        return ServiceResponse(ServiceExecStatus.SUCCESS, response.text)
    except Exception as e:
        return ServiceResponse(ServiceExecStatus.ERROR, str(e))

def openai_audio_to_text(
    audio_file_url: str,
    api_key: str,
    language: str = "en",
    temperature: float = 0.2,
) -> ServiceResponse:
    """
    Convert an audio file to text using OpenAI's transcription service.

    Args:
        audio_file_url (`str`):
            The file path or URL to the audio file that needs to be
            transcribed.
        api_key (`str`):
            The API key for the OpenAI API.
        language (`str`, defaults to `"en"`):
            The language of the input audio. Supplying the input language
 in
            [ISO-639-1](https://en.wikipedia.org/wiki/List_of_ISO_639-1
_codes)
            format will improve accuracy and latency.
        temperature (`float`, defaults to `0.2`):
            The temperature for the transcription, which affects the
            randomness of the output.

    Returns:
        `ServiceResponse`:
            A dictionary with two variables: `status` and `content`.
            If `status` is `ServiceExecStatus.SUCCESS`,
```

```
87              the `content` contains a dictionary with key 'transcription'
and
88              value as the transcribed text.
89
90      Example:
91
92          .. code-block:: python
93
94              audio_file_url = "/path/to/audio.mp3"
95              api_key = "YOUR_API_KEY"
96              print(openai_audio_to_text(audio_file_url, api_key))
97
98          > {
99          >     'status': 'SUCCESS',
100         >     'content': {'transcription': 'This is the transcribed text
from
101         the audio file.'}
102         > }
103     """
104     try:
105         import openai
106     except ImportError as e:
107         raise ImportError(
108             "The `openai` library is not installed. Please install it by
"
109             "running `pip install openai`.",
110         ) from e
111
112     client = openai.OpenAI(api_key=api_key)
113     audio_file_url = os.path.abspath(audio_file_url)
114     with open(audio_file_url, "rb") as audio_file:
115         try:
116             transcription = client.audio.transcriptions.create(
117                 model="whisper-1",
118                 file=audio_file,
119                 language=language,
120                 temperature=temperature,
121             )
122             return ServiceResponse(
123                 ServiceExecStatus.SUCCESS,
124                 {"transcription": transcription.text},
125             )
126         except Exception as e:
127             return ServiceResponse(
128                 ServiceExecStatus.ERROR,
129                 f"Error: Failed to transcribe audio {str(e)}",
130 )
```

Meanwhile, AgentScope's vector database module, LlamaIndexKnowledge, is implemented based on the BM25Retriever from the llamaindex library. However, the original implementation relies on an outdated version of llamaindex, and recent updates to the library introduced structural changes that break compatibility with the original import statements.

```
1  from llama_index.retrievers.bm25 import BM25Retriever
```

To ensure a consistent environment without modifying the frameworks built-in vector database logic, we resolved the issue by duplicating the LlamaIndexKnowledge module and updating the import paths to match the newer llamaindex version.

```
1  from llama_index.legacy.retrievers.bm25_retriever import BM25Retriever
```

> What integer-rounded percentage of the total length of the harlequin shrimp recorded in Omar Valencfia-Mendez 2017 paper was the sea star fed to the same type of shrimp in G. Curt Fiedler's 2002 paper?

**Response**

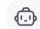 Thought: The question requires specific data from two scientific papers: Omar Valencia-Mendez's 2017 paper and G. Curt Fiedler's 2002 paper. I need to find the total length of the harlequin shrimp recorded in the 2017 paper and the size of the sea star fed to the shrimp in the 2002 paper. Then, I can calculate the percentage and round it to the nearest integer.

Action: google_search
Action Input: "Omar Valencia-Mendez 2017 harlequin shrimp total length"

Figure 16: LlamaIndex frequently fails to invoke tools correctly.

## D.4 CREWAI

When our MOA invokes a large number of agents (>=12), CrewAI system occasionally fails to call all agents completely during execution as intended. For example, when we request 12 sub-agents to be activated, some queries may only trigger 9 or fewer agents.

## D.5 LLAMAINDEX

As shown in Figure 16, LlamaIndex frequently fails to invoke tools correctly, primarily due to the lack of prompt constraints and insufficient post-processing checks on LLM outputs. Without explicit guidance and validation mechanisms, the LLM often produces outputs that do not conform to the expected dictionary format, resulting in tool invocation failures.

## D.6 PHIDATA

As shown in Figure 17, in the ReAct workflow, Phidata passes the available tools to the LLM via the "tools" field. Unlike Llamaindex, which emphasizes the functionality and usage of tools in the system prompt, Phidata rarely invokes the code execution tool when processing queries from humaneval.

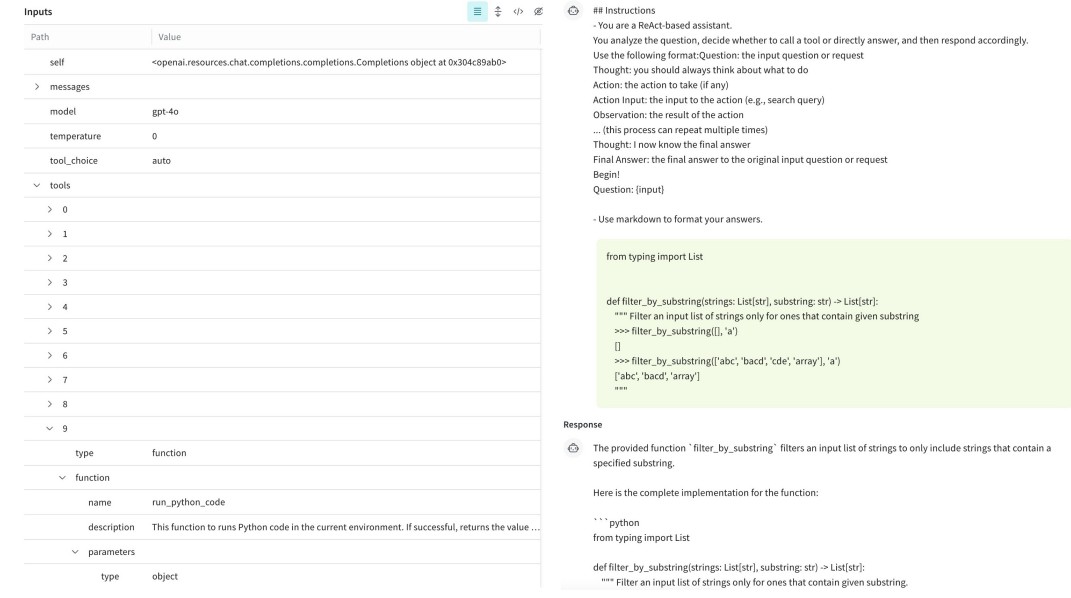

Figure 17: Phidata passes the available tools to the LLM via the "tools" field.

### D.7 PYDANTICAI

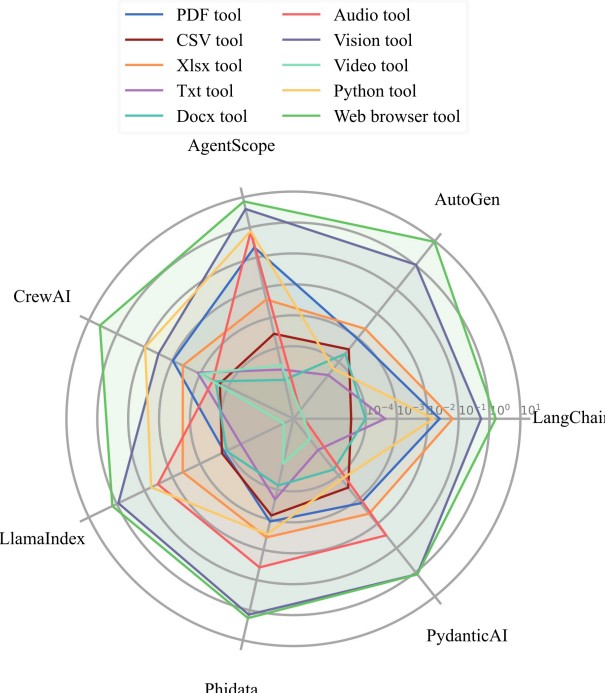

Figure 18: Visualization of the average execution time per run of different tools across different frameworks.

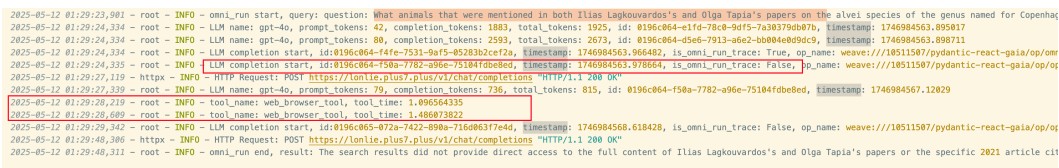

Figure 19: PydanticAI's simultaneous invocations of the same tool.

By further visualizing the experimental data (see Figure 18), we found that within the PydanticAI ReAct framework, the same tool was often invoked simultaneously multiple times, potentially leading to inefficiencies. Additionally, similar to Phidata, the code execution tool was seldom triggered (see Figure 19).

Furthermore, The MoA implementation in the PydanticAI framework is tool-based, and not all three models are invoked for every query. We observe that when the number of sub-agents is 3, 6, 9, 12, and 15, there were 232, 89, 229, 485, and 663 instances, respectively, where sub-agents were not invoked. These skipped invocations are randomly distributed across different queries, resulting in lower token consumption than expected.

## E TOOL IMPLEMENTATION

The implementation details of all tools are presented below. Tables 47–50 list all the tools we use and provide explanations for the additional tools we develop.

Table 47: Tool Implementations in LangChain

| Functionality | LangChain |
|---|---|
| Search | langchain_google_community.GoogleSearchAPIWrapper |
| PDF loader | langchain_community.document_loadersy.PDFLoader |
| CSV reader | langchain_community.document_loaders.csv_loader |
| XLSX reader | langchain_community.document_loadersUnstructuredExcelLoader |
| Text file reader | langchain_community.document_loaders.UnstructuredFileLoader |
| doc reader | langchain_community.document_loadersy.Docx2txtLoader |
| MP3 loader | langchain_community.document_loaders.assemblyai.AssemblyAIAudioTranscriptLoader |
| Figure loader | transformers.VisionEncoderDecoderModel() |
| Video loader | AudioSegment.from_file |
| Code executor | langchain_experimental.tools.PythonREPLTool |

Table 48: Tool Implementations in AutoGen and AgentScope

| Functionality | AutoGen | AgentScope |
|---|---|---|
| Search | requests + BeautifulSoup | agentscope.service.google_search |
| PDF loader | PyPDF2.PdfReader | PyPDF2.PdfReader |
| CSV reader | pd.read_csv | pd.read_csv |
| XLSX reader | pd.read_excel | pd.read_excel |
| Text file reader | open(file).read | open(file).read |
| doc reader | docx.Document | docx.Document |
| MP3 loader | whisper.load_model | agentscope.service.openai_audio_to_text |
| Figure loader | transformers.VisionEncoderDecoderModel | agentscope.service.openai_image_to_text |
| Video loader | AudioSegment.from_file | AudioSegment.from_file |
| Code executor | autogen.coding.LocalCommandLineCodeExecutor | agentscope.service.execute_python_code |

Table 49: Tool Implementations in CrewAI and LlamaIndex

| Functionality | CrewAI | LlamaIndex |
|---|---|---|
| Search | crewai_tools.SerperDevTool | llama_index.tools.google.GoogleSearchToolSpec |
| PDF loader | PyPDF2.PdfReader | llama_index.readers.file.PDFReader |
| CSV reader | PyPDF2.PdfReader | PandasCSVReader |
| XLSX reader | pd.read_excel | PandasExcelReader |
| Text file reader | open(file).read | llama_index.core.SimpleDirectoryReader |
| doc reader | docx.Document | llama_index.core.SimpleDirectoryReader |
| MP3 loader | whisper.load_model | VideoAudioReader |
| Figure loader | crewai_tools.VisionTool | ImageReader |
| Video loader | AudioSegment.from_file | VideoAudioReader |
| Code executor | crewai_tools.CodeInterpreterTool | lama_index.tools.code_interpreter.CodeInterpreterToolSpec |

For frameworks that do not include the required tools, we adopted a unified implementation as follows.

## E.1 SEARCH

### E.1.1 AUTOGGEN

```
def google_search(query: str, num_results: int = 2, max_chars: int = 500)
 -> list:  # type: ignore[type-arg]
    import os
    import time
    import requests
```

Table 50: Tool Implementations in Phidata and PydanticAI

| Functionality | Phidata | PydanticAI |
|---|---|---|
| Search | phi.tools.googlesearch.GoogleSearch | requests |
| PDF loader | PyPDF2.PdfReader | PyPDF2.PdfReader |
| CSV reader | phi.tools.csv_tools.CsvTools | PyPDF2.PdfReader |
| XLSX reader | pd.read_excel | pd.read_excel |
| Text file reader | phi.tools.file.FileTools | open(file).read |
| doc reader | docx.Document | docx.Document |
| MP3 loader | whisper.load_model | whisper.load_model |
| Figure loader | transformers.VisionEncoderDecoderModel() | transformers.VisionEncoderDecoderModel() |
| Video loader | AudioSegment.from_file | AudioSegment.from_file |
| Code executor | phi.tools.python.PythonTools | langchain_experimental.utilities.python.PythonREPL |

```python
from bs4 import BeautifulSoup
from dotenv import load_dotenv
load_dotenv()
google_api_key = os.environ['GOOGLE_KEY']
search_engine_id = os.environ['GOOGLE_ENGINE']
if not search_engine_id or not search_engine_id:
    raise ValueError("API key or Search Engine ID not found")
url = "https://www.googleapis.com/customsearch/v1"
params = {
    "key": google_api_key,
    "cx": search_engine_id,
    "q": query,
    "num": num_results
}
response = requests.get(url, params=params)  # type: ignore[arg-type]
if response.status_code != 200:
    print(response.json())
    raise Exception(f"Error in API request: {response.status_code}")
results = response.json().get("items", [])
def get_page_content(url: str) -> str:
    try:
        response = requests.get(url, timeout=10)
        soup = BeautifulSoup(response.content, "html.parser")
        text = soup.get_text(separator=" ", strip=True)
        words = text.split()
        content = ""
        for word in words:
            if len(content) + len(word) + 1 > max_chars:
                break
            content += " " + word
        return content.strip()
    except Exception as e:
        print(f"Error fetching {url}: {str(e)}")
        return ""
enriched_results = []
for item in results:
    body = get_page_content(item["link"])
    enriched_results.append(
        {
            "title": item["title"],
            "link": item["link"],
            "snippet": item["snippet"],
            "body": body
        }
    )
return enriched_results
```

### E.1.2   PYDANTICAI

```python
def google_search(query, num=None):
    """
    Make a query to the Google search engine to receive a list of results.

    Args:
        query (str): The query to be passed to Google search.
        num (int, optional): The number of search results to return.
Defaults to None.

    Returns:
        str: The JSON response from the Google search API.

    Raises:
        ValueError: If the 'num' is not an integer between 1 and 10.
    """
    try:
        QUERY_URL_TMPL = ("https://www.googleapis.com/customsearch/v1?key
={key}&cx={engine}&q={query}")
        url = QUERY_URL_TMPL.format(
            key=os.environ['GOOGLE_KEY'],
            engine=os.environ['GOOGLE_ENGINE'],
            query=urllib.parse.quote_plus(str(query))
        )
        if num is not None:
            if not 1 <= num <= 10:
                raise ValueError("num should be an integer between 1 and
10, inclusive")
            url += f"&num={num}"
        response = requests.get(url)
        return response.text
    except Exception as e:
        return f"Error: {e}"
```

### E.2   PDF LOADER

```python
def pdf_load(file_path: str) -> ServiceResponse:
    try:
        reader = PdfReader(file_path)
        text = ""
        for page in reader.pages:
            text += page.extract_text() + "\n"
        return ServiceResponse(status=ServiceExecStatus.SUCCESS,content=
text)
    except Exception as e:
        return ServiceResponse(ServiceExecStatus.ERROR, str(e))
```

### E.3   CSV READER

```python
import pandas as pd

def csv_load(path:str)->ServiceResponse:
    try:
        df = pd.read_csv(path)
        csv_str = df.to_string(index=False)
        return ServiceResponse(status=ServiceExecStatus.SUCCESS,content=
csv_str)
    except Exception as e:
        return ServiceResponse(ServiceExecStatus.ERROR, str(e))
```

### E.4 XLSX READER

```python
def xlsx_load(path:str)->ServiceResponse:
    try:
        excel_file = pd.read_excel(path, sheet_name=None)
        result = ""
        for sheet_name, df in excel_file.items():
            result += f"Sheet: {sheet_name}\n"
            result += df.to_string(index=False) + "\n\n"
        return ServiceResponse(status=ServiceExecStatus.SUCCESS,content=
result.strip())
    except Exception as e:
        return ServiceResponse(ServiceExecStatus.ERROR, str(e))
```

### E.5 TEXT FILE READER

```python
import pandas as pd

def txt_load(path:str)->ServiceResponse:
    try:
        with open(path, 'r', encoding='utf-8') as f:
            txt_str = f.read()
        return ServiceResponse(status=ServiceExecStatus.SUCCESS,content=
txt_str)
    except Exception as e:
        return ServiceResponse(ServiceExecStatus.ERROR, str(e))
```

### E.6 DOCX READER

```python
from docx import Document

def docs_load(path:str)->ServiceResponse:
    try:
        doc = Document(path)
        docx_str = "\n".join([para.text for para in doc.paragraphs])
        return ServiceResponse(status=ServiceExecStatus.SUCCESS,content=
docx_str)
    except Exception as e:
        return ServiceResponse(ServiceExecStatus.ERROR, str(e))
```

### E.7 MP3 LOADER

```python
import whisper
from typing import cast

def load_audio(file):
    model = whisper.load_model(name="base")
    model = cast(whisper.Whisper, model)
    result = model.transcribe(str(file))
    return result["text"]
```

### E.8 FIGURE LOADER

```python
from transformers import DonutProcessor, VisionEncoderDecoderModel
import re
from PIL import Image

def load_image(path):
    image = Image.open(path)
```

```
 7      processor = DonutProcessor.from_pretrained(
 8                          "naver-clova-ix/donut-base-finetuned-cord-v2"
 9                      )
10      model = VisionEncoderDecoderModel.from_pretrained(
11          "naver-clova-ix/donut-base-finetuned-cord-v2"
12      )
13      device = 'cpu'
14      model.to(device)
15      # prepare decoder inputs
16      task_prompt = "<s_cord-v2>"
17      decoder_input_ids = processor.tokenizer(
18          task_prompt, add_special_tokens=False, return_tensors="pt"
19      ).input_ids
20      pixel_values = processor(image, return_tensors="pt").pixel_values
21      outputs = model.generate(
22          pixel_values.to(device),
23          decoder_input_ids=decoder_input_ids.to(device),
24          max_length=model.decoder.config.max_position_embeddings,
25          early_stopping=True,
26          pad_token_id=processor.tokenizer.pad_token_id,
27          eos_token_id=processor.tokenizer.eos_token_id,
28          use_cache=True,
29          num_beams=3,
30          bad_words_ids=[[processor.tokenizer.unk_token_id]],
31          return_dict_in_generate=True,
32      )
33      sequence = processor.batch_decode(outputs.sequences)[0]
34      sequence = sequence.replace(processor.tokenizer.eos_token, "").replace(
35          processor.tokenizer.pad_token, ""
36      )
37      # remove first task start token
38      text_str = re.sub(r"<.*?>", "", sequence, count=1).strip()
39      return text_str
```

### E.9 VIDEO LOADER

```
 1  import whisper
 2  from typing import cast
 3  from pydub import AudioSegment
 4  from pathlib import Path
 5
 6  def load_video(file):
 7      video = AudioSegment.from_file(Path(file), format=file[-3:])
 8      audio = video.split_to_mono()[0]
 9      file_str = str(file)[:-4] + ".mp3"
10      audio.export(file_str, format="mp3")
11      model = whisper.load_model(name="base")
12      model = cast(whisper.Whisper, model)
13      result = model.transcribe(str(file))
14      return result["text"]
```

### E.10 DATA RETRIEVAL

```
 1  def create_vector_db():
 2      import faiss
 3      import pickle
 4      from sentence_transformers import SentenceTransformer
 5      from data.mmlu import merge_csv_files_in_folder
 6      dataset=merge_csv_files_in_folder(path to MMLU/dev)
 7      docs = []
 8      for item in dataset:
```

```
9            text = item[0].replace(",please answer A,B,C,or D.",",")+f"
    answer:{item[1]}."
10           docs.append(text)
11      embed_model = SentenceTransformer('all-MiniLM-L6-v2')
12      doc_embeddings = embed_model.encode(docs)
13      dimension = doc_embeddings.shape[1]
14      index = faiss.IndexFlatL2(dimension)
15      index.add(doc_embeddings)
16      faiss.write_index(index, "db/index.faiss")
17      with open("db/index.pkl", "wb") as f:
18          pickle.dump(docs, f)
19
20  def load_vector_db():
21      import faiss
22      import pickle
23      from sentence_transformers import SentenceTransformer
24      class db:
25          def __init__(self):
26              self.index = faiss.read_index("db/index.faiss")
27              with open("db/index.pkl", "rb") as f:
28                  self.docs = pickle.load(f)
29              self.embed_model = SentenceTransformer('all-MiniLM-L6-v2')
30          def search(self, query, k=5):
31              query_embedding = self.embed_model.encode([query])
32              D, I = self.index.search(query_embedding, k)
33              return [self.docs[i] for i in I[0]]
34      return db()
```

## E.11 SCIENCEWORLD ENVIRONMENT INTERFACE

```
1   from typing import List
2   from sentence_transformers import SentenceTransformer
3   import numpy as np
4   embedding_model = SentenceTransformer("sentence-transformers/all-MiniLM-
    L6-v2")
5   _cached_actions: List[str] = []
6   _cached_embeddings: np.ndarray = None
7
8   def observe() -> str:
9       obs = env.look()
10      return obs
11
12  def valid_action(action: str) -> str:
13      global _cached_actions, _cached_embeddings, embedding_model
14      new_actions = env.get_valid_action_object_combinations()
15      if new_actions != _cached_actions or _cached_embeddings is None:
16          _cached_actions = new_actions
17          _cached_embeddings = embedding_model.encode(new_actions,
    convert_to_numpy=True)
18      if action in _cached_actions:
19          return action
20      else:
21          intent_emb = embedding_model.encode([action], convert_to_numpy=
    True)[0]
22          sims = np.dot(_cached_embeddings, intent_emb) / (
23          np.linalg.norm(_cached_embeddings, axis=1) * np.linalg.norm(
    intent_emb)
24          + 1e-8
25          )
26          best_idx = int(np.argmax(sims))
27          final_action = _cached_actions[best_idx]
28          print(f"[Grounded] '{action}'  '{final_action}'")
29          return final_action
```

```python
def exe_action(action: str) -> str:
    action=valid_action(action)
    obs, reward, done, info = env.step(action)
    return obs
```

## E.12 PROBLEM SOLVER

```python
def twoSum(nums: List[int], target: int) -> List[int]:
    """
    Given an array of integers nums and an integer target, return indices
 of the two numbers such that they add up to target.
    Args:
        nums (List): an array of integers
        target (Int): an integer target
    Returns:
        List[int]: indices of the two numbers such that they add up to
target.
    """
    try:
        n = len(nums)
        for i in range(n):
            for j in range(i + 1, n):
                if nums[i] + nums[j] == target:
                    return [i, j]

        return []
    except Exception as e:
        return str(e)

def lengthOfLongestSubstring(s: str) -> int:
    """
    Given a string s, find the length of the longest substring without
duplicate characters.
    Arg:
        s (String): a string

    Returns:
        Int: the length of the longest substring without duplicate
characters.
    """
    try:
        left = 0
        right = 0
        max_len = 0

        while right < len(s):
            if s[right] in s[left:right]:
                max_len = max(max_len, right-left)
                left = s.index(s[right], left, right)+1
            max_len = max(max_len, right-left+1)
            right += 1
        return max_len
    except Exception as e:
        return str(e)

def findMedianSortedArrays(nums1: List[int], nums2: List[int]) -> float:
    """
    Given two sorted arrays nums1 and nums2 of size m and n respectively,
 return the median of the two sorted arrays.
    Args:
```

```
51         nums1 (List[int]): sorted array 1
52         nums2 (List[int]): sorted array 2
53     Returns:
54         float: the median of the two sorted arrays
55     """
56     try:
57         m, n = len(nums1), len(nums2)
58
59         def kth_small(k):
60             i = j = 0
61             while True:
62                 if i == m:
63                     return nums2[j + k - 1]
64                 if j == n:
65                     return nums1[i + k - 1]
66                 if k == 1:
67                     return min(nums1[i], nums2[j])
68                 pivot_i = min(i + (k >> 1) - 1, m - 1)
69                 pivot_j = min(j + (k >> 1) - 1, n - 1)
70                 if nums1[pivot_i] < nums2[pivot_j]:
71                     k -= pivot_i + 1 - i
72                     i = pivot_i + 1
73                 else:
74                     k -= pivot_j + 1 - j
75                     j = pivot_j + 1
76
77         return (
78             kth_small((m + n + 1 >> 1))
79             if m + n & 1
80             else (kth_small((m + n >> 1) + 1) + kth_small((m + n >> 1)))
81             * 0.5
82         )
83     except Exception as e:
84         return str(e)
85
86 ...
```

# F   USAGE OF LARGE LANGUAGE MODELS

In the preparation of this paper, we employed large language models to assist with language refinement and stylistic improvements. Typical prompts included instructions such as "please polish the following academic text while preserving its technical meaning", "improve clarity and conciseness without altering the content", or "translate the following text into fluent academic English."

The LLMs were not used for generating research ideas, designing experiments, conducting analyses, or interpreting results. All technical content, methodology, and conclusions are the sole work of the authors, who take full responsibility for the accuracy and validity of the presented material.

