# OpenReview forum: "AgentRace: Benchmarking Efficiency in LLM Agent Frameworks"
_ICLR.cc/2026/Conference — Submitted to ICLR 2026_

### Official Review · Reviewer_cW3P · 2025-10-30

**Soundness:** 2
**Presentation:** 2
**Contribution:** 1
**Rating:** 2
**Confidence:** 4

**Summary:**

The paper presents AgentRace, a benchmark-style evaluation framework for assessing the efficiency (time, token count, communication overhead) of various large-language-model based agent systems. It aggregates several existing agent frameworks and datasets, runs empirical experiments, and reports a set of insights about where inefficiencies arise in agent pipelines (e.g., prompt length growth, tool orchestration latency, communication bottlenecks). The benchmark can serve the community as a standard way to compare agent frameworks on efficiency.

**Strengths:**

- The authors bring together multiple agent frameworks and datasets, and set up a unified testing environment. This kind of integration is non-trivial and may be valuable as a reproducibility/system contribution.

- Focusing on efficiency (rather than just task performance or accuracy) is timely: as agent systems move toward production and deployment, latency, cost (token usage), and orchestration overhead become important.

**Weaknesses:**

- Limited methodological or conceptual novelty. The work does not propose a new algorithm, novel evaluation metric, or analytic model of efficiency. Instead it aggregates existing datasets and frameworks, and measures commonly used metrics (time, tokens, communication size). As such, the contribution is more “engineering benchmark” than “scientific advance.”

- Overlap with existing observability/agent-profiling tools. In industry there are many tools and platforms that provide observability and profiling for LLM-agent systems: for example, platforms tracking latency, token cost, tool call breakdowns, and more. (OpenTelemetry, LangFuse, datadog etc) This raises the question as what does this paper’s benchmark provide beyond what these existing tools or frameworks already do?

- The paper does not appear to consider system-level optimisation concerns, e.g., caching of tokens, optimized/common toolset for fair comparison etc

- The metrics measured (runtime, token count, communication size) are conventional, and many of the conclusions drawn (e.g., “longer prompts lead to greater cost”, “tool orchestration adds latency”) are intuitive or not surprising. Thus the contribution of “new insight” is limited.

**Questions:**

- You mention measuring efficiency across frameworks — can you clarify how your benchmark differs from or improves upon existing observability or profiling platforms (e.g., those tracking LLM agent behaviour, token cost, tool calls) in industry or research?

- How do you ensure fairness across agent frameworks in your experiments (for example tool implementations, prompt variants)? Could differences in configuration dominate your reported differences?

---

> ### Author Response · Authors · 2025-11-26
>
> Dear Reviewer cW3P:
>
> We sincerely thank you for the valuable feedback and helpful suggestions. Below, we address each concern point-by-point.
>
>
> > Q1. Limited methodological or conceptual novelty. The work does not propose a new algorithm, novel evaluation metric, or analytic model of efficiency. Instead it aggregates existing datasets and frameworks, and measures commonly used metrics (time, tokens, communication size). As such, the contribution is more “engineering benchmark” than “scientific advance.”
>
> A1: We understand the concern regarding novelty. However, we believe that designing an easy-to-use benchmark platform, conducting a comprehensive assessment, and providing insights are also important contributions. For example, DecodingTrust [1] received NeurIPS 2023 Outstanding Datasets and Benchmarks Paper Award, which conducted a comprehensive assessment of trustworthiness in GPT models. We focus on surfacing system-level insights rather than proposing new datasets or metrics. We respectfully argue that **a systematic and reproducible assessment of existing frameworks** under a unified and extensible benchmark is itself a valuable contribution—particularly in the rapidly evolving domain of LLM agents.
>
> [1] Wang, Boxin, et al. "DecodingTrust: A Comprehensive Assessment of Trustworthiness in GPT Models." **NeurIPS 2023 DB Track Outstanding Paper Award**.
>
> We also note that novelty assessments are inherently subjective; for instance, Reviewer CxQp explicitly highlighted that our paper has a novel focus.

---

> > ### Author Response · Authors · 2025-11-26
> >
> > > Q2. Overlap with existing observability/agent-profiling tools. In industry there are many tools and platforms that provide observability and profiling for LLM-agent systems: for example, platforms tracking latency, token cost, tool call breakdowns, and more. (OpenTelemetry, LangFuse, datadog etc) This raises the question as what does this paper’s benchmark provide beyond what these existing tools or frameworks already do?
> >
> > A2: Our contribution is **not** another observability or profiling tool, but a **benchmark**. Existing platforms such as OpenTelemetry, Langfuse, or Datadog help developers monitor their own applications (latency, tokens, errors, etc.), while AgentRace defines a **standardized suite of frameworks, workflows, datasets, and metrics** to enable **controlled, apples-to-apples comparison** of LLM agent frameworks. In other words, those tools answer “what is my system doing?”; AgentRace answers “how do different frameworks behave under the same workloads, and what design choices drive efficiency differences?”.
> >
> > To make this distinction concrete, the table below contrasts the functionality of AgentRace with typical observability tools:
> >
> > | Dimension                         | AgentRace (this paper)                                      | Langfuse                      | Datadog         | OpenTelemetry               |
> > |----------------------------------|--------------------------------------------------------------|-------------------------------|-------------------------------|-----------------------------|
> > | Primary goal                     | Benchmark efficiency of LLM agent frameworks                 | Observability for LLM apps    | Monitoring & observability    | Telemetry standard          |
> > | Scope of evaluation              | 7 frameworks × 3 workflows × 6 datasets in one pipeline      | One instrumented app          | One instrumented app/service  | Any instrumented app        |
> > | Standardized workloads/datasets  | Yes                                                          | No                            | No                            | No                          |
> > | Unified workflows (ReAct/RAG/MoA)| Yes                                                          | No                            | No                            | No                          |
> > | Unified tool implementations     | Yes                                                          | No                            | No                            | No                          |
> > | LLM token metrics                | Yes                                                          | Yes                           | Yes                           | Yes                         |
> > | LLM latency/execution time       | Yes                                                          | Yes                           | Yes                           | Yes                         |
> > | Tool-level efficiency metrics    | Yes                                                          | Yes                           | Yes                           | Yes                         |
> > | Multi-agent communication size   | Yes                                                          | No                            | No                            | No                          |
> > | Scalability experiments (#agents)| Yes                                                          | No                            | No                            | No                          |
> > | Accuracy/correctness reported    | Yes                                                          | No                            | No                            | No                          |
> > | Cross-framework comparison       | Yes                                                          | No                            | No                            | No                          |

---

> > > ### Author Response · Authors · 2025-11-26
> > >
> > > > Q3. The paper does not appear to consider system-level optimisation concerns, e.g., caching of tokens, optimized/common toolset for fair comparison etc
> > >
> > > A3: The frameworks do not provide system-level optimizations such as caching of tokens. Whenever a toolkit or functionality was missing, we adopted **a minimal and consistent implementation** that mirrors common usage patterns without framework-specific optimization, e.g., adopting a popular toolkit with a simple implementation. For example, the PDF reader is realized through a minimal integration of `PyPDF2.PdfReader` within the frameworks’ tool classes. We summarize the implemented tools in Appendix E of the revised paper.
> > >
> > > We acknowledge that framework-specific implementations may influence results. However, we view such behaviors as integral to each framework's practical performance characteristics. Elements such as tool invocation, error recovery, and task orchestration reflect real-world robustness. Therefore, our benchmark intentionally captures these differences to represent each framework's actual end-to-end efficiency rather than an idealized abstraction.
> > >
> > > Additionally, we have added experiments to demonstrate the efficiency difference due to the implementation difference. In Appendix B.9 of the revised paper, we compare the performance of different tools. To isolate the impact of concrete tool implementations, we compare tools implemented by different frameworks against our own implementations on the same input dataset with 200 queries. As shown below, tool implementations vary substantially across frameworks: even for identical inputs, the same tool (e.g., XLSX reader, figure loader, code executor) can differ by more than an order of magnitude in runtime, depending on the framework’s internal design. This indicates that tool choice is not merely an engineering detail, but a key performance factor that can significantly affect the efficiency and responsiveness of multi-agent systems. Moreover, our implementations are typically the most lightweight, demonstrating that they introduce minimal overhead beyond the underlying operations.
> > >
> > >
> > >
> > > **Tool Comparison**
> > >
> > > | Framework   | PDF loader | CSV reader | XLSX reader | Text file reader | Doc reader | MP3 loader | Code executor |
> > > |-------------|------------|------------|-------------|------------------|------------|------------|----------------|
> > > | langchain   | 0.04005    | 0.0178     | 0.90453     | 0.78435          | 0.00586    | 4.657798   | 0.002355       |
> > > | autogen     | x          | x          | x           | x                | x          | x          | 0.0000752      |
> > > | agentscope  | x          | x          | x           | x                | x          | 5.9112     | 0.05335        |
> > > | crewai      | x          | x          | x           | x                | x          | x          | 0.0000752      |
> > > | llamaindex  | 0.011795   | 0.0562     | 0.34445     | 0.0010935        | 0.0026685  | 1.747      | 0.0001185      |
> > > | phidata     | x          | 0.00086    | x           | 0.0019865        | x          | x          | 0.000865       |
> > > | pydantic    | x          | x          | x           | x                | x          | x          | 0.0000752      |
> > > | ours        | 0.006545   | 0.00813    | 0.25137     | 0.0002695        | 0.01206    | 1.63566    | 0.0000752      |
> > >
> > > We also analyze efficiency discrepancies between our independently implemented GAIA prompt and the framework-native prompts on successful cases, as summarized in Table below. Since the Agentscope prompt is embedded and cannot be modified, the comparison is restricted to LangChain and LlamaIndex. The results show that our prompt can reduce both the total number of tokens and the end-to-end latency by roughly 25%. This further highlights that prompt design is an important factor for improving the efficiency of multi-agent systems.
> > >
> > > **GAIA Prompt Comparison**
> > >
> > > | Model                 | Prompt token | Output token | Total token | Total Time |
> > > |-----------------------|--------------|--------------|-------------|------------|
> > > | llamaindex_our_prompt | 10835.26     | 412.84       | 11248.11    | 28.1029    |
> > > | llamaindex            | 10888.17433  | 582.111      | 11470.282   | 36.1454    |
> > > | langchain_our_prompt  | 3389.34      | 253.97       | 3543.31     | 8.333      |
> > > | langchain             | 3982.695     | 360.955      | 4343.645    | 12.54203333 |

---

> > > > ### Author Response · Authors · 2025-11-26
> > > >
> > > > > Q4. The metrics measured (runtime, token count, communication size) are conventional, and many of the conclusions drawn (e.g., “longer prompts lead to greater cost”, “tool orchestration adds latency”) are intuitive or not surprising. Thus the contribution of “new insight” is limited.
> > > >
> > > >
> > > > A4: We aim to evaluate the efficiency of different LLM agent frameworks, and we use the typical metrics to reflect the efficiency. We do not claim novelty on the used datasets and metrics.
> > > >
> > > > We agree with your opinion that several findings are not surprising. However, they are fundamental for understanding the efficiency of LLM agent frameworks, so we do not remove them. We need to let people know the important facts about the frameworks. Moreover, we have provided the underlying mechanisms to show the in-depth analysis for the insights. We have added some additional insights, including
> > > > - Larger memory windows do not necessarily improve accuracy and can substantially degrade efficiency (Insight 7 in the revised paper).
> > > > - The scalability of misc overhead in multi-turn reasoning is architecture-dependent, emerging when agent frameworks link context growth to repeated aggregation and parsing operations (Insight 9 in the revised paper).
> > > > - Token usage and accuracy are not strongly correlated, and spending more tokens or LLM calls does not reliably lead to better correctness (insight 11 in the revised paper).
> > > > - Different LLM-agent frameworks exhibit varying levels of adaptability to SLMs. Some frameworks remain robust when deployed with SLMs, whereas others fail to perform the task effectively. (Insight 12 in the revised paper).
> > > > - Different frameworks adopt distinct tool implementations and prompt designs, which can substantially impact efficiency. (Insight 14 in the revised paper).
> > > >
> > > > In our current version, we have 14 insights and 15 underlying mechanisms in total, while we believe that only 3 of them are not surprising (insight 1, 2 and 7 in our revised paper).

---

### Official Review · Reviewer_mKnN · 2025-10-30

**Soundness:** 3
**Presentation:** 3
**Contribution:** 3
**Rating:** 6
**Confidence:** 4

**Summary:**

This paper introduces AgentRace, a benchmark platform specifically designed to evaluate the efficiency of LLM agent frameworks. This is presented as the first systematic benchmark focusing on efficiency, filling a gap left by prior benchmarks that mostly emphasized task success or reasoning correctness. The authors evaluate 7 different LLM agent frameworks on a diverse set of tasks and workflows, measuring metrics such as execution time, token usage, and communication overhead while ensuring functional correctness. Through comprehensive experiments, the paper uncovers previously undocumented sources of inefficiency in these agent frameworks and distills 9 key insights along with 12 underlying mechanisms that explain bottlenecks. In addition to identifying problems, the work provides actionable suggestions such as using selective memory summarization, prompt compression, and choosing more efficient architectures to improve agent performance. Importantly, the authors are releasing the AgentRace benchmark suite and all experimental results to the community, which will enable other researchers and practitioners to reproduce the findings and evaluate new frameworks on the same footing.

**Strengths:**

1. Addresses a Critical Gap: The paper tackles the under-explored dimension of efficiency in LLM agent frameworks. It is explicitly positioned as the first benchmark to systematically evaluate LLM agents’ efficiency, complementing prior work that mostly focused on accuracy or functionality.
2. Insightful Findings with Root-Cause Analysis: The paper doesn’t just report performance numbers; it delivers insights and digs into why certain frameworks behave inefficiently.
3. Actionable Guidance for Improvement: Alongside identifying issues, the paper offers concrete suggestions to improve efficiency.
4. Reproducibility and Open Resource: The authors emphasize reproducibility. The benchmark platform is implemented to run experiments with a single command, and they plan to open-source the entire AgentRace suite and share all results.
5. Clarity and Organization: Despite the technical nature of the benchmark, the paper is clearly written and well-structured.

**Weaknesses:**

1. Limited Theoretical Novelty: The paper’s primary contribution is an empirical benchmark and analysis, rather than a new algorithm or a novel model. While benchmarks are valuable (especially one addressing a clear need), some might argue that the work is more of a practical evaluation study than a conceptual breakthrough. The insights, although useful, stem from measurements and engineering analysis rather than from developing new learning methods or theoretical insights. The impact therefore hinges on how useful the community finds the benchmark and whether these efficiency issues spur further research.
2. Some Insights Are Expected or Confirmatory: A number of the findings, in retrospect, align with intuition. For instance, it’s not surprising that calling a large LLM (like GPT-4) dominates the runtime in any agent workflow – the paper indeed confirms that LLM inference is the primary bottleneck in all frameworks. Similarly, it’s known that adding lots of unnecessary text to prompts will slow things down and increase token costs, which the paper identifies in certain frameworks. In summary, some insights confirm known best practices (like “don’t use overly verbose prompts” or “avoid deep abstraction stacks when speed matters”). This doesn’t negate their value – documenting and quantifying these issues is important – but it means the surprise factor or conceptual novelty of some insights is limited.
3. Possible Implementation Biases: The authors strove for fairness by implementing missing functionalities themselves for each framework. While this is commendable and largely necessary, it also introduces some risk: the performance of a framework in the benchmark might be influenced by the authors’ custom implementation of a tool or workflow for that framework. For example, if Framework A didn’t originally support a certain tool and the authors added it in Python with minimal optimization, Framework A might appear slower partly due to that implementation rather than its core design. It’s a tricky issue – not a clear flaw. The authors could perhaps discuss this source of uncertainty more, but it remains a caveat of doing a multi-framework comparison.

**Questions:**

1. You mention that ~50% of the benchmark functionalities were custom-implemented to ensure all frameworks support all tasks. Could you elaborate on how you maintained fairness and consistency in these implementations? For instance, when a framework lacked a certain tool (say a PDF reader or search), did you implement it in a minimal way in that framework’s style? How do we know that any performance differences observed aren’t due to an inadvertently less optimized implementation on one framework versus another?
2. Your default experiments use GPT-4 (the “GPT-4o” API) as the core LLM for all agents, and you note additional experiments with other models in Appendix B.5. Did those additional experiments reveal any meaningful differences in the efficiency insights? For example, when using a smaller model like LLaMA or an open-source model, do the same frameworks still show the same relative inefficiencies (e.g., the same ones are slower or consume more tokens)? I’m curious if any frameworks are particularly optimized for certain model types or sizes. In short, to what extent do you think the findings are model-agnostic versus tied to using a very large model like GPT-4?
3. The agent ecosystem is rapidly evolving. How do you envision AgentRace staying up-to-date? For example, if LangChain or other frameworks introduce significant efficiency improvements next month, is the idea that the benchmark repository will incorporate those changes for future users? Also, might you include newly released frameworks (or future versions of the current ones) in subsequent evaluations? Essentially, I’m asking if AgentRace is intended as a one-off study or a continuously maintained benchmark suite. A related question: have you considered an easy way for external contributors to add their own framework to AgentRace for evaluation (which would be great for community adoption)?

---

> ### Author Response · Authors · 2025-11-26
>
> Dear Reviewer *mKnN*,
>
> We sincerely thank you for the valuable feedback and helpful suggestions. Below, we address each concern point-by-point.
>
>
>
> > Q1. Limited Theoretical Novelty: The paper’s primary contribution is an empirical benchmark and analysis, rather than a new algorithm or a novel model. While benchmarks are valuable (especially one addressing a clear need), some might argue that the work is more of a practical evaluation study than a conceptual breakthrough. The insights, although useful, stem from measurements and engineering analysis rather than from developing new learning methods or theoretical insights. The impact therefore hinges on how useful the community finds the benchmark and whether these efficiency issues spur further research.
>
>
> A1: We understand the concern regarding novelty. However, we believe that designing an easy-to-use benchmark platform, conducting a comprehensive assessment, and providing insights are also important contributions. For example, DecodingTrust [1] received NeurIPS 2023 Outstanding Datasets and Benchmarks Paper Award, which conducted a comprehensive assessment of trustworthiness in GPT models. We focus on surfacing system-level insights rather than proposing new datasets or metrics. We respectfully argue that **a systematic and reproducible assessment of existing frameworks** under a unified and extensible benchmark is itself a valuable contribution—particularly in the rapidly evolving domain of LLM agents.
>
> [1] Wang, Boxin, et al. "DecodingTrust: A Comprehensive Assessment of Trustworthiness in GPT Models." **NeurIPS 2023 DB Track Outstanding Paper Award**.
>
> We also note that novelty assessments are inherently subjective; for instance, Reviewer CxQp explicitly highlighted that our paper has a novel focus.
>
> > Q2. Some Insights Are Expected or Confirmatory: A number of the findings, in retrospect, align with intuition. For instance, it’s not surprising that calling a large LLM (like GPT-4) dominates the runtime in any agent workflow – the paper indeed confirms that LLM inference is the primary bottleneck in all frameworks. Similarly, it’s known that adding lots of unnecessary text to prompts will slow things down and increase token costs, which the paper identifies in certain frameworks. In summary, some insights confirm known best practices (like “don’t use overly verbose prompts” or “avoid deep abstraction stacks when speed matters”). This doesn’t negate their value – documenting and quantifying these issues is important – but it means the surprise factor or conceptual novelty of some insights is limited.
>
> A2: We agree with your opinion that several findings are not surprising. However, as you mentioned, they are very important for understanding the efficiency of LLM agent frameworks, so we do not want to remove them. We have added some additional insights, including
> - Larger memory windows do not necessarily improve accuracy and can substantially degrade efficiency (Insight 7 in the revised paper).
> - The scalability of misc overhead in multi-turn reasoning is architecture-dependent, emerging when agent frameworks link context growth to repeated aggregation and parsing operations (Insight 9 in the revised paper).
> - Token usage and accuracy are not strongly correlated, and spending more tokens or LLM calls does not reliably lead to better correctness (insight 11 in the revised paper).
> - Different LLM-agent frameworks exhibit varying levels of adaptability to SLMs. Some frameworks remain robust when deployed with SLMs, whereas others fail to perform the task effectively. (Insight 12 in the revised paper).
> - Different frameworks adopt distinct tool implementations and prompt designs, which can substantially impact efficiency. (Insight 14 in the revised paper).
>
> In our current version, we have 14 insights and 15 underlying mechanisms in total, while we think only 3 of these insights are not so surprising (insight 1, 2 and 7 in our revised paper).

---

> > ### Author Response · Authors · 2025-11-26
> >
> > > Q3. Possible Implementation Biases: The authors strove for fairness by implementing missing functionalities themselves for each framework. While this is commendable and largely necessary, it also introduces some risk: the performance of a framework in the benchmark might be influenced by the authors’ custom implementation of a tool or workflow for that framework. For example, if Framework A didn’t originally support a certain tool and the authors added it in Python with minimal optimization, Framework A might appear slower partly due to that implementation rather than its core design. It’s a tricky issue – not a clear flaw. The authors could perhaps discuss this source of uncertainty more, but it remains a caveat of doing a multi-framework comparison.
> >
> >
> > A3: To reduce bias, whenever a toolkit or functionality was missing, we adopted **a minimal and consistent implementation** that mirrors common usage patterns without framework-specific optimization, e.g., adopting a popular toolkit with a simple implementation. For example, the PDF reader is implemented through a minimal integration of `PyPDF2.PdfReader` within the frameworks’ tool classes. We summarize the implemented tools in Tables 51-54 (Appendix E) of the revised paper.
> >
> >
> > Additionally, we have added experiments to demonstrate the efficiency difference due to the implementation difference. In Appendix B.9 of the revised paper, we compare the performance of different tools. To isolate the impact of concrete tool implementations, we compare tools implemented by different frameworks against our own implementations on the same input dataset with 200 queries. As shown below, tool implementations vary substantially across frameworks: even for identical inputs, the same tool (e.g., XLSX reader, figure loader, code executor) can differ by more than an order of magnitude in runtime, depending on the framework’s internal design. This indicates that tool choice is not merely an engineering detail, but a key performance factor that can significantly affect the efficiency and responsiveness of multi-agent systems. Moreover, our implementations are typically the most lightweight, demonstrating that they introduce minimal overhead beyond the underlying operations.
> >
> >
> >
> > **Tool Comparison**
> >
> > | Framework   | PDF loader | CSV reader | XLSX reader | Text file reader | Doc reader | MP3 loader | Code executor |
> > |-------------|------------|------------|-------------|------------------|------------|------------|----------------|
> > | langchain   | 0.04005    | 0.0178     | 0.90453     | 0.78435          | 0.00586    | 4.657798   | 0.002355       |
> > | autogen     | x          | x          | x           | x                | x          | x          | 0.0000752      |
> > | agentscope  | x          | x          | x           | x                | x          | 5.9112     | 0.05335        |
> > | crewai      | x          | x          | x           | x                | x          | x          | 0.0000752      |
> > | llamaindex  | 0.011795   | 0.0562     | 0.34445     | 0.0010935        | 0.0026685  | 1.747      | 0.0001185      |
> > | phidata     | x          | 0.00086    | x           | 0.0019865        | x          | x          | 0.000865       |
> > | pydantic    | x          | x          | x           | x                | x          | x          | 0.0000752      |
> > | ours        | 0.006545   | 0.00813    | 0.25137     | 0.0002695        | 0.01206    | 1.63566    | 0.0000752      |
> >
> > We also analyze efficiency discrepancies between our independently implemented GAIA prompt and the framework-native prompts on successful cases, as summarized in Table below. Since the Agentscope prompt is embedded and cannot be modified, the comparison is restricted to LangChain and LlamaIndex. The results show that our prompt can reduce both the total number of tokens and the end-to-end latency by roughly 25\%. This further highlights that prompt design is an important factor for improving the efficiency of multi-agent systems.
> >
> > **GAIA Prompt Comparison**
> >
> > | Model                 | Prompt token | Output token | Total token | Total Time |
> > |-----------------------|--------------|--------------|-------------|------------|
> > | llamaindex_our_prompt | 10835.26     | 412.84       | 11248.11    | 28.1029    |
> > | llamaindex            | 10888.17433  | 582.111      | 11470.282   | 36.1454    |
> > | langchain_our_prompt  | 3389.34      | 253.97       | 3543.31     | 8.333      |
> > | langchain             | 3982.695     | 360.955      | 4343.645    | 12.54203333 |

---

> > > ### Author Response · Authors · 2025-11-26
> > >
> > > > Q4. Your default experiments use GPT-4 (the “GPT-4o” API) as the core LLM for all agents, and you note additional experiments with other models in Appendix B.5. Did those additional experiments reveal any meaningful differences in the efficiency insights? For example, when using a smaller model like LLaMA or an open-source model, do the same frameworks still show the same relative inefficiencies (e.g., the same ones are slower or consume more tokens)? I’m curious if any frameworks are particularly optimized for certain model types or sizes. In short, to what extent do you think the findings are model-agnostic versus tied to using a very large model like GPT-4?
> > >
> > >
> > > A4: We have added experiments with two additional relatively small models, **Qwen3-Next-80B-A3B-Instruct** and **GLM-Z1-32B**. From these results, our earlier insights remain applicable: the behavior observed with these models closely matches our previous findings. For example, LLM inference continues to dominate end-to-end runtime (Insight 1), and tools that search for figure-related content still introduce disproportionately high latency (Insight 2). This supports our claim that the insights primarily reflect **framework properties**, which generalize across different backbone models. The full results are reported in Appendix B.7.2 of the revised paper; a subset is shown below.
> > >
> > > **Consistency of Token Consumption and Latency Across Repeated GAIA Experiments Using the Qwen3-Next-80B-A3B-Instruct Model**
> > >
> > > |                   | LangChain      | AutoGen       | AgentScope        | CrewAI          | LlamaIndex       | Phidata         | PydanticAI       |
> > > | ----------------- | -------------- | ------------- | ----------------- | --------------- | ---------------- | --------------- | ---------------- |
> > > | Prompt Tokens     | 1749.53±385.94 | 1611.04±37.30 | 30625.53±10718.28 | 6460.77±1239.88 | 63091.83±6747.67 | 5840.53±2429.54 | 22459.01±5579.84 |
> > > | Completion Tokens | 1229.91±93.81  | 901.54±31.80  | 2198.02±919.46    | 714.32±187.93   | 1191.05±57.85    | 932.41±323.51   | 998.50±175.72    |
> > > | Total Tokens      | 2979.44±446.92 | 2512.58±13.96 | 32823.55±11622.51 | 7175.09±1314.15 | 64282.88±6727.64 | 6772.93±2719.86 | 23457.51±5727.77 |
> > > | LLM Time          | 8.21±0.97      | 6.58±0.38     | 19.22±5.52        | 8.59±1.06       | 15.83±1.32       | 8.79±2.23       | 7.97±0.96        |
> > > | Tool Time         | 0.0007±0.0002  | 7.82±2.67     | 2.67±0.18         | 1.45±0.33       | 2.54±0.54        | 0.72±0.25       | 1.26±0.32        |
> > > | Total Time        | 8.30±0.99      | 10.26±0.91    | 21.97±5.70        | 10.10±0.99      | 27.72±1.65       | 11.06±2.31      | 9.26±1.15        |
> > >
> > > **Consistency of Token Consumption and Latency Across Repeated GAIA Experiments Using the GLM-Z1-32B Model**
> > >
> > > |                   | LangChain        | AutoGen        | AgentScope       | CrewAI           | PydanticAI      |
> > > | ----------------- | ---------------- | -------------- | ---------------- | ---------------- | --------------- |
> > > | Prompt Tokens     | 2282.20±925.89   | 1623.11±16.58  | 9559.79±667.28   | 13003.68±3132.60 | 7342.68±923.11  |
> > > | Completion Tokens | 11326.93±4660.23 | 4686.52±892.59 | 9926.80±2115.86  | 2735.74±634.10   | 3162.13±222.13  |
> > > | Total Tokens      | 13609.13±5507.32 | 6309.65±880.64 | 19486.60±2782.31 | 15739.43±3729.82 | 10504.80±843.93 |
> > > | LLM Time          | 294.32±132.55    | 153.52±72.42   | 226.10±28.11     | 51.58±11.60      | 103.88±42.14    |
> > > | Tool Time         | 0.0021±0.0013    | 2.27±2.06      | 0.72±0.30        | 3.10±0.46        | 0.66±0.28       |
> > > | Total Time        | 294.44±132.59    | 155.96±70.56   | 226.89±28.16     | 54.93±12.00      | 104.56±42.16    |
> > >
> > > We also experimented with smaller models such as **Llama-3-7B** and **Mistral-7B-Instruct**. However, these models frequently failed on GAIA and related tasks due to limited capability, yielding very low success rates and preventing us from drawing meaningful additional insights.

---

> > > > ### Author Response · Authors · 2025-11-26
> > > >
> > > > > Q5. The agent ecosystem is rapidly evolving. How do you envision AgentRace staying up-to-date? For example, if LangChain or other frameworks introduce significant efficiency improvements next month, is the idea that the benchmark repository will incorporate those changes for future users? Also, might you include newly released frameworks (or future versions of the current ones) in subsequent evaluations? Essentially, I’m asking if AgentRace is intended as a one-off study or a continuously maintained benchmark suite. A related question: have you considered an easy way for external contributors to add their own framework to AgentRace for evaluation (which would be great for community adoption)?
> > > >
> > > > A5: Thank you for your suggestion. AgentRace is **not** intended as a one-off snapshot, but as a **continuously maintainable benchmark suite**. The codebase is already modularized into Data / Agent / Framework / Analysis modules, so updating to new framework versions (e.g., a more efficient LangChain release) typically amounts to bumping a version and, if needed, adjusting a thin adapter.
> > > >
> > > > Moreover, we also explicitly intend the repository to be **extensible and community-driven**. We will (i) document a simple adapter interface for adding a new framework (implementing a small set of standard hooks: initialize agent, run ReAct/RAG/MoA, register tools), and (ii) welcome external pull requests that add frameworks or update versions. This way, AgentRace can serve both as a frozen reference configuration for reproducible comparison in this paper and as a **living benchmark** that the community can extend as agent frameworks evolve.

---

### Official Review · Reviewer_CxQp · 2025-10-31

**Soundness:** 3
**Presentation:** 3
**Contribution:** 3
**Rating:** 4
**Confidence:** 4

**Summary:**

This paper introduces AgentRace, a benchmark platform specifically designed to evaluate the efficiency of LLM agent frameworks. AgentRace provides a modular, extensible system for controlled and reproducible comparisons of runtime, communication, and tool latency across 7 frameworks, 3 workflows and 5 datasets. Through systematic experiments, the authors identify insights and underlying mechanisms that cause inefficiencies and propose corresponding optimization suggestions.

**Strengths:**

- Novel focus: Shifts benchmarking from “how correct” to “how efficient,” addressing a real deployment bottleneck;
- Systematic design: Modular architecture covering frameworks, workflows, datasets, and metrics;
- Comprehensive analysis: Covers diverse efficiency dimensions (inference, tool latency, communication, scalability);

**Weaknesses:**

- Task coverage: The tested agent tasks (GAIA, HumanEval, MMLU) may not sufficiently stress long-horizon reasoning or multi-step planning. Including multi-turn, dynamic environments (e.g., ScienceWorld, WebArena) would strengthen claims about scalability and real-world efficiency;
- Variable control: It’s unclear whether comparisons are made on successful or failed cases. Efficiency should ideally be measured conditional on correctness to ensure fairness;
- Correlation analysis missing: The paper qualitatively observes token–accuracy or token–time relationships but doesn’t quantify them. Computing a Pearson correlation coefficient would make the relationship concrete;
- Related work: There are some papers focusing on the efficiency of agents [1,2], which the authors should discuss in Related Work section;

[1] Efficient Agents: Building Effective Agents While Reducing Cost. https://arxiv.org/pdf/2508.02694

[2] OPTIMA: Optimizing Effectiveness and Efficiency for LLM-Based Multi-Agent System. https://arxiv.org/pdf/2410.08115

**Questions:**

See the Weaknesses part.

---

> ### Author Response · Authors · 2025-11-26
>
> Dear Reviewer *CxQp*,
>
> We sincerely thank you for the valuable feedback and helpful suggestions. Below, we address each concern point-by-point.
>
> > Q1. - Task coverage: The tested agent tasks (GAIA, HumanEval, MMLU) may not sufficiently stress long-horizon reasoning or multi-step planning. Including multi-turn, dynamic environments (e.g., ScienceWorld, WebArena) would strengthen claims about scalability and real-world efficiency.
>
> A1: We appreciate the reviewer’s suggestion regarding long-horizon and dynamic-environment benchmarks. In addition to our original text- and image-based datasets, we have now incorporated experiments on the ScienceWorld benchmark. The results are available in Section B.3 of the revised paper, and we also put part of the results below. From the results, our previous insights still apply to this dataset. Additionally, since ScienceWorld is an interactive environment that requires multiple steps, the efficiency advantages of specific frameworks are more obvious. We also observe that the tool-output aggregation and parsing in LlamaIndex can take a significant amount of time during execution. This issue is barely noticeable when the number of reasoning rounds is small, but it becomes increasingly prominent as the rounds grow and the context expands (as shown in Figure 7 of the revised paper).
>
> **ScienceWorld**
>
> | Category | Type            | LangChain                   | AgentScope                  | CrewAI                      | LlamaIndex                  | Phidata                     | PydanticAI                  |
> |----------|------------------|-----------------------------|------------------------------|------------------------------|------------------------------|------------------------------|------------------------------|
> | **Token** | Prompt          | 163844.6 ± 10233.6          | 177881.9 ± 16973.9           | 64194.4 ± 28803.6            | 131962.6 ± 3138.8           | 92710.4 ± 5904.6            | 120264.6 ± 8372.3            |
> | **Token** | Output          | 2409.3 ± 426.7              | 1698.6 ± 126.7               | 473.6 ± 252.9                | 1373.1 ± 39.1               | 1362.9 ± 127.5              | 1582.5 ± 28.0                |
> | **Token** | Total           | 166253.9 ± 10640.5          | 179580.5 ± 17099.1           | 64668.0 ± 29056.1            | 133335.8 ± 3177.9           | 94073.3 ± 6028.8            | 121847.1 ± 8394.7            |
> | **Time**  | llm             | 77.385 ± 14.858             | 77.637 ± 8.703               | 23.304 ± 8.138               | 55.284 ± 6.154              | 47.635 ± 7.586              | 54.456 ± 2.584               |
> | **Time**  | observe         | 0.002 ± 0.000               | 0.004 ± 0.001                | 0.007 ± 0.001                | 0.002 ± 0.001               | 0.000 ± 0.001               | 0.003 ± 0.001               |
> | **Time**  | execute action  | 2.014 ± 0.099               | 2.144 ± 0.096                | 0.241 ± 0.015                | 2.830 ± 0.099               | 1.946 ± 0.235               | 2.488 ± 0.128               |
> | **Time**  | embedding       | 14.777 ± 1.737              | 18.548 ± 1.142               | 11.396 ± 0.837               | 31.138 ± 2.500              | 15.819 ± 2.751              | 23.878 ± 4.387              |
> | **Time**  | total           | 101.095 ± 16.223            | 98.920 ± 8.026               | 35.29 ± 8.654                | 317.573 ± 7.120             | 68.424 ± 8.301              | 80.220 ± 1.345              |
>
> > Q2. Variable control: It’s unclear whether comparisons are made on successful or failed cases. Efficiency should ideally be measured conditional on correctness to ensure fairness.
>
> A2: We thank the reviewer for raising this point. In our current setup, efficiency metrics (runtime, token count, communication size) are computed **per query over all cases that are successfully finished**, regardless of correctness. Hard failures that never produce an answer (e.g., repeated malformed tool calls) are treated as missing and not mixed into averages. This matches a practitioner’s perspective: **in deployment, one pays for all attempts, not only the ones that output the correct answer**.
>
> Moreover, we have added results restricted to correct cases in Tables 43 of the revised paper. From these results, our insights on successfully completed cases still apply when we condition on correctness. Additionally, as we will explain in our response to the next question, we find that the Pearson correlation between token count and accuracy is low, indicating that there is no clear difference in token usage between correct and incorrect cases.

---

> > ### Author Response · Authors · 2025-11-26
> >
> > > Q3. - Correlation analysis missing: The paper qualitatively observes token–accuracy or token–time relationships but doesn’t quantify them. Computing a Pearson correlation coefficient would make the relationship concrete;
> >
> > A3: Thank you for your suggestions. We have quantified the relationship between token consumption and task accuracy by computing the Pearson correlation coefficient in Section B.4 of the revised paper. The results are presented below.
> >
> > For token-time relationship, datasets that involve tools with inherently long processing times (e.g., MMLU with RAG retrieval and OK-VQA with image-processing tools) tend to yield smaller correlation coefﬁcients. In contrast, datasets where LLM calls dominate the execution process (e.g., AlpacaEval and HumanEval) exhibit noticeably stronger correlations.
> >
> > For token-accuracy relationship, the correlation between token consumption and accuracy is generally weak across all framework–dataset pairs (all $|r| < 0.35$). Moreover, most coefficients are negative, indicating that within a given framework successful queries tend to use slightly fewer tokens than failed ones. This suggests that simply ''trying harder'' with more steps and longer prompts does not systematically improve correctness; on the contrary, harder instances often trigger longer trajectories that still end in failure. We have included the results in Section B.5.2 and B.5.3 of the revised paper.
> >
> >
> > **Token–Time Pearson Correlation**
> >
> > | Dataset        | LangChain | AutoGen  | AgentScope | CrewAI  | LlamaIndex | Phidata | PydanticAI |
> > |----------------|-----------|----------|------------|---------|------------|---------|------------|
> > | **GAIA**       | 0.7554    | 0.11     | 0.7355     | 0.8745  | 0.6273     | 0.7821  | 0.6276     |
> > | **OK-VQA**     | -         | -0.0463  | 0.4527     | 0.295   | 0.0676     | 0.0284  | 0.0779     |
> > | **ScienceWorld** | 0.9558  | -        | 0.9426     | 0.9581  | 0.9902     | 0.9265  | 0.8784     |
> >
> >
> > **Token–Accuracy Pearson Correlation**
> >
> > | Dataset          | Lanchain | AutoGen  | AgentScope | CrewAI   | LlamaIndex | Phidata  | PydanticAI |
> > |------------------|----------|----------|------------|----------|------------|----------|------------|
> > | **GAIA**         | -0.1200  | -0.0123  | -0.0884    | -0.2026  | -0.0371    | -0.0551  | 0.2408     |
> > | **OK-VQA**       | -        | -0.1215  | -0.2054    | -0.1796  | -0.0440    | -0.1249  | -0.1447    |
> > | **ScienceWorld** | -0.2730  | -        | -0.3113    | -0.0237  | -0.0919    | -0.1485  | -0.1485    |
> >
> > > Q4.  - Related work: There are some papers focusing on the efficiency of agents [1,2], which the authors should discuss in Related Work section;
> >
> >
> > A4: We appreciate the reviewer pointing out the two relevant papers. We have incorporated a discussion of studies on efficient agents in Section 2.1. EFFICIENT AGENTS [1] analyzes the effectiveness-efficiency trade-off of agent pipelines and introduces ''cost-of-pass'' as a metric to quantify computational cost. OPTIMA [2] focuses on multi-agent settings and evaluates token efficiency and communication overhead enabled by optimized training strategies. Although these studies do not provide general-purpose LLM agent frameworks, they highlight the importance of systematically measuring efficiency—reinforcing the motivation of our work to benchmark efficiency across widely used agent frameworks.}
> >
> > [1] Efficient Agents: Building Effective Agents While Reducing Cost. https://arxiv.org/pdf/2508.02694
> > [2] OPTIMA: Optimizing Effectiveness and Efficiency for LLM-Based Multi-Agent System. https://arxiv.org/pdf/2410.08115

---

### Official Review · Reviewer_Vc4t · 2025-11-01

**Soundness:** 2
**Presentation:** 3
**Contribution:** 2
**Rating:** 6
**Confidence:** 3

**Summary:**

This paper introduces AgentRace, a novel benchmark for evaluating the efficiency (runtime, cost, communication) of LLM agent frameworks, arguing existing benchmarks focus too heavily on effectiveness. The modular platform tests 7 frameworks (e.g., LangChain, AutoGen) across 3 workflows (ReAct, RAG, MoA) and 5 datasets, measuring execution time, token use, communication size, and accuracy. The authors identify key efficiency bottlenecks, such as LLM inference (worsened by verbose prompts), varied tool latency, overlooked RAG retrieval costs, and framework-dependent scalability in multi-agent setups.

**Strengths:**

Important and Timely Problem: The paper addresses a critical and underexplored gap in LLM agent evaluation. As agents move from research prototypes to production systems, efficiency (latency, cost, and scalability) becomes a first-order concern. This work provides the first comprehensive, public benchmark to systematically study this.

Comprehensive and Well-Designed Benchmark: The AgentRace platform is a significant contribution. Its modular design (Data, Agent, Framework, Analysis) is sound and extensible. The coverage of 7 major frameworks and 3 core agent workflows (ReAct, RAG, MoA) allows for a broad and relevant comparison. The use of a unified instrumentation layer (tracers and loggers) is a good methodological choice for ensuring fair comparisons.

Actionable Insights: The paper does not just present leaderboards; it delivers concrete, actionable insights for developers. By distinguishing between high-level insights and specific underlying mechanisms, the authors provide a clear diagnosis of why certain frameworks are inefficient and offer clear targets for optimization.

**Weaknesses:**

De-prioritization of Task Accuracy: The paper's primary focus is efficiency, and it explicitly states that accuracy is not the main focus (Appendix B.1). However, the reported accuracy on complex tasks like GAIA is very low (e.g., 0.1-0.2 range in Table 5). This raises questions about the meaningfulness of the efficiency metrics. For example, is a framework's low latency on a task simply because it failed quickly? The paper would be stronger if it explored the relationship between efficiency and correctness more deeply (e.g., does high token consumption correlate with more attempts to solve the problem, even if it ultimately fails?).

Limited Model Diversity: The vast majority of the paper's insights are derived from experiments using a relatively small model pool. Efficiency metrics like latency, cost, and even tool-use behavior are highly dependent on the underlying model. The insights drawn from a powerful, closed-source API model may not generalize to smaller, open-source models that might be run locally. The paper includes a small experiment with Claude-3 (Appendix B.5) but only on two frameworks and one dataset, which is insufficient to address this.

Confounding Variables from Framework-Specific Implementations: The "Bugs and Features" section (Appendix D) is very insightful but also highlights that many of the observed efficiency differences may stem from implementation-specific "bugs" or design choices rather than the inherent architecture of the framework itself. For example, CrewAI failing to call all agents (D.4), LlamaIndex failing on tool calls (D.5), or AgentScope's tools calling OpenAI (D.3). This makes a true "apples-to-apples" comparison of framework overhead difficult. The benchmark is, in practice, evaluating the authors' specific implementation of these workflows on top of the frameworks as much as the frameworks themselves.

**Questions:**

Could the authors further elaborate on the relationship between efficiency and accuracy?

The insights are based on a powerful (and relatively slow) gpt-4o model. How would the primary bottlenecks (Insight 1) shift when using smaller, open-source, or locally-hosted models?

---

> ### Author Response · Authors · 2025-11-26
>
> Dear Reviewer Vc4t,
>
> We sincerely thank you for the valuable feedback and helpful suggestions. Below, we address each concern point-by-point.
>
> > Q1: The reported accuracy on complex tasks like GAIA is very low (e.g., 0.1-0.2 range in Table 5). This raises questions about the meaningfulness of the efficiency metrics. For example, is a framework's low latency on a task simply because it failed quickly? The paper would be stronger if it explored the relationship between efficiency and correctness more deeply (e.g., does high token consumption correlate with more attempts to solve the problem, even if it ultimately fails?).
>
> A1: The low accuracy on GAIA is expected in our setting. Our evaluation strictly follows each framework’s **default** tool-use mechanism, without any framework-specific tuning, prompt engineering, or handcrafted hints. This aligns with our goal of measuring **framework-level behavior** under realistic, out-of-the-box usage. In contrast, many GAIA leaderboard submissions rely on customized agents, task-specific prompt templates, or hardcoded routines, which naturally yield higher accuracy but conflate framework design with application-specific engineering.
>
> We appreciate the suggestion to explore the link between efficiency and correctness more deeply. In the revised paper, we have added two sets of analyses:
>
> 1. **Pearson correlation between efficiency and accuracy.**
>    For GAIA (text-based), OK-VQA (image-based), and ScienceWorld (multi-turn), we compute the Pearson correlation between accuracy and both the number of tokens and the number of LLM calls, over **all** completed cases (successful and failed). The results (also reported in Section B.5.2) are summarized below.
>
>    **Token–Accuracy Pearson Correlation**
>
>    | Dataset          | LangChain | AutoGen  | AgentScope | CrewAI   | LlamaIndex | Phidata  | PydanticAI |
>    |------------------|-----------|----------|------------|----------|------------|----------|------------|
>    | **GAIA**         | -0.1200   | -0.0123  | -0.0884    | -0.2026  | -0.0371    | -0.0551  | 0.2408     |
>    | **OK-VQA**       | –         | -0.1215  | -0.2054    | -0.1796  | -0.0440    | -0.1249  | -0.1447    |
>    | **ScienceWorld** | -0.2730   | –        | -0.3113    | -0.0237  | -0.0919    | -0.1485  | -0.1485    |
>
>    **Rounds–Accuracy Pearson Correlation**
>
>    | Dataset          | LangChain | AutoGen | AgentScope | CrewAI   | LlamaIndex | Phidata  | PydanticAI |
>    |------------------|-----------|---------|------------|----------|------------|----------|------------|
>    | **GAIA**         | -0.1008   | -0.1211 | -0.1508    | -0.2456  | 0.0053     | -0.0789  | 0.1868     |
>    | **OK-VQA**       | –         | –       | -0.2093    | -0.2237  | -0.0406    | -0.0317  | -0.0515    |
>    | **ScienceWorld** | -0.2640   | –       | -0.3115    | 0.0034   | -0.3141    | -0.0564  | -0.1566    |
>
>    Across all framework–dataset pairs, the correlations are generally weak (all $|r| < 0.35$), and most are negative. This indicates that, within a given framework, failed queries often consume *more* tokens and reasoning rounds than successful ones. In other words, frameworks are not “fast because they fail quickly”; harder instances typically trigger longer trajectories that still end in failure.
>
> 2. **Accuracy–efficiency tradeoff via memory size.**
>    We also study the accuracy–efficiency tradeoff from the perspective of built-in historical memory in CrewAI, which stores past conversations for future LLM calls. We vary the memory window size from 1 to the maximum (i.e., all history retained for the current task) and report the results in Appendix B.1. As the memory window grows, token consumption increases steadily (more context tokens), while accuracy first improves slightly and then declines. This shows that a moderate amount of memory can help performance, but excessive memory leads to substantially higher token cost and reduced efficiency without improving accuracy, highlighting a concrete design knob for balancing correctness and efficiency.
>
> Together, these analyses demonstrate that our efficiency metrics are not merely capturing frameworks that fail fast; instead, failures often incur higher cost, and design choices such as memory length materially affect the accuracy–efficiency tradeoff.

---

> ### Author Response · Authors · 2025-11-26
>
> > Q2: Limited Model Diversity: The vast majority of the paper's insights are derived from experiments using a relatively small model pool. Efficiency metrics like latency, cost, and even tool-use behavior are highly dependent on the underlying model. The insights drawn from a powerful, closed-source API model may not generalize to smaller, open-source models that might be run locally. The paper includes a small experiment with Claude-3 (Appendix B.5) but only on two frameworks and one dataset, which is insufficient to address this.
>
> A2: We have added experiments with two additional relatively small models, **Qwen3-Next-80B-A3B-Instruct** and **GLM-Z1-32B**. From these results, our earlier insights remain applicable: the behavior observed with these models closely matches our previous findings. For example, LLM inference continues to dominate end-to-end runtime (Insight 1), and tools that search for figure-related content still introduce disproportionately high latency (Insight 2). This supports our claim that the insights primarily reflect **framework properties**, which generalize across different backbone models. The full results are reported in Appendix B.7.2 of the revised paper; a subset is shown below.
>
> **Consistency of Token Consumption and Latency Across Repeated GAIA Experiments Using the Qwen3-Next-80B-A3B-Instruct Model**
>
> |                   | LangChain      | AutoGen       | AgentScope        | CrewAI          | LlamaIndex       | Phidata         | PydanticAI       |
> | ----------------- | -------------- | ------------- | ----------------- | --------------- | ---------------- | --------------- | ---------------- |
> | Prompt Tokens     | 1749.53±385.94 | 1611.04±37.30 | 30625.53±10718.28 | 6460.77±1239.88 | 63091.83±6747.67 | 5840.53±2429.54 | 22459.01±5579.84 |
> | Completion Tokens | 1229.91±93.81  | 901.54±31.80  | 2198.02±919.46    | 714.32±187.93   | 1191.05±57.85    | 932.41±323.51   | 998.50±175.72    |
> | Total Tokens      | 2979.44±446.92 | 2512.58±13.96 | 32823.55±11622.51 | 7175.09±1314.15 | 64282.88±6727.64 | 6772.93±2719.86 | 23457.51±5727.77 |
> | LLM Time          | 8.21±0.97      | 6.58±0.38     | 19.22±5.52        | 8.59±1.06       | 15.83±1.32       | 8.79±2.23       | 7.97±0.96        |
> | Tool Time         | 0.0007±0.0002  | 7.82±2.67     | 2.67±0.18         | 1.45±0.33       | 2.54±0.54        | 0.72±0.25       | 1.26±0.32        |
> | Total Time        | 8.30±0.99      | 10.26±0.91    | 21.97±5.70        | 10.10±0.99      | 27.72±1.65       | 11.06±2.31      | 9.26±1.15        |
>
> **Consistency of Token Consumption and Latency Across Repeated GAIA Experiments Using the GLM-Z1-32B Model**
>
> |                   | LangChain        | AutoGen        | AgentScope       | CrewAI           | PydanticAI      |
> | ----------------- | ---------------- | -------------- | ---------------- | ---------------- | --------------- |
> | Prompt Tokens     | 2282.20±925.89   | 1623.11±16.58  | 9559.79±667.28   | 13003.68±3132.60 | 7342.68±923.11  |
> | Completion Tokens | 11326.93±4660.23 | 4686.52±892.59 | 9926.80±2115.86  | 2735.74±634.10   | 3162.13±222.13  |
> | Total Tokens      | 13609.13±5507.32 | 6309.65±880.64 | 19486.60±2782.31 | 15739.43±3729.82 | 10504.80±843.93 |
> | LLM Time          | 294.32±132.55    | 153.52±72.42   | 226.10±28.11     | 51.58±11.60      | 103.88±42.14    |
> | Tool Time         | 0.0021±0.0013    | 2.27±2.06      | 0.72±0.30        | 3.10±0.46        | 0.66±0.28       |
> | Total Time        | 294.44±132.59    | 155.96±70.56   | 226.89±28.16     | 54.93±12.00      | 104.56±42.16    |
>
> We also experimented with smaller models such as **Llama-3-7B** and **Mistral-7B-Instruct**. However, these models frequently failed on GAIA and related tasks due to limited capability, yielding very low success rates and preventing us from drawing meaningful additional insights.

---

> ### Author Response · Authors · 2025-11-26
>
> > Q3: Confounding Variables from Framework-Specific Implementations: The "Bugs and Features" section (Appendix D) is very insightful but also highlights that many of the observed efficiency differences may stem from implementation-specific "bugs" or design choices rather than the inherent architecture of the framework itself. For example, CrewAI failing to call all agents (D.4), LlamaIndex failing on tool calls (D.5), or AgentScope's tools calling OpenAI (D.3). This makes a true "apples-to-apples" comparison of framework overhead difficult. The benchmark is, in practice, evaluating the authors' specific implementation of these workflows on top of the frameworks as much as the frameworks themselves.
>
> A3: We acknowledge that frameworks' own specific implementations can influence results. However, we view such behaviors as integral to each framework's practical performance characteristics. Elements such as tool invocation, error recovery, and task orchestration reflect real-world robustness. Therefore, our benchmark intentionally captures these differences to represent each framework's actual end-to-end efficiency rather than an idealized abstraction.
>
> For frameworks that lack a specific implementation or functionality, we deliberately keep our own implementations **minimal and consistent** for fairness. For example, the PDF reader is implemented through a minimal integration of `PyPDF2.PdfReader` within the frameworks’ tool classes. We summarize the implemented tools in Appendix E of the revised paper.
>
> Additionally, we have added experiments to demonstrate the effect of the frameworks’ tool selection and prompt design. To isolate the impact of concrete tool implementations, we compare tools implemented by different frameworks against our own implementations on the same input dataset with 200 queries. As shown below, tool implementations vary substantially across frameworks: even for identical inputs, the same tool (e.g., XLSX reader, figure loader, code executor) can differ by more than an order of magnitude in runtime, depending on the framework’s internal design. This indicates that tool choice is not merely an engineering detail, but a key performance factor that can significantly affect the efficiency and responsiveness of multi-agent systems. Moreover, our implementations are typically the most lightweight, demonstrating that they introduce minimal overhead beyond the underlying operations.
>
>
>
> **Tool Comparison**
>
> | Framework   | PDF loader | CSV reader | XLSX reader | Text file reader | Doc reader | MP3 loader | Code executor |
> |-------------|------------|------------|-------------|------------------|------------|------------|----------------|
> | langchain   | 0.04005    | 0.0178     | 0.90453     | 0.78435          | 0.00586    | 4.657798   | 0.002355       |
> | autogen     | x          | x          | x           | x                | x          | x          | 0.0000752      |
> | agentscope  | x          | x          | x           | x                | x          | 5.9112     | 0.05335        |
> | crewai      | x          | x          | x           | x                | x          | x          | 0.0000752      |
> | llamaindex  | 0.011795   | 0.0562     | 0.34445     | 0.0010935        | 0.0026685  | 1.747      | 0.0001185      |
> | phidata     | x          | 0.00086    | x           | 0.0019865        | x          | x          | 0.000865       |
> | pydantic    | x          | x          | x           | x                | x          | x          | 0.0000752      |
> | ours        | 0.006545   | 0.00813    | 0.25137     | 0.0002695        | 0.01206    | 1.63566    | 0.0000752      |
>
>
> We also analyze efficiency discrepancies between our independently implemented GAIA prompt and the framework-native prompts on successful cases, as summarized in Table below. Since the Agentscope prompt is embedded and cannot be modified, the comparison is restricted to LangChain and LlamaIndex. The results show that our prompt can reduce both the total number of tokens and the end-to-end latency by roughly 25\%. This further highlights that prompt design is an important factor for improving the efficiency of multi-agent systems.
>
> **GAIA Prompt Comparison**
>
> | Model                 | Prompt token | Output token | Total token | Total Time |
> |-----------------------|--------------|--------------|-------------|------------|
> | llamaindex_our_prompt | 10835.26     | 412.84       | 11248.11    | 28.1029    |
> | llamaindex            | 10888.17433  | 582.111      | 11470.282   | 36.1454    |
> | langchain_our_prompt  | 3389.34      | 253.97       | 3543.31     | 8.333      |
> | langchain             | 3982.695     | 360.955      | 4343.645    | 12.54203333 |
>
>
> The results are available in **Appendix B.9** of the revised paper.

---

### Author Response · Authors · 2025-11-26
**Summary of Changes**

Dear Reviewers,

Thank you for your time and effort in reviewing our paper. We appreciate that the reviews recognize our paper as addressing a timely problem (Vc4t, mKnN, cW3P), having a novel focus (CxQp), being comprehensive (Vc4t, CxQp) and systematic (CxQp), providing actionable insights (Vc4t, mKnN), being clearly written (mKnN), and offering good reproducibility (mKnN, cW3P).

In response to the concerns raised, we have made the following revisions (marked in red in the updated manuscript):

- Added 5 additional insights and 3 underlying mechanisms, resulting in **14 insights** and **15 mechanisms** in total.
- Added a discussion of **efficient agents** in Section 2.1.
- Added experiments on an additional dataset, **ScienceWorld**, focusing on multi-step planning (Appendix B.3).
- Added experiments on the effect of **memory size** and the **accuracy–efficiency tradeoff** (Appendix B.1).
- Added **correlation analyses** between token consumption and latency/accuracy, and between reasoning rounds and efficiency (Appendix B.5).
- Added experiments with **smaller LLMs** to demonstrate the generalizability of our insights (Appendix B.7.2).
- Added analysis of the effect of different **tool and prompt implementations** (Appendix B.9).
- Listed the **concrete tool implementations** in detail (Appendix E).

We have put a lot of effort into the rebuttal. We hope that these additional analyses and clarifications satisfactorily address your concerns and that you will kindly reconsider your rating.

Best Regards,

Paper #14734 Authors

---

### Author Response · Authors · 2025-11-27
**Looking forward to your feedback**

Dear Reviewers,

We have put a lot of effort into the rebuttal and revision. We hope you can review it at your earliest convenience so that we can address any further concerns you may have. Thank you very much for your time and effort!!

Best regards,

Authors of Paper #14734

---

### Author Response · Authors · 2025-12-02
**Summary of Revision**

Dear Area Chairs and Reviewers,

Thank you for your time and effort. We have put substantial effort into the rebuttal and revision, including extensive additional experiments to address the concerns raised by the reviewers. We summarize our contributions and revisions below:

The core contributions of the paper are as follows:

- We introduce **AgentRace**, the first benchmark platform explicitly designed to evaluate the **efficiency** of LLM agent frameworks through a modular and extensible architecture.
- We conduct the first **comprehensive cross-framework study**, revealing previously unreported sources of inefficiency.
- We provide **actionable insights** for both practitioners and researchers to optimize the deployment of efficient LLM-based agents.
- We release the full benchmark suite and experimental results, providing a platform to identify the efficiency issues of LLM agents.

Across the reviews, several strengths were consistently highlighted:

- All reviewers emphasized that our work targets a **timely and underexplored problem**, shifting agent evaluation from correctness to **efficiency**, a critical bottleneck for real-world deployment.
- Reviewers *Vc4t*, *CxQp*, and *cW3P* praised the systematic and modular benchmark design, noting its breadth: **7 frameworks**, **3 core workflows** (ReAct, RAG, MoA), and unified instrumentation for fair comparison.
- Reviewers *Vc4t* and *mKnN* highlighted that the paper goes well beyond reporting leaderboards, offering clear **root-cause analyses** and **actionable efficiency insights**.
- Reviewers *mKnN* and *cW3P* recognized the emphasis on reproducibility, the unified testing environment, and the open-source commitment, underscoring AgentRace as a **valuable systems contribution**.

Together, these assessments reflect a strong consensus: **AgentRace fills an important gap, is methodically designed, and delivers practical insights for advancing efficient LLM agent systems.**

---

> ### Author Response · Authors · 2025-12-02
>
> Alongside these strengths, reviewers also raised several concerns. Our responses are as follows:
>
> | Reviewer id(s)                  | Concerns                                                         | Our response                                                                                                                                                                                                                                                          |
> | ------------------------------- | ---------------------------------------------------------------- | --------------------------------------------------------------------------------------------------------------------------------------------------------------------------------------------------------------------------------------------------------------------- |
> | Vc4t W1&Q1, CxQp W2             | Efficiency should ideally be measured conditional on correctness | Added results restricted to correct cases in **Table 43** of the revised paper.<br>Added experiments on **memory size effects** and the **accuracy–efficiency tradeoff** (Appendix B.1).                                                                              |
> | Vc4t W2&Q2, mKnN Q2             | Limited model diversity                                          | Added experiments using **smaller LLMs** to demonstrate the generalizability of our insights (Appendix B.7.2).                                                                                                                                                        |
> | Vc4t W3, mKnN W3&Q1, cW3P W3&Q2 | Tool implementation effects influencing cross-framework fairness | Clarified that such behaviors are part of each framework's practical performance characteristics.<br>Added analyses on the effects of different **tool** and **prompt implementations** (Appendix B.9) and provided the concrete tool details.                        |
> | CxQp W1                         | Limited task coverage                                            | Added experiments on the **ScienceWorld** dataset focusing on multi-step planning (Appendix B.3).                                                                                                                                                                     |
> | CxQp W3                         | Missing correlation analysis                                     | Added **correlation analyses** between token consumption, latency and accuracy, as well as between reasoning rounds and efficiency (Appendix B.5).                                                                                                                    |
> | CxQp W4                         | Related work on efficient agents                                 | Added discussion of **efficient agents** in Section 2.1.                                                                                                                                                                                                              |
> | mKnN W1, cW3P W1                | Limited theoretical novelty                                      | Clarified in replies to Reviewers mKnN and cW3P that the novelty stems from building a **systematic, reproducible benchmark** and conducting a **comprehensive empirical study**, even if some findings align with intuition.                                         |
> | mKnN W2, cW3P W4                | Insights seen as intuitive or confirmatory                       | Acknowledged that some insights are expected, while highlighting newly added, non-obvious findings that deepen the analysis. <br> Added **five additional insights** and **three additional mechanisms**, resulting in **14 insights** and **15 mechanisms** overall. |
> | mKnN Q3                         | Concerns about benchmark maintenance and scalability             | Explained that AgentRace is structured as a **continuously maintainable** benchmark suite and is scalable to additional frameworks in our reply to Reviewer mKnN.                                                                                                     |
> | cW3P W2&Q1                      | Potential overlap with existing observability tools              | Clarified that AgentRace is a **benchmark**, not a monitoring tool, enabling standardized **cross-framework comparison** in our reply to Reviewer cW3P.                                                                                                               |
>
> We hope you can take the above into consideration. Thank you again!
>
> Best Regards,
>
> Paper #14734 Authors

---

### Meta-Review · Area_Chair_fhgf · 2026-01-07

**Summary:**

This paper proposes AgentRace, a benchmark aiming to assess the efficiency of different LLM agent frameworks. Reviewers agree the paper’s goal that benchmarking efficiency in LLM agent frameworks is timely and potentially high-impact, and they value the substantial engineering effort. The main concerns are: (1) only efficiency metrics without reporting correctness results at all in the main body of the paper; (2) limited task coverage; (3) model diversity; (4) limited novelty, and many of the conclusions are considered known and few novel  insights are reviewed.

I have read the paper as well as the rebuttal, my biggest concern, aligning with some reviewers, is the overlook of correctness and over-emphasis on efficiency metrics. While I understand that this benchmark is born to measure efficiency, comparing efficiency without caring about correctness is meaningless in my opinion. For example, a system could be trivially wrong to output short responses while being efficient; talking about efficiency when the accuracy is as low as 0.1 is not that meaningful. Unfortunately, the accuracy of different frameworks is rarely discussed in the main body of the paper, and since efficacy-efficiency often presents a tradeoff, I think their entanglement is worth separate discussion in the paper, otherwise, simply talking efficiency alone is not very useful. Another concern is on the task coverage, as agent frameworks influence more and more on agents nowadays, while most of the tasks included in this benchmark are more like MMLU-style traditional QA tasks. I think this community focuses more on agent framework efficiency in agentic tasks, thus the future practicality of AgentRace is questionable. Therefore, I lean rejection of this paper.

**Reviewer Concerns:**

Concerns addressed:

1. Correctness-conditioning and correlation: added analysis relating efficiency metrics to correctness, and reporting efficiency conditioned on correct cases (reducing the “fast fail” ambiguity).
2. Model diversity: added experiments on additional backbones/sizes to argue bottlenecks persist beyond a single model choice.

Concerns still outstanding:

1. Broad task coverage as detailed above
2. Over-emphasis on efficiency

**Reviewer Scores:**

First, no reviewers explicitly note they will change the score. From my opinion, I think all the reviewers would have maintained their score due to the core outstanding concerns as above raised by the reviewers.

---

> ### Public Comment · ~Zeyu_Qiu1 · 2026-03-24
>
> Dear Area Chair,
>
> We thank the meta-reviewer and all reviewers for their careful reading and constructive feedback.
>
> We fully agree with the central concern raised in the meta-review: efficiency metrics that do not account for correctness can indeed be misleading. In response, we conducted a comprehensive re-analysis in which all efficiency metrics are computed **strictly conditional on correct cases**. Importantly, these results show that our main efficiency findings and relative framework rankings remain consistent when conditioning on correctness. In the revised manuscript, we make this conditioning explicit in the primary metric definitions and position correctness as a prerequisite for meaningful efficiency comparison. We therefore believe that the core concern raised in the meta-review has been substantively addressed.
>
> Regarding task coverage, we agree that no single benchmark can exhaust the full space of agentic behaviors. Our intent was not to be task-exhaustive, but to cover **representative agent-centric interaction patterns** relevant to modern LLM frameworks. Beyond MMLU-style and traditional QA tasks, AgentRace includes:
>
> - tool-integrated complex reasoning (GAIA),
> - executable code generation with programmatic verification (HumanEval),
> - sequential decision-making in a dynamic environment (ScienceWorld, added in revision)
>
> Together, these settings capture tool interaction, executable environments, and long-horizon planning, which we view as core dimensions of agentic behavior at the framework level. We acknowledge that this positioning was not sufficiently emphasized in the original narrative, and we will revise the task taxonomy and framing accordingly.
>
> We appreciate the reviewers’ feedback and hope this clarification helps contextualize the revisions and their impact on the original concerns.
>
>
> Sincerely,
> The Authors of Paper #14734

---

### Decision · Program_Chairs · 2026-01-26

Reject